# D-2-hydroxyglutarate impairs DNA repair through epigenetic reprogramming

Fengchao Lang[1,3], Karambir Kaur[1,3], Haiqing Fu [2], Javeria Zaheer[1], Diego Luis Ribeiro [1], Mirit I. Aladjem [2] & Chunzhang Yang [1]✉

Cancer-associated mutations in IDH are associated with multiple types of human malignancies, which exhibit distinctive metabolic reprogramming, production of oncometabolite D-2-HG, and shifted epigenetic landscape. IDH mutated malignancies are signatured with "BRCAness", highlighted with the sensitivity to DNA repair inhibitors and genotoxic agents, although the underlying molecular mechanism remains elusive. In the present study, we demonstrate that D-2-HG impacts the chromatin conformation adjustments, which are associated with DNA repair process. Mechanistically, D-2-HG diminishes the chromatin interactions in the DNA damage regions via revoking CTCF binding. The hypermethylation of cytosine, resulting from the suppression of TET1 and TET2 activities by D-2-HG, contributes to the dissociation of CTCF from DNA damage regions. CTCF depletion leads to the disruption of chromatin organization around the DNA damage sites, which abolishes the recruitment of essential DNA damage repair proteins BRCA2 and RAD51, as well as impairs homologous repair in the IDH mutant cancer cells. These findings provide evidence that CTCF-mediated chromatin interactions play a key role in DNA damage repair proceedings. Oncometabolites jeopardize genome stability and DNA repair by affecting high-order chromatin structure.

Mutations in the isocitrate dehydrogenase (IDH1/2) are highly frequent genetic abnormalities in human cancerous diseases such as glioma, leukemia, chondrosarcoma, and cholangiocarcinoma[1–3]. Cancer-associated IDH mutants exhibit neomorphic activity, which results in the productivity of oncometabolite D-2-hydroxyglutarate (D-2-HG). D-2-HG competitively inhibits α-ketoglutarate (αKG)-dependent dioxygenases, and hence substantially affects cellular biology and disease progression[4,5]. Impairment of DNA repair is frequently observed in IDH-mutated malignancies. Several pioneering studies revealed the potential connection between D-2-HG and DNA repair pathway. Wang et al. showed that D-2-HG results in sensitivity to alkylating agents by suppressing alkB homolog (ALKBH) DNA repair enzymes[6]. Sulkowski et al. and our previous work revealed that IDH1 mutation sensitizes tumor cells to PARP1 inhibitors by impacting DNA damage repair responses[7,8]. In addition, Sulkowski et al. demonstrated that D-2-HG and other oncometabolites affect DNA repair by affecting histone

demethylation, which establishes "BRCAness" in these cancer cells[9,10]. Due to the dysfunction in the DNA repair pathways, these oncometabolite-driven cancers are sensitive to DNA repair inhibitors, such as therapeutic agents targeting PARP, ATR, and NAMPT[11–13]. However, the precise mechanism of D-2-HG-associated impact on DNA repair pathways is still not completely elucidated.

Shifted epigenetic landscape is closely associated with IDH-mutated malignancies, due to the inhibition of demethylases by D-2-HG. For example, ten-eleven translocation methylcytosine dioxygenase (TET1/2/3) are epigenetic enzymes that are involved in DNA demethylation, gene regulation, DNA repair[14–16], and 3D genome structure[17]. TET1/2 catalyzes the conversion of 5-methylcytosine (5mC) to 5-hydroxymethylcytosine (5hmC), as well as the conversion from 5hmC to 5-formylcytosine (5fC), from 5fC to 5-carboxylcytosine (5caC), respectively. In IDH-mutant cells, D-2-HG inhibits the catalytic activity of TET1/2 and establishes genome-wide DNA hypermethylation[5]. DNA

[1]Neuro-Oncology Branch, Center for Cancer Research, National Cancer Institute, Bethesda, MD, USA. [2]Developmental Therapeutic Branch, Center for Cancer Research, National Cancer Institute, Bethesda, MD, USA. [3]These authors contributed equally: Fengchao Lang, Karambir Kaur. ✉e-mail: yangc2@nih.gov

hypermethylation compromises the binding of methylation-sensitive factors, such as CCCTC-binding factor (CTCF). Flavahan et al. revealed that hypermethylation affects the affinity of CTCF and insulator function, and therefore alters gene transcription[18]. Besides function as an insulator, CTCF is involved in homologous recombination (HR) DNA double-strand break (DSB) repair by facilitating the recruitment of BRCA2 and RAD51[19,20]. However, it remains elusive whether changes in the epigenetic landscape in IDH-mutant cells are relevant to the compromised DNA repair activity.

Several recent advancements revealed that chromatin remodeling facilitates DNA repair, which is mediated by histone modification, histone chaperones and nucleosome remodeling to adjust the accessibility of DNA repair enzymes to the damage sites. DNA repair-related chromatin remodeling often associates with conformational changes in chromatin structure. DNA damage-associated γH2A.X modification spreads to ~1 M bp around the DNA damage regions[21]. The repairment process often accompanies with the enhancement of topologically associating domain (TAD) that overlaps with the γH2A.X loci, suggesting dynamic changes in chromatin loops are necessary for efficient DNA repair[22].

In the present study, we demonstrated that the demethylase activity of TET1/2 is essential for the recruitment of CTCF to DNA damage sites, which assists to resolve the DNA damage. Moreover, CTCF facilitates chromatin remodeling in the DNA damage site, as well as the assembly of HR repair complex comprised of RAD51 and BRCA2. D-2-HG compromises TET1/2 activity as well as CTCF recruitment to DNA DSBs. Overall, our findings demonstrated an epigenetic mechanism that oncometabolite D-2-HG hampers HR DNA repair, through DNA hypermethylation and limited CTCF-guided DNA repair.

## Results

### D-2-HG compromises HR DNA repair via inhibition of TET1/2

Several pioneering studies demonstrate that cancer-associated IDH1 mutations predispose sensitivity to genotoxic stress[7,8,23]. To determine whether the kinetics of DNA repair differ between IDH1$^{WT}$ and IDH1$^{R132H}$, we conducted a time-course experiment measuring the number of γH2A.X foci in DSB Inducible via AsiSI cells (DIvA). We discovered that γH2A.X foci persist longer in IDH1$^{R132H}$ DIvA cells compared to IDH1$^{WT}$ DIvA cells (Supplementary Fig. 1a, b). This impaired DNA damage repair is not attributed to defects in the non-homologous end joining (NHEJ) pathway, as evidenced by the comparable repair efficiency observed in both cell types using the EJ5 or EJ7 NHEJ reporter systems (Supplementary Fig. 1c, d). Further, we confirmed this phenomenon in glioma cell lines, that the acquisition of R132C or R132H variants of IDH1 resulted in remarkably increased formation of γH2A.X puncta in cell nuclei and modification of γH2A.X level when exposed to genotoxic agents such as temozolomide (TMZ) or carmustine (BCNU) (Fig. 1a, b and Supplementary Fig. 2a). The number of γH2A.X puncta is reversible with the presence of IDH mutant neomorphic activity inhibitor ivosidenib (AG-120). Similarly, we identified stronger modification of γH2A.X and phosphorylated ATM (pATM) for cells exposed to oncometabolite D-2-HG after genotoxic agent treatment (Fig. 1c–e and Supplementary Fig. 2b). In addition, the enhancement of γH2A.X, pATM, and DNA fragmentation are reversible with αKG, suggesting that the D-2-HG-associated chemosensitization is through the activity of αKG-dependent dioxygenases (Fig. 1c–e and Supplementary Fig. 2c). The activity of cytosine demethylase TET1/2 is compromised with the pathological level of D-2-HG[5], which may interfere with the DNA repair process due to alterations in the epigenetic status of the chromatin for appropriate DNA repair proceedings. We demonstrated through comet assay that the ectopic expression of TET1 or TET2 relieved the DNA damage with the presence of TMZ and D-2-HG (Fig. 1f and Supplementary Fig. 2c). Further, we generated TET1/2 knockout (KO) cell lines and revealed that TMZ-induced DNA fragmentation is more severe in TET1/2 KO cells. Replenish of TET1/2 rescued the DNA

fragmentation (Fig. 1g and Supplementary Fig. 2d). Immunoblotting assay confirmed that the modification of γH2A.X and pATM are enhanced in TET1/2 KO cells when exposed to TMZ, whereas reversible with the ectopic expression of TET1/2 (Fig. 1h, i). To better understand the role of TET1/2 in D-2-HG-induced "BRCAness," we established an inducible DRGFP reporter system (DD-Sce-I-GR) that can monitor the efficiency of homologous repair (HR) and we discovered that HR is compromised with the presence of D-2-HG, whereas this phenomenon is reversible with of αKG or ectopic expression of TET1/2, suggesting the HR DNA repair is affected due to D-2-HG inhibition on TET1/2 (Fig. 1j, k and Supplementary Fig. 2e, f). Similarly, in cells expressing IDH1$^{R132H}$ variant, the modification of γH2A.X and pATM was reduced with the overexpression of TET1/2 (Fig. 1l and Supplementary Fig. 2g, h). Lastly, γH2A.X puncta are elevated in TET1/2 KO cells, whereas this phenomenon is reversed with TET1/2 overexpression (Fig. 1m, n and Supplementary Fig. 2i, j).

### D-2-HG inhibition on TET1/2 alters DNA damage protein recruitment in DSB sites

DNA repair is a comprehensive process that requires a myriad of DNA repair proteins recruited to the damage site with appropriate spatial and chronological orders[24]. Through a micro-radiation assay, we discovered that the recruitment of HR repair protein BRCA2 or RAD51 to γH2A.X sites was suppressed with the presence of D-2-HG. The phenomenon is reversible with the presence of αKG or overexpression of TET1/2 (Fig. 2a–d). Further, we investigated the recruitment of RAD51 recruitment to γH2A.X in DD-Sce-I-GR DRGFP reporter cells (Fig. 2e and Supplementary Fig. 3a) The phenomenon is also recorded in U251 cells with TMZ treatment (Fig. 2f, g and Supplementary Fig. 3b–e). We found less colocalization event of RAD51 and γH2A.X signal with the presence of D-2-HG, whereas the phenomenon is reversed with αKG or TET1/2 overexpression (Fig. 2g and Supplementary Fig. 3). To better understand the recruitment of BRCA2/RAD51 to the DNA damage site, we performed a ChIP PCR assay to quantify the coverage of BRCA2/RAD51 at the DNA damage site (Fig. 2h). We found that Sce-I induction resulted in elevated coverage of γH2A.X, BRCA2, and RAD51 in the DNA regions −180 or +150 bp loci next to the DSB. With the presence of D-2-HG, the association of γH2A.X to these regions was found enhanced, whereas the coverages of BRCA2 or RAD51 were reduced. The effect of D-2-HG is reversed with αKG. Importantly, the recruitment of BRCA2 and RAD51 is only seen in the proximal regions of the DSB site, but not distal regions in the chromosome such as the loci of *ACTB* (Fig. 2h).

### Cytosine hypermethylation hampers CTCF binding and subsequent assembly of HR complex

The accumulation of D-2-HG leads to genome-wide CpG island hypermethylator phenotype (CIMP) by suppressing TET1/2 activity[25]. The loss of TET1/2 activity leads to loss of 5hmC throughout the genome (Fig. 3a and Supplementary Fig. 4a). Replenishment of TET1 or TET2 recovers the level of 5hmC modification in D-2-HG treated cells or IDH mutant cells (Fig. 3a and Supplementary Fig. 4a, b). To better understand the DNA methylation status in the DNA damage sites, we screened the cytosine methylation status adjacent to the Sce-I DSB site in DRGFP system through MeDIP/hMeDIP. The presence of D-2-HG resulted in a potent accumulation of 5mC in the proximal regions of the Sce-I site, whereas the 5hmC levels were depleted in each locus (Fig. 3b and Supplementary Fig. 4c). The imbalance of 5mC/5hmC could be rectified with the presence of αKG, suggesting the extensive 5mC level is caused by TET1/2 blockade through D-2-HG.

Further, we performed a micro-irradiation assay and confirmed the presence of 5hmC in the site of laser-induced DSB, whereas 5hmC accumulation is not seen with the presence of D-2-HG (Fig. 3c, e). The suppression of 5hmC at the micro-irradiation sites could be rescued with the presence of αKG or TET1/2 overexpression.

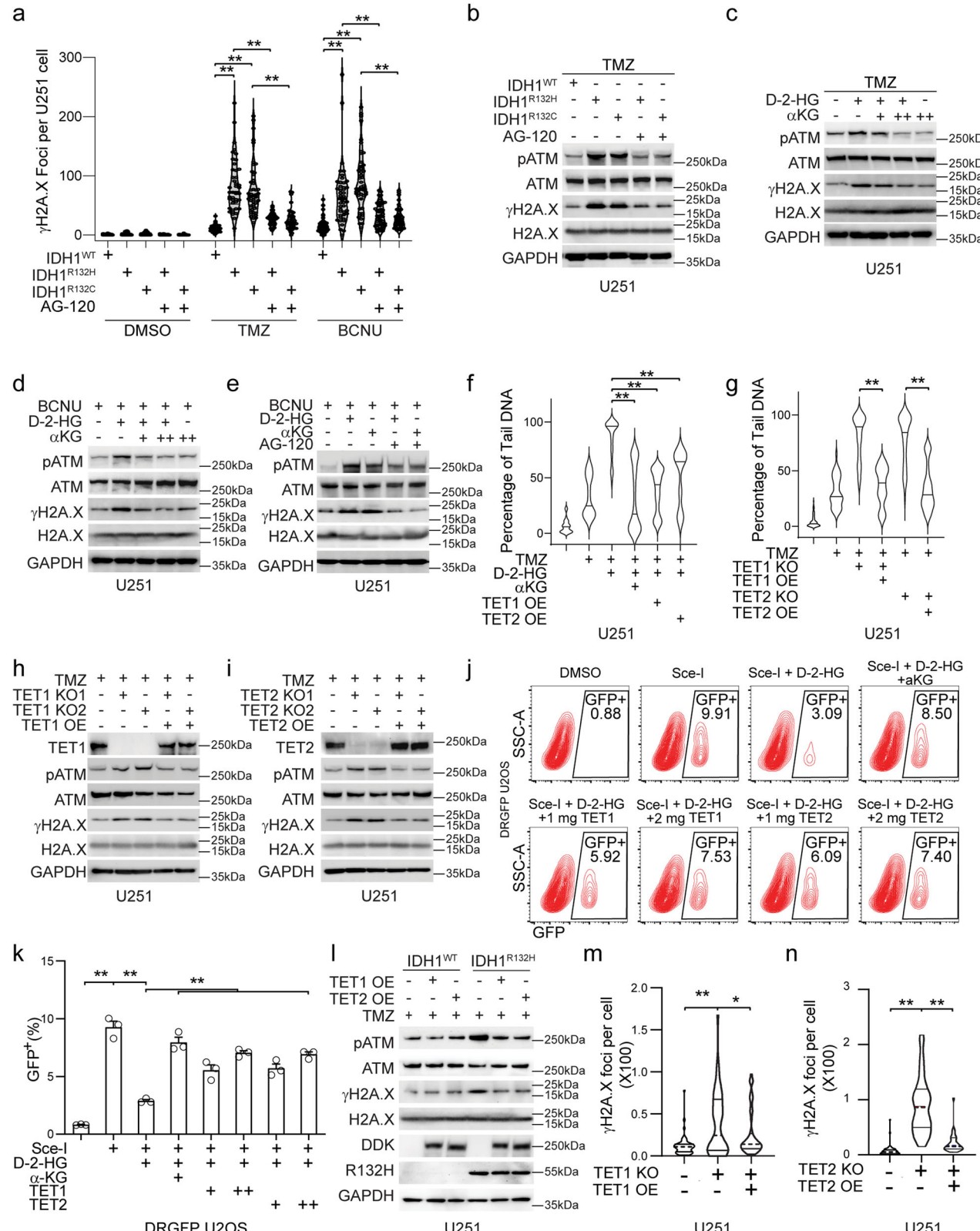

The alteration of CpG methylation status may either directly affect the recruitment of DNA damage elements, or through impact on methylation-sensitive DNA binding proteins. In the present study, we discovered that CTCF, a multifunctional nuclear protein, was affected by the chromatin methylation status during DNA repair. CTCF is recruited to Sce-I-induced DSB site (Supplementary Fig. 4d), which colocalized with γH2A.X signal (Fig. 3d). The presence of D-2-HG

abolished the recruitment of CTCF, which is also reversible with αKG or TET1/2 overexpression (Fig. 3d, f and Supplementary Fig. 4d).

The coincidence of CTCF in DNA damage site suggests a possible role of CTCF to facilitate the recruitment of other DNA repair proteins to the DNA damage loci. To test this hypothesis, we performed a loss-of-function study with small interference RNA targeting CTCF. The loss of CTCF resulted in the disassociation of RAD51 from chromatin with

**Fig. 1 | D-2-HG compromises HR DNA repair via inhibition of TET1/2. a** γH2A.X foci number. U251 cells were treated with DMSO, 150 μM TMZ or 150 μM BCNU for 24 h. Group differences were tested with one-way ANOVA. **p < 0.01. All p < 0.0001; data are mean ± SEM from three independent experiments. **b–e** Activation of DDR. IDH1$^{WT}$, IDH1$^{R132H}$ U251 or U251 cells were treated with 150 μM TMZ ± 5 μM AG-120 (**b**); 150 μM TMZ/150 μM BCNU, 0.5 mM D-2-HG, 1 mM αKG (+), 10 mM αKG (++) for 24 h (**c–e**). **f, g** Comet assay. **f** U251 were treated with 150 μM TMZ, 0.5 mM D-2-HG, 10 mM αKG for 24 h. **p < 0.01. All the indicated p < 0.0001; **g** U251 with TET1/TET2 knockout were treated with DMSO or 150 μM TMZ for 24 h. **p < 0.01. All the indicated p < 0.0001; group differences were tested with one-way ANOVA. Data are mean ± SEM from three independent experiments. **h, i** Western blotting. U251 with control sgRNA, TET1/TET2 knockout, TET1/TET2 knockout with TET1 or TET2 overexpression were treated with 150 μM TMZ for 24 h. **j** Flow cytometry analysis. DD-Sce-I-GR DRGFP U2OS were treated with 0.5 μM Shield1 and 0.2 mM TA, 0.5 mM D-2-HG, 10 mM αKG, TET1 or TET2 overexpression for 24 h. **k** Quantification for (**j**). *p < 0.05, **p < 0.01. D-2-HG vs D-2-HG + 1 μg TET1, p = 0.01. Other p < 0.0001. Group differences were tested with one-way ANOVA. Data are mean ± SEM from three independent experiments. **l** Activation of DDR. U251 IDH1$^{WT}$/IDH1$^{R132H}$ with TET1 or TET2 overexpression were treated with 150 μM TMZ. **m, n** γH2A.X foci number. U251 with TET1/TET2 knockout, TET1/TET2 overexpression was treated with 150 μM TMZ for 24 h. *p < 0.05, **p < 0.01. TET2 KO+ vs TET2 OE, p = 0.0393. All other p < 0.0001. Group differences were tested with one-way ANOVA. Data are mean ± SEM from three independent experiments. Source Data are provided as a Source Data file.

elevated TMZ-induced DNA damage (Supplementary Fig. 4e, f). Further, we discovered that the recruitment of RAD51 to TMZ-induced DNA damage regions is compromised with CTCF depletion (Supplementary Fig. 4g, h). The laser micro-irradiation assay confirmed that BRCA2 or RAD51 are less efficiently recruited to the DNA damage regions when CTCF is genetically silenced (Supplementary Fig. 4i, j). To further illustrate the recruitment process of CTCF, BRCA2 and RAD51 during DNA damage response, we proceeded with a ChIP enrichment approach and revealed that Sce-I-induced DSB leads to a time-dependent enrichment of γH2A.X, CTCF, BRCA2, and RAD51 to the DSB sites (Fig. 3g). We found that shortly after the establishment of the DSB, γH2A.X was first presented at the damaging site. CTCF was found recruited to the DNA damaging sites 1–2 h after Sce-I induction. The presence of BRCA2 and RAD51 was found later which peaked at 4–6 h after the DSB. Importantly, the presence of D-2-HG abolished the recruitment of CTCF, BRCA2, or RAD51, leaving a persistent γH2A.X at the damage site. The application of αKG or TET1/2 overexpression partially rescued the normal recruitment sequence of HR DNA repair elements.

To further understand how D-2-HG impacts the DNA repair pathway in the context of replication stress, we investigated the recruitment of BRCA2/RAD51/CTCF at genomic fragile sites (Fig. 3h). Similar to our findings in Sce-I-induced DSB, D-2-HG resulted in a substantially elevation of Aphidicolin (APH)-induced DNA damage at the fragile sites, highlighted with strong γH2A.X enrichment. The elevation of DNA damage is related to the loss of BRCA2/RAD51/CTCF recruitment at the damaging site. Replenish with αKG or TET1/2 partially rescued the DNA damage at these regions. The alterations in the genetic fragile sites indicate that D-2-HG may prompt chemosensitivity through a TET1/2-related mechanism.

### D-2-HG affects the assembly of HR DNA repair complex through TET1 and TET2 inhibition

The close longitudinal and spatial connection among CTCF, BRCA2, and RAD51 within the DNA damage sites strongly indicates that CTCF contributes to the DNA repair process by facilitating the assembly of DNA repair complex. To test this hypothesis, we visualized and quantified the physical interaction of CTCF-guided HR repair complex through the proximity ligation assay (PLA) assay. We discovered that TMZ resulted in a strong affinity among RAD51, BRCA2, and CTCF. In contrast, the RAD51-CTCF or RAD51-BRCA2 interactions are largely abolished with the presence of D-2-HG, whereas αKG or ectopic expression of TET1/2 partially rescued the RAD51-CTCF or RAD51-BRCA2 interactions (Fig. 4a and Supplementary Fig. 5a, b). Further, TET1/2 is essential in the assembly of the RAD51-CTCF or RAD51-BRCA2 interactions, as the PLA signal foci were found diminished in TET1 KO or TET2 KO cells. Re-expression of TET1/2 resulted in elevated RAD51-CTCF and RAD51-BRCA2 puncta (Fig. 4b and Supplementary Fig. 5c, d). A similar phenomenon was also observed with the BRCA2-CTCF interaction (Fig. 4c, d). Quantitative analysis on PLA foci confirmed the significant impact of D-2-HG and TET1/2 in the physical interactions of

RAD51-CTCF or BRCA2-CTCF, respectively (Fig. 4g, h). The loss of CTCF expression also resulted in limited assembly of the BRCA2/RAD51 HR repair complex (Supplementary Fig. 5e, f). Biochemical tests confirmed the limited formation of DNA repair complex comprised of BRCA2, RAD51, and CTCF (Supplementary Fig. 5g, h). The disruption of BRCA2-RAD51-CTCF interactions after CTCF knockdown further demonstrates the roles of CTCF in organizing DNA repair complex during DNA damage.

Additionally, we investigated the effects of D-2-HG on the formation of CTCF/BRCA2/RAD51 interactions through co-immunoprecipitation assay. We found that TMZ exposure results in the formation of CTCF/BRCA2/RAD51 complex, whereas the protein–protein interaction is compromised in cells carrying IDH1$^{R132H}$ or IDH1$^{R132C}$ variants. The formation of this complex could be restored with AG-120 (Fig. 4i). Similarly, in IDH wild-type cells, D-2-HG diminished the formation of CTCF/BRCA2/RAD51 complex, whereas the protein–protein interaction could be rescued with αKG or TET1/2 overexpression (Fig. 4j).

Lastly, we confirmed that the DNA damage-induced interactions of BRCA2/RAD51/CTCF requires the presence of TET1/2, as the BRCA2-CTCF or RAD51-CTCF affinity is reduced with genetic KO of TET1/2 (Fig. 4k, l and Supplementary Fig. 5i). Ectopic expression of TET1/2 in KO cell lines restored the assembly of BRCA2/RAD51/CTCF complex during DNA damage repair process.

The HR process requires an initial step of end resection to expose the single-strand DNA and ssDNA serves as a platform for the recruitment of repair proteins, such as RPA and RAD51. It has been shown that CTCF may facilitate end resection by interacting with Mre11 and promotes CtIP recruitment to DSB sites[26]. We found that CTCF depletion attenuated the recruitment of RPA to DNA breaks (Supplementary Fig. 6a, b). Overexpression of the TET1 catalytic domain enhanced RPA recruitment, whereas the mutant TET1 catalytic domain failed to restore RPA recruitment in IDH mutant cells (Supplementary Fig. 6a, c). To assess end resection efficiency in IDH1-mutant DIvA cells, we analyzed two DSB sites previously reported[27]. A significant reduction in end resection was observed in IDH mutant cells, as indicated by reduced amplification (Supplementary Fig. 6d, e). Furthermore, the interaction between Mre11 and CtIP was impaired in the presence of D-2-HG (Supplementary Fig. 6f). Notably, overexpression of the TET1 catalytic domain but not the mutant TET1 catalytic domain rescue the Mre11-CtIP interaction during the DNA damage repair process (Supplementary Fig. 6g). In conclusion, IDH mutation may also suppress HR repair through CTCF-guided end resection.

### DNA damage response deficiency in IDH mutant glioma stem cells

To validate the findings in cell lines carrying intrinsic IDH mutants, we investigated the role of TET1/2 in the DNA damage response and tumorigenicity using patient-derived glioma stem cells (GSCs), BT054, and BT142. Overexpression of TET1/2 in these cells resulted in fewer unresolved DNA breaks, as evidenced by a reduction in

γH2A.X puncta and increased RAD51 recruitment to DNA damage sites (Supplementary Fig. 7a–h). These observations were further confirmed by western blot analysis (Supplementary Fig. 7i). Time-lapse imaging following laser micro-irradiation demonstrated that RFP-RAD51 recruitment to DSB sites was significantly delayed or entirely absent in D-2-HG treated U251 and IDH mutant BT054 cells, αKG restored recruitment of RFP-RAD51 (Supplementary Fig. 7j–m).

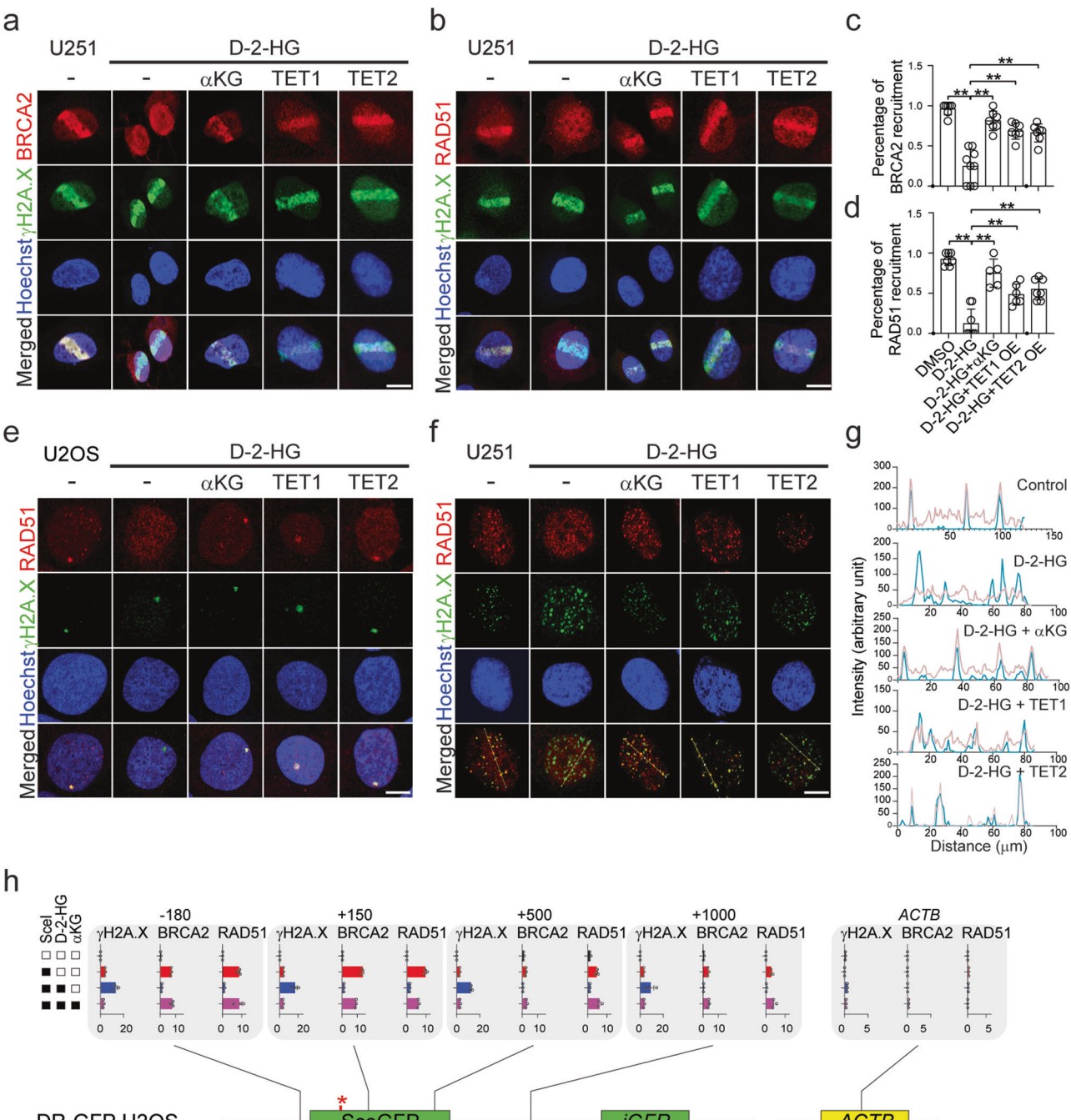

**Fig. 2 | D-2-HG inhibition on TET1/2 alters DNA damage protein recruitment in DNA damage regions. a**, **b** Laser micro-irradiation assay. U251 cells were treated with 0.5 mM D-2-HG, 10 mM αKG or TET1 and TET2 overexpression for 24 h. Scale bar: 5 μm. **c**, **d** Quantification of the cells with BRCA2 or RAD51 recruitment to DSB sites. A total of 40 cells from 5–9 views (**c** DMSO, $n = 7$; D-2-HG, $n = 9$, D-2-HG + αKG, $n = 7$, D-2-HG + TET1, $n = 7$, D-2-HG + TET2, $n = 7$; **d** DMSO, $n = 7$; D-2-HG, $n = 8$, D-2-HG + αKG, $n = 5$, D-2-HG + TET1, $n = 6$, D-2-HG + TET2, $n = 7$) were obtained for quantifications. $**p < 0.01$. **c** D-2-HG vs D-2-HG + αKG, $p < 0.0001$; D-2-HG vs D-2-HG + TET1, $p < 0.0001$; D-2-HG vs D-2-HG + TET2, $p < 0.0001$; **d** D-2-HG vs D-2-HG + αKG, $p < 0.0001$; D-2-HG vs D-2-HG + TET1, $p = 0.0006$; D-2-HG vs D-2-HG + TET2, $p < 0.0001$; Group differences were tested with one-way ANOVA. Data are mean ± SEM from three independent experiments. **e** Immunofluorescence staining shows the induction of γH2A.X and RAD51 puncta in the nucleus. DD-Sce-I-GR DRGFP U2OS cells were treated with 0.5 μM Shield1 and 0.2 mM TA in the presence of 0.5 mM D-2-HG with 10 mM αKG, TET1 or TET2 overexpression for 24 h. Scale bar: 5 μm. **f** Immunofluorescence staining shows the recruitment of RAD51 DSB sites. U251 cells were treated with 150 μM TMZ for 24 h. Scale bar: 5 μm. **g** Line profile analysis for data shown in (**f**). Red, RAD51; cyan, γH2A.X. **h** ChIP-qPCR shows the recruitment of BRCA2, RAD51, and γH2A.X modification around the Sce-I site. DD-Sce-I-GR DRGFP U2OS cells were treated with 0.5 μM Shield1 and 0.2 mM TA in the presence of 0.5 mM D-2-HG with 10 mM αKG. ACTB loci were used as the negative control. Data are presented as mean values ± SEM from three independent experiments. Source Data are provided as a Source Data file.

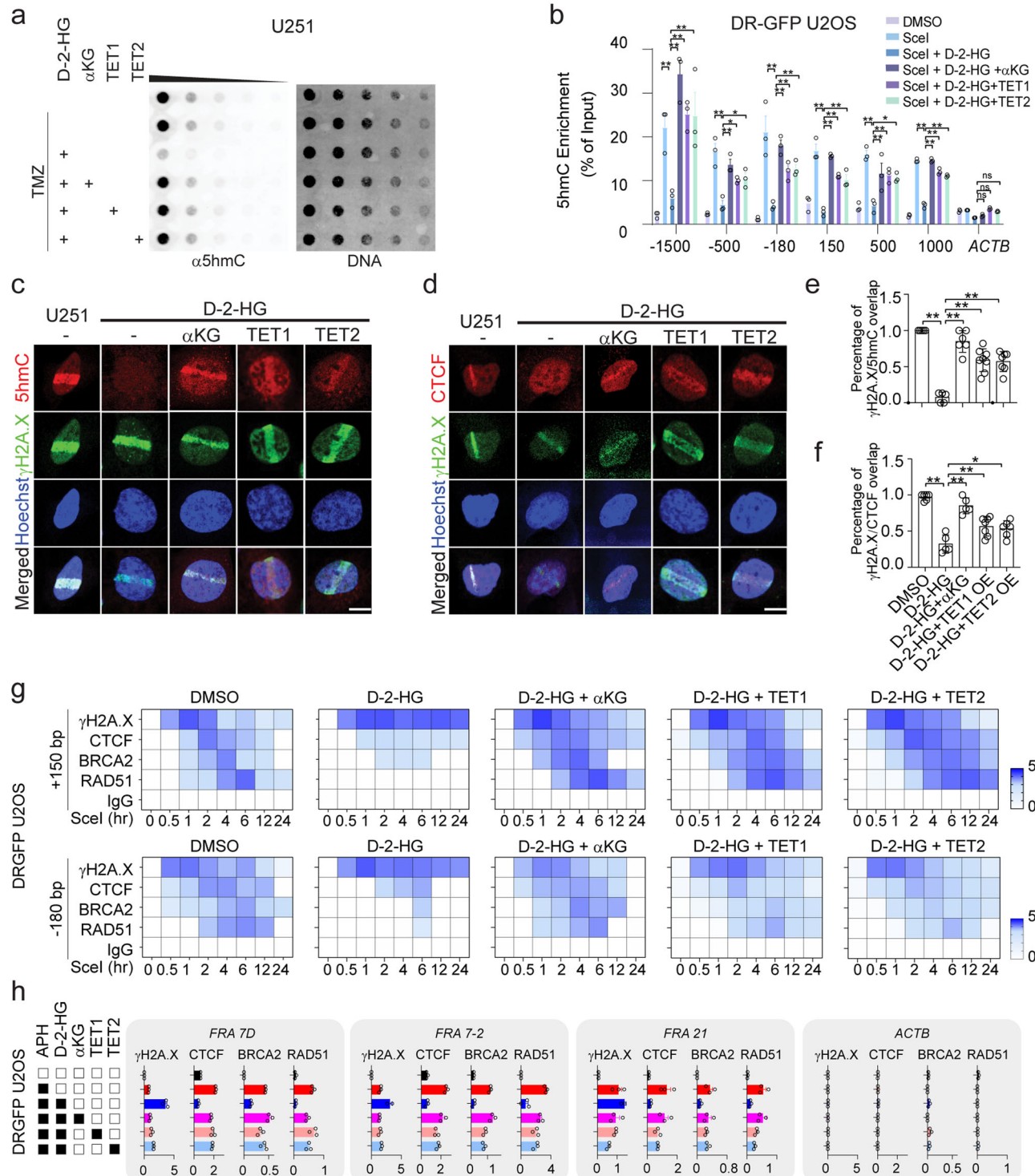

Moreover, IDH mutant GSC cells are more sensitive to TMZ. The TMZ sensitivity was reversed by αKG treatment or TET1/TET2 overexpression (Supplementary Fig. 7n, o). Consistent with this, αKG treatment or TET1/TET2 overexpression reduced apoptosis and enhanced clonality in GSCs, evidently by Annexin V staining and colony formation assay, respectively (Supplementary Fig. 7p–u). AG-120 alleviated TMZ-induced DNA breaks in IDH1-mutant GSC cells (Supplementary Fig. 8a, b). Furthermore, inhibition of mutant IDH1 increased the resistance of GSC cells to TMZ treatment (Supplementary Fig. 8c, 8d), highlighting the connection between IDH mutation and sensitivity to alkylating agent.

## IDH mutation disrupts loop formation around DNA damage sites

IDH mutations result in hypermethylation of H3K9me3, which impairs the recruitment of DNA damage repair proteins to DSB sites[10]. We confirmed a rapid increase in H3K9me3 levels following DSB induction in DRGFP cells (Supplementary Fig. 9a). Notably, the absence of TET1 and TET2 did not fully abolish the rescue effects of α-KG on DDR, suggesting that multiple mechanisms may underlie the inhibitory effects of IDH mutations on DDR (Supplementary Fig. 9b). TADs are the scaffold structure for the γH2A.X modification and DSB repair complex assembling[22]. Flavahan et al. demonstrated that IDH mutations disrupted the insulator function of CTCF at transcription events thus leading to

**Fig. 3 | Cytosine hypermethylation hampers CTCF binding and subsequent recruitment of HR elements. a** 5hmC Dot blot assay. Cells were treated with 150 μM TMZ, 0.5 mM D-2-HG, 10 mM αKG, TET1 or TET2 overexpression for 24 h, 400, 200, 100, 50, 25, 12.5 ng gDNA was used for the assay. 5hmC was monitored with the antibodies. **b** MeDIP assay. DRGFP U2OS cells were treated with 0.5 μM Shield1 and 0.2 mM TA for 2 h. *$p < 0.05$, **$p < 0.01$. ns: non-significant. For −1500 to 1000 locus, Sce-I+ vs D-2-HG, D-2-HG vs D-2-HG + αKG, all $p < 0.0001$; −1500, D-2-HG vs D-2-HG + TET1/TET2, $p < 0.0001$; −500, D-2-HG vs D-2-HG + TET1, $p = 0.0423$; D-2-HG vs D-2-HG + TET2, $p = 0.0230$; −180, D-2-HG vs D-2-HG + TET1, $p = 0.0014$; D-2-HG vs D-2-HG + TET2, $p = 0.0006$; 150, D-2-HG vs D-2-HG + TET1, $p = 0.0010$; D-2-HG vs D-2-HG + TET2, $p = 0.0034$; 500, D-2-HG vs D-2-HG + TET1, $p = 0.0060$; D-2-HG vs D-2-HG + TET2, $p = 0.0156$; 1000, D-2-HG vs D-2-HG + TET1, $p = 0.0024$; D-2-HG vs D-2-HG + TET2, $p = 0.0096$; group differences were tested with one-way ANOVA. Data are mean ± SEM from three independent experiments. **c, d** Laser micro-irradiation. Scale bar: 5 μm. **e, f** Quantification of data shown in (**c**) and (**d**). Over 60 cells were obtained from 5–7 imaging views (**e**, DMSO, $n = 5$; D-2-HG, $n = 5$, D-2-HG + αKG, $n = 6$, D-2-HG + TET1, $n = 8$, D-2-HG + TET2, $n = 7$; **f** DMSO, $n = 6$; D-2-HG, $n = 5$, D-2-HG + αKG, $n = 5$, D-2-HG + TET1, $n = 7$, D-2-HG + TET2, $n = 6$). *$p < 0.05$, **$p < 0.01$. **e** All indicated $p < 0.0001$; **f** DMSO vs D-2-HG, $p < 0.0001$; D-2-HG vs D-2-HG + αKG, $p < 0.0001$; D-2-HG vs D-2-HG + TET1, $p = 0.0069$; D-2-HG vs D-2-HG + TET2, $p = 0.0277$; group differences were tested with one-way ANOVA. Data are mean ± SEM from three independent experiments. **g** ChIP-qPCR assay. DRGFP U2OS cells were treated with 0.5 μM Shield1 and 0.2 mM TA for 0.5, 1, 2, 4, 6, 12, or 24 h. Two genomic regions −180 bp and +150 bp adjacent of DSB sites were monitored. **h** ChIP-qPCR assay targeting FRA7D, FRA7-2, and FRA21. Data are presented as mean ± SEM from three independent experiments. Source Data are provided as a Source Data file.

oncogene activation[18]. Mechanistically, IDH1 mutation impacts CTCF function through methylating cytosine in the CTCF binding motif. We hypothesize that the impairment of CTCF-associated DNA repair process may be relevant to the loss of CTCF chromatin association and shifts in chromatin conformation. To test this, we determined the loop extrusion and TAD maintenance in IDH1^R132H DIvA cells. We recorded significantly elevated γH2A.X and 53BP1 puncta in IDH1^R132H compared to IDH1^WT counterpart upon the same 4-OHT treatment (Supplementary Fig. 9c, d). Chromatin immunoprecipitation (ChIP) assay confirmed the loss of CTCF coverage in the proximity regions of AsiSI sites (DSB1: chr1: 89,458,596–89,458,603; DSB2: chr20: 32,032,086–32,032,093; Fig. 5a, b).

Additionally, we conducted circular chromatin conformation capture sequencing (4C-seq) to assess the DNA contact frequencies flanking the AsiSI-associated DSB sites. We observed remarkable ectopic chromatin interactions around the DSB sites along with the formation of DNA damage, spanning approximately 50 kb to 1 Mb. The ectopic changes were minimally seen in IDH mutant cells (Fig. 5b). CTCF knockdown significantly reduced chromatin interactions around DSB sites (Supplementary Fig. 9e). Overexpression of the TET1 catalytic domain, but not its mutant form, rescued IDH mutation-induced inhibition of CTCF binding at DSB sites and restored chromatin interactions (Supplementary Fig. 9f). Consistent with ChIP-seq finding, IDH1-R132H mutation decreased CTCF affinity for chromatin DNA, as demonstrated by chromatin-bound assays. Notably, TET1 catalytic domain overexpression effectively restored CTCF binding to chromatin DNA (Supplementary Fig. 9g). These data demonstrated that depleted TET activity and loss of CTCF binding are associated with diminished chromatin interactions around the DSB sites.

Further, we measured the cytosine methylation through methylation-specific bisulfite sequencing at six CTCF binding sites adjacent to two AsiSI sites (Supplementary Fig. 9h). We identified substantially elevated cytosine methylation at the location where the CTCF peaks are reduced in IDH1^R132H cells (Fig. 5c and Supplementary Fig. 9i). The ectopic expression of TET1 catalytic domain rescued the recruitment of DDR factors and D-2-HG-induced 5mC hypermethylation at the DSB sites (Fig. 5d, e).

## Discussion
### Cancer metabolic signature reprograms therapeutic vulnerability
Cancer cells reprogram their metabolic routes to meet the increased demands of proliferation. Cancer's distinctive metabolic phenotype further impacts other critical hallmarks such as genetic instability, evading growth suppression, or resisting cell death, to proceed with the malignancy. D-2-HG, a well-characterized oncometabolite that is synthesized through IDH mutant enzyme, serves as a competitive inhibitor for DNA/histone demethylases, triggers genome-wide epigenetic shifts, and hence, prompts the aggressive phenotypes in cancer. For example, D-2-HG-induced hypermethylation supports cancer

stemness through hampering physiological differentiation processes. Moreover, IDH mutation which produces D-2-HG enables the activity of robust mitogens such as PDGFA[28] and PI3K/mTOR/Akt[29,30] to facilitate disease manifestation. In the clinical setting, IDH-mutated glioma demonstrates a more favorable outcome compared to its IDH wild-type counterpart. One contributing factor to this phenomenon is the impairment of DNA repair pathways by the oncometabolite D-2-HG, although the mechanism of inhibition may vary across different tumors and genetic backgrounds[10,31,32]. The compromised DNA damage response in IDH mutant glioma presents a potential treatment opportunity with a DNA repair inhibitor, as these tumors are susceptible to DNA damage[7,8]. Besides these pioneering findings, our study expands the understanding of D-2-HG. Our discovery highlights the critical role of TET1/2 in the HR DNA repair process, which is compromised with pathological levels of D-2-HG (Fig. 1). TET1/2 is functionally relevant to the HR repair, as depletion of TET1/2 activity through either D-2-HG of genetic KO leads to the loss of 5hmC and accumulation of 5mC as well as sensitivity to DNA damage (Figs. 2 and 5). Overall, our findings highlight that appropriate epigenetic status such as cytosine methylation plays a pivotal role in effective and organized DNA repair proceedings (Supplementary Fig. 10).

### CTCF serves as a facilitator for DNA repair through organizing chromatin structure
Our present study highlights the role of CTCF as an essential epigenetic mediator for DNA repair. CTCF is a highly conserved zinc finger protein, which exhibits multiple roles in genome regulation, through transcriptional activation, repression, and insulation[33]. The involvement of CTCF in genomic integrity was previously mentioned in a couple of pioneering studies, that CTCF facilitates the recruitment of HR repair elements to the DSB loci[19,20,26,34]. We and others demonstrated that the chromatin integration of CTCF is affected by the methylation status of the CpG island within the CTCF binding motif[35]. In the context of D-2-HG-associated CpG island hypermethylation, CTCF-associated function, such as the insulator effect, was compromised[18,28]. In the present study, we demonstrated that the D-2-HG led to an accumulation of 5mC in the DSB loci, which compromises the recruitment of CTCF (Fig. 3). Additionally, longitudinal monitoring on DSB loci revealed that CTCF binding is a prerequisite event for the assembly of DNA repair complex involving BRCA2 and RAD51. Mechanistically, CTCF facilitates the recruitment and assembly of BRCA2/RAD51 HR repair complex. D-2-HG blocks CTCF binding and the subsequent establishment of DNA repair complexes during DNA damage response via altering local epigenetic patterns at the DSB (Fig. 4). Further, we discovered that D-2-HG-mediated depletion of CTCF at the DNA damage loci is accompanied by reduced TAD remodeling, which may be critical for ideal DNA repair protein accessibility (Fig. 5). Overall, our finding highlights the central role of CTCF in initiating and organizing the DNA repair process, which is affected by the metabolic and epigenetic status.

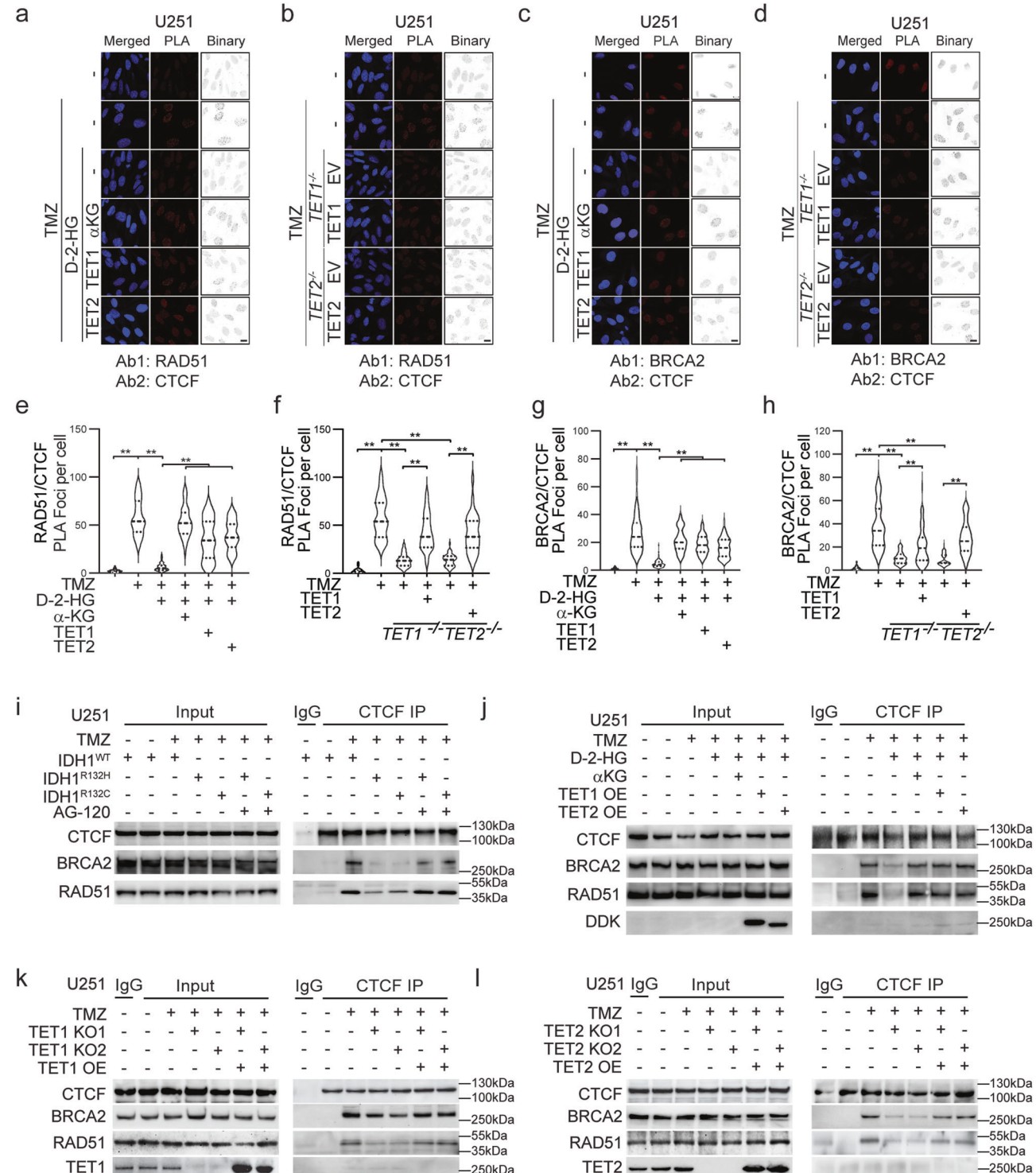

**Fig. 4 | D-2-HG affects the assembly of HR repair complex through TET1 and TET2 inhibition. a–d** PLA assay shows the interactions among CTCF and BRCA2 or RAD51 in the presence of 150 μM TMZ with/without 0.5 mM D-2-HG, 10 mM αKG, TET1 or TET2 overexpression for 24 h. Scale bar: 15 μm. **e–h** Quantification of PLA foci shown in (**a–d**). Over 40 cells among three biological replicates were obtained for quantifications. The statistical significance of differences among groups was tested using the one-way analysis of variance (ANOVA). **$p < 0.01$. **e, g** TMZ− vs TMZ+, DMSO vs D-2-HG, D-2-HG vs D-2-HG + αKG, D-2-HG vs D-2-HG + TET1, D-2-HG vs D-2-HG + TET2, $p < 0.0001$; **f, h** TMZ− vs TMZ+, TET1 KO− vs TET1 KO+, TET1 KO+ vs TET1 OE, TET2 KO− vs TET2 KO+, TET2 KO+ vs TET2 OE, $p < 0.0001$. **i** Co-IP

experiments show the interactions among CTCF, BRCA2, and RAD51 in the U251 IDH1[WT], IDH1[R132H], and IDH1[R132C] cells with/without 10 mM αKG, TET1 or TET2 overexpression. **j** Co-IP experiments show the interactions among CTCF, BRCA2, and RAD51 in U251 cells in the presence of 150 μM TMZ with/without 0.5 mM D-2-HG, 10 mM αKG, TET1 or TET2 overexpression. **k, l** Co-IP experiment shows the interactions between BRCA2 and RAD51 in the absence of TET1 or TET2. TET1 and TET2 overexpression were utilized to evaluate their rescue effects on the knockout consequence. Cells were treated with TMZ (150 μM) for 24 h. Source Data are provided as a Source Data file.

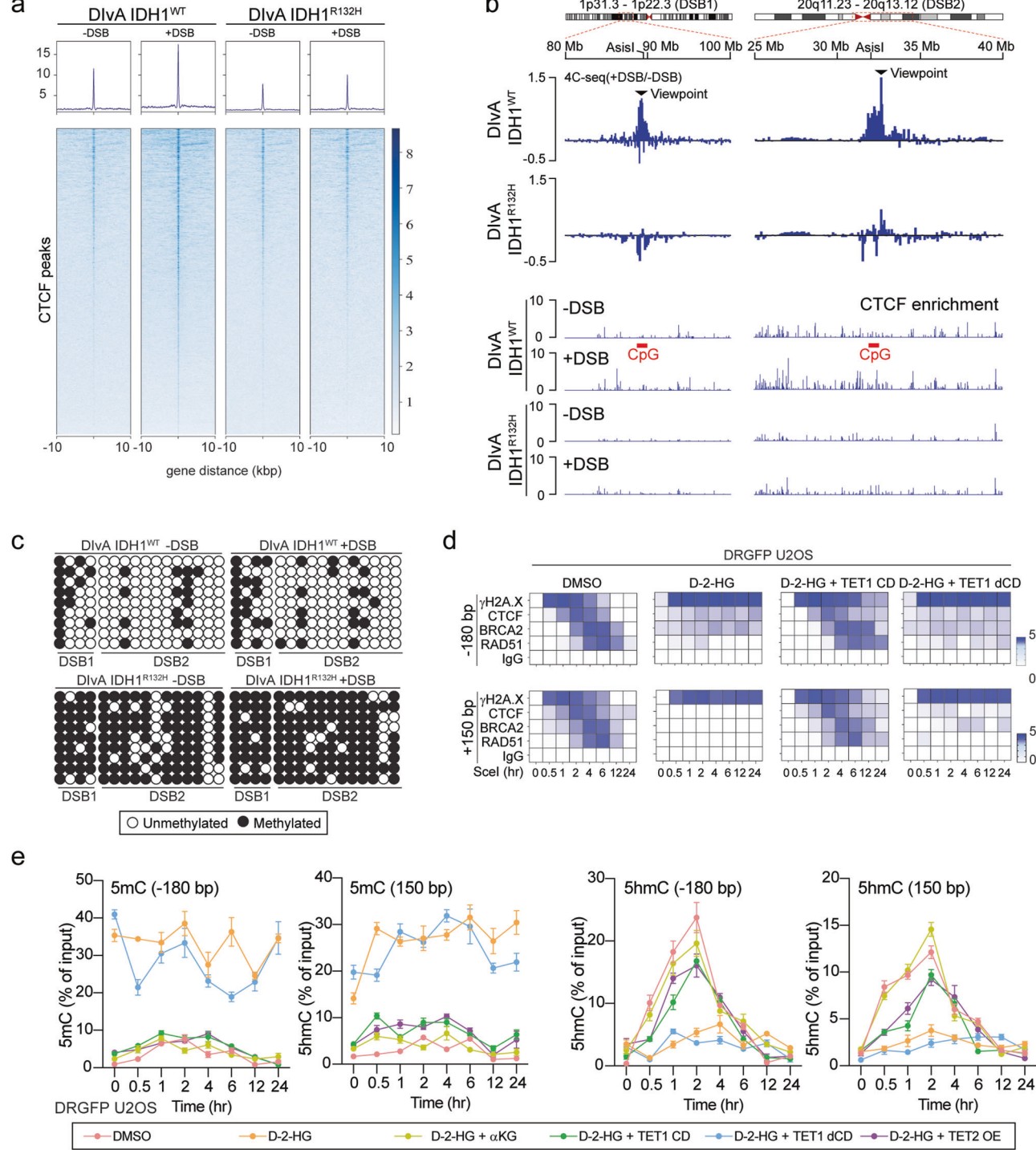

**Fig. 5 | IDH mutation disrupts chromatin structure around DSB through DNA hypermethylation. a** ChIP-seq assay shows alteration of CTCF coverage around AsiSI DSB sites in IDH1[WT] and IDH1[R132H] DIvA cells. Two biological replicates were performed. **b** Top, genomic tracks show differential 4C-seq signal (log2[+DSB/−DSB]) in IDH1[WT] and IDH1[R132H] DIvA cells treated with 4-OHT (300 nM) for 4 h. Bottom, CTCF enrichment at the same genomics loci. Two biological replicates were performed. **c** Bisulfite PCR shows the DNA methylation status of the CpG islands around AsiSI sites shown in (**b**). **d** ChIP-qPCR assay shows the CTCF, BRCA2, RAD51 binding, and γH2A.X modification around the DSB sites. DD-Sce-I-GR DRGFP

U2OS cells were treated with 0.5 μM Shield1 and 0.2 mM TA for 0.5, 1, 2, 4, 6, 12, or 24 h before collection for ChIP-qPCR. Two genomic regions (−180 bp and +150 bp) adjacent of DSB sites were monitored. Cells were transfected with TET1 CD/dCD. The recruitments were quantified relative to the IgG control. **e** MeDIP/hMeDIP assay shows epigenetic status around the Sce-I site. DD-Sce-I-GR DRGFP U2OS cells were treated with D-2-HG (0.5 mM), αKG (10 mM), or overexpression of TET CD/dCD, or TET2. DSB was induced with 0.5 μM Shield1 and 0.2 mM TA. Data are presented as mean ± SEM from three independent experiments. Source Data are provided as a Source Data file.

TMZ is known to initially introduce an $O^6$-methyl adduct, which can lead to DSBs following replication. Our group and others have observed an increase in γH2A.X modification within 24 h[36–40]. The rapid enhancement of the γH2A.X signal suggests the potential involvement of non-canonical signaling pathways, such as replication stress-associated DSBs, which contribute to DNA damage beyond the classical mechanisms.

## Metabolic and epigenetic predisposition on "BRCAness"

"BRCAness" is a distinctive cancer phenotype, highlighted by sensitivity to DNA repair inhibitors. "BRCAness" is commonly established by genetic predispositions such as loss-of-function changes in BRCA1/2 gene. The genetically defined losses of DNA repair elements predispose inhibition of the remnant DNA repair pathways such as PARP-associated base excision repair. IDH-mutated cancers exhibit "BRCAness" and favorable disease outcomes under genotoxic therapies. Several pioneering studies suggested that IDH mutation leads to BRCAness by affecting histone methylation and the recruitment of DNA repair proteins[9,10]. Our present study suggests a parallel mechanism that is mediated by cytosine methylation. We demonstrated that TET1/2-mediated cytosine demethylation is essential for the adjustment of chromatin conformation status (Fig. 5) and the recruitment of DNA repair protein (Fig. 3). Overall, our findings reveal the molecular mechanisms of TET1/2-dependent effects on "BRCAness" in IDH mutant malignancy. Targeting the distinctive vulnerability of DNA repair pathways could be valuable to improve the therapy outcome. Additional patient-derived models and in vivo studies are warranted to further elucidate the relationship between IDH-mutant cancers and their sensitivity to DNA repair inhibitors.

# Methods

## Cell lines and reagents

The U251 MG cell line was obtained from Sigma Aldrich. The U2OS DRGFP cell line was purchased from the ATCC. Cells were cultured in DMEM-Ham F-12 medium (DMEM/F-12, 1:1; Life Technologies) supplemented with the 10% FBS (Cellgro) and 1% antibiotics (100 U/mL penicillin and 10 mg/mL streptomycin) at 37 °C in humidified air with 5% $CO_2$. Patient-derived GSCs, BT054 and BT142, were maintained in Neurobasal-A media (Gibco) supplemented with B27, N2, EGF (20 ng/mL), and bFGF (20 ng/mL). TMZ was purchased from Sigma. DMSO was purchased from Fisher Scientific. Octyl-(R)-2HG and Octyl-α-KG were purchased from Sigma (Sigma, SML2200, SML2205). Shield1 was purchased from Takara (Takara, 632189). Triamcinolone acetonide (TA) was purchased from Sigma (T6501-250MG). 4-Hydroxytamoxifen (4-OHT) was purchased from Sigma (Sigma, H7904). The TET1 catalytical domain (TET1 CD) and catalytically dead construct (TET1 dCD) were kindly provided by Dr. Anjana Rao.

## HR/NHEJ reporter assay

The fragment of Shield1 ligand-dependent destabilization domain (DD), Sce-I and glucocorticoid receptor (GR) was cloned to the pLenti-C-Myc-DDK-IRES-BSD lentiviral vector (Origene) with MluI and XhoI restriction sites, for the expression of fusion protein DD-Sce-I-GR. The vector is transduced into the DRGFP U2OS cell line. The DD-Sce-I-GR cassette allows constitutive expression of DD-Sce-I-GR, whereas the fusion protein is rapidly degraded in the cells and shows no DSB induction in normal cell culture conditions. Ligand-based Sce-I induction allows monitoring of the longitudinal kinetics of DNA damage repair protein recruitment to the DSB site. Five hundred micromolar Shield1 and 100 nM TA were used to stabilize DD-Sce-I-GR, which allows immediate stabilization and nuclear translocation of DD-Sce-I-GR. The induction of HR efficiency was measured by flow cytometry.

For ChIP validation with the DRGFP reporter system, cells were treated with Shield1 (500 μM) and TA (100 nM) and fixed in 1% formaldehyde at different time points. IP was performed with indicated antibodies. Input DNA was used as the control to calculate the enrichment of the DNA pull-down. Primers for ChIP experiments are included in Supplementary Table 1.

EJ5 and EJ7-GFP reporter cells were obtained from Dr. Jeremy Stark's laboratory[41,42]. The EJ5 cells measure total NHEJ activity, including both canonical (c-NHEJ) and alternative (alt-NHEJ) pathways, by quantifying the repair of Sce-I-induced DSB. We established a stable cell line similar to the process for DRGFP cell line. EJ5 cells were infected with the lentivirus expressing DD-Sce-I-GR. Cells were treated with Shield1 (500 μM) and TA (100 nM) for DSB induction. The NHEJ efficiency was measured by flow cytometry. For EJ7-GFP reporter system, the DSB is induced by transfection of 7a sgRNA (Addgene #113620) and 7b (Addgene #113624) plasmids.

## DSB inducible via AsiSI cell line (DIvA) with IDH1 mutation

DIvA cells were obtained from Dr. Gaelle Legube's laboratory[43]. The AsiSI restriction enzyme is fused to an estrogen receptor hormone-binding domain, which is activated and translocated into the nucleus with the presence of 4-OHT (300 nM). The AsiSI restriction enzyme specifically cleaves DNA at 5'-GCGAT^CGC-3' sequence within the genome. IDH wild-type or IDH R132H mutant DIvA cells were generated with lentivirus infection as previously described[7]. For DSB induction, 4-OHT (300 nM) was added to the culture medium for 4 h.

## Western blotting

Cells were lysed in RIPA lysis buffer supplemented with protease and phosphatase inhibitor cocktail (Thermo Fisher) on ice. Cell lysates were centrifuged at $12,000 \times g$ for 10 min. Protein concentration was determined using a BCA protein assay (Thermo Scientific). A total of 30 μg protein extracts were separated on NuPAGE 4%–12% Bis-Tris gels (Life Technologies) and transferred to Immobilon-P polyvinylidene fluoride membranes (Millipore). The membranes were probed with primary antibodies and visualized by chemiluminescence according to the manufacturer's protocol (Thermo Fisher). The primary antibodies used include anti-γH2A.X (Millipore, 05-636, 1:1000), H2A.X (Cell Signaling Technology, 2595, 1:1000), RAD51 (Cell Signaling Technology, ab88572, 1:1000), BRCA2 (Bethyl Laboratories, 1:1000), CTCF (Abcam, ab70303, 1:1000), ATM S1981 (Cell Signaling Technology, 5883S, 1:1000), ATM (Cell Signaling Technology, 92356S, 1:1000), DDK (Origene, TA50011-1, 1:2000), TET1 (GeneTex, GTX124207, 1:1000), TET2 (Thermo Fisher, PA5-78514, 1:1000), IDH1-R132H (Millipore, MACB1103, 1:1000), H3 (Cell Signaling Technology, 9715, 1:2000), CtIP (Active Motif, 61942, 1:000), Mre11 (GeneTex, GTX70212, 1:1000), and GAPDH (Cell Signaling Technology, 2118 L, 1:2000).

## Immunoprecipitation

Immunoprecipitation was performed with Pierce Crosslink Magnetic IP/Co-IP Kit (Thermo Fisher, 88805). Dynabeads Protein G (Thermo Fisher, 10003D) were incubated with 2 μg of antibodies for 15 min at room temperature. The cell lysates were incubated with Dynabeads Protein G-coupled antibody overnight at room temperature for 2 h with rotation. An aliquot was incubated with IgG as a control to determine background binding.

## Immunofluorescence

Cells were fixed with 4% paraformaldehyde at 4 °C for 60 min. The cells were then permeabilized with 0.2% Triton X-100 in PBS for 10 min at room temperature and incubated with 5% BSA for 1 h at room temperature. The samples were incubated with primary antibodies at 4 °C overnight and followed with Alexa Fluor 594 goat anti-mouse IgG (Thermo Fisher, A-11012) and Alexa Fluor 488 goat anti-rabbit IgG antibodies (Thermo Fisher, A-11034). The primary antibodies including CTCF (Abcam, ab70303), γH2A.X (Millipore, 05-636), RAD51 (Abcam, ab88572), BRCA2 (Bethyl Laboratories, A303-434A), 5hmC (Active

motif, 39069), TET1 (GeneTex, GTX124207), TET2 (Thermo Fisher, PA5-78514), RPA70/RPA1 (Cell Signaling Technology, 2198), or 53BP1 (Novus Biological, NB100-304) were used at 1:1000 dilution. Nuclei were stained with Hoechst 33342 for 3 min. Images were captured with a Zeiss LSM 710 Confocal Microscope. In total, 40–60 cells among several biological replicates were used to obtain statistically meaningful data.

## Proximity ligation assay (PLA)

PLA assay was performed according to the instructions of the Duolink Proximity Ligation Assay kit (Sigma). Cells were fixed with 4% paraformaldehyde at 4 °C for 60 min and permeabilized with 0.2% Triton X-100 in PBS for 10 min at room temperature, incubated with blocking solution in a heated humidity chamber for 60 min at 37 °C. The samples were incubated with primary antibodies at 4 °C overnight. PLA minus and PLA plus probes were added followed by incubation for 1 h at 37 °C. Ligase was used to join the two hybridized oligonucleotides into a closed circle. The DNA was then amplified with rolling circle amplification. Images were captured with a Zeiss LSM 710 Confocal Microscope. The primary antibodies including CTCF (Abcam, ab70303), RAD51 (Abcam, ab88572), and BRCA2 (Bethyl Laboratories, A303-434A) were used at 1:1000 dilution.

## Laser micro-irradiation

The cells were incubated with Hoechst 33342 (10 μg/mL) for 10 min before irradiation. Zeiss LSM 780 confocal microscope was used for the laser micro-irradiation through a 63× objective lens. The cells were incubated with DMEM/F-12 in glass-bottom dishes placed in chambers to prevent evaporation on a 37 °C hotplate. The cells were scanned 20 iterations with the 405 nm laser at 30% power. Cells were recovered for 10 min before fixation and immunostaining.

## Dot blotting

Genomic DNA was denatured in 0.1 M NaOH at 95 °C for 10 min. DNA was then neutralized with 1 M $NH_4OAc$ on ice and proceeded with twofold serial dilution. Denatured DNA was resuspended in 100 μL Tris-HCl and spotted on nitrocellulose membranes within the Bio-dot microfiltration apparatus (Bio-Rad). The samples were pulled through by applying a gentle vacuum. The UV crosslink was performed in Stratalinker UV crosslinker (Stratagene) at an energy of 120,000 μJ/$cm^2$. The membrane was blocked with superblock supplemented with 0.1% Tween 20 for 1 h, followed by the incubation with the anti-5hmC antibody (Sigma, MABE176, dilution, 1:1000) overnight at 4 °C and HRP-conjugated anti-rabbit IgG secondary antibody (1:1000) for 1 h at room temperature. After washing three times, the membrane was treated with chemiluminescence reagents (Bio-Rad) and scanned by the ChemiDoc imaging system from Bio-Rad.

## MeDIP and hMeDIP assay

MeDIP or hMeDIP assay was performed according to the instructions of the MeDIP Kit (Active Motif, 55009) and hMeDIP Kit (Active Motif, 55010), respectively. Genomic DNA (500 ng) was fragmented with Covaris S220 sonicator and denatured at 95 °C for 10 min. The single-strand DNA was incubated with 5mC or 5hmC antibody overnight at 4 °C. After an overnight incubation, the antibody/DNA complex is captured using protein G magnetic beads. Subsequently, the enriched DNA is eluted from the beads. Eluted DNA was column purified with Active Motif's Chromatin IP DNA Purification Kit (Catalog No. 58002) and tested using real-time PCR.

## Comet assay

The comet assay was performed as previously described[44]. Cells were resuspended in 0.75% low melting point agarose and placed on glass slides (Trevigen). Cells were lysed in 1.2 M NaCl, 100 mM EDTA, 0.1% sodium lauryl sarcosinate, and 0.26 M NaOH (pH > 13). Electrophoresis

of the cell spreads was performed at 21 V for 30 min within pre-cooled electrophoresis buffer (200 mM NaOH, 1 mM EDTA, pH > 13) in a CometAssay ES unit. DNA was labeled with SYBR Safe DNA Gel stain and visualized under a Zeiss LSM 710 confocal microscope.

## ChIP-seq

ChIP-seq assay was performed using ChIP-IT High Sensitivity (Active Motif, 53040) according to the manufacturer's instructions. Sonication was performed with Covaris S220 at a power of 140 W for 20 min. Control IgG, the antibodies targeting CTCF (Abcam, ab70303), γH2A.X (Millipore, 05-636), RAD51 (Abcam, ab88572), BRCA2 (Bethyl Laboratories, A303-434A), were used for pulling down the target DNA. Library construction was performed with NEBNext® Ultra™ II DNA Library Prep Kit for Illumina. Pooled libraries were subjected to sequencing on the Illumina NextSeq2000 platform.

## Isolation of chromatin-bound proteins

The isolation of chromatin-bound proteins was performed as previously described[45]. One million cells were collected using a cell scraper and washed with 1 mL ice-cold PBS. The cells were lysed in 100 μL ice-cold E1 buffer (50 mM HEPES-KOH, 140 mM NaCl, 1 mM EDTA, 10% glycerol, 0.5% NP-40, 0.25% Triton X-100, and 1 mM DTT). The pellet was resuspended in E2 buffer (10 mM Tris-HCl, pH 8.0, 200 mM NaCl, 1 mM EDTA, and 0.5 mM EGTA) and centrifuged. The resulting pellet was resuspended in 100 μL E3 buffer (500 mM Tris-HCl, pH 6.8, and 500 mM NaCl) and sonicated in a water bath sonicator on ice for 5 min, with 30 s ON and 30 s OFF cycle at maximum power to shear genomic DNA. The samples were then centrifuged at 16,000 × g at 4 °C for 10 min. The supernatant was collected and used for subsequent western blot analysis.

## Quantifying end resection with qPCR

The single-stranded DNA generated near two AsiSI-induced DSBs in the genome ("Chr 1, site 1" at Chr1:89231183 and "Chr 22, site 2" at Chr22:38864102) was analyzed using a previously described procedure[46]. Genomic DNA was digested with *Bsr*GI or *Ban*I enzyme (New England Biolabs) or neither (mock digested). Samples were heat inactivated at 65 °C and analyzed by qPCR. ΔCt was calculated by subtracting the Ct value of the mock-digested sample from the Ct value of the digested sample. The percentage of ssDNA generated by resection at selected sites (ssDNA %) was calculated with the following equation: ssDNA % = $1/(2^{(\Delta Ct-1)} + 0.5) \times 100$.

## Flow cytometry

Cells were collected and washed with PBS. Then cells were filtered through a nylon mesh cell strainer snap cap before detection with BD Accuri C6 flow cytometer or FACSCanto II. For longer preservation of samples, the cells were kept in 1% paraformaldehyde solutions. GFP signals were monitored with 488 nm laser, 530/30 emission nm bandpass filter. The plottings were presented with a Contour Plot type.

## Circular chromosome conformation capture sequencing (4C-seq)

The 4C-seq experiments were performed as described[47] with minor modifications. A total of 15 million cells were cross-linked with 2% formaldehyde for 10 min at room temperature, lysed, and digested with DpnI (100 U, NEB). Digested DNA was then ligated with 50 U T4 DNA ligase (Promega), purified, and digested with NlaIII overnight (NEB). After a second ligation, DNA was purified before proceeding to library preparation. For 4C-seq library preparation, 800 ng of 4 C DNA were amplified using two-step PCR reactions with indexed primers and Illumina adapters. Libraries were purified with the QIAquick PCR Purification Kit (Qiagen), pooled, and subjected to 150-bp single-end sequencing on the Illumina NextSeq2000 equipment. For 4C-seq data analysis, all sequencing data underwent FastQC analysis. Sequence

mapping was done using Pipe4C (R package, R/4.2 version) and peaks were called with PeakC (R package, R/4.2 version). The 4C-seq tracks were generated through deepTools (3.5.5 version) and visualized with Integrative Genomics Viewer (IGV, 2.18.4 version).

## RNA interference

Cells were seeded at $2 \times 10^5$ in a 6-well plate and allowed to grow for 24 h. One hundred pico molar siRNAs (Integrated DNA technology) were transfected to the cells with 7.5 μL Lipofectamine RNAiMax (Thermo Fisher). The transfection was continued for 72 h before subsequent assays and tests. The sequence of the small interference RNA used in this study are: siCTCF1.1.F: CAA CAG CUA UCA UUC AGG UUG AAG A; siCTCF1.1.R: CGG UUG UCG AUA GUA AGU UCCA ACU UCU; siCTCF1.2.F: ACG CCA ACA UGA GAC CUG UAA UAA A; siCTCF1.2.R: CCU GCG GUU GUA CUC UGG ACA UUA UUU. All oligos were synthesized at Integrated DNA Technology.

## CRISPR-cas9 mediated-knockout

Two single guide RNAs (sgRNAs) of TET1 (TET1-1, TET1-2) and TET2 (TET2-1, TET2-2) were ligated to pSpCas9(BB)-2A-Puro (PX459) V2.0 vector (Addgene #62988). The plasmids were transfected to cells and the cells were selected with 1 μg/mL puromycin for 4 days. The gene KO efficiency was validated through western blotting.

## Statistics and reproducibility

Statistical analysis was carried out with GraphPad Prism software (RRID:SCR_002798, 16 v8.4.3). All blotting experiments were repeated at least twice. A one-way ANOVA was applied for statistical comparisons unless otherwise indicated, in which case a Student's $t$-test was used for comparisons between two groups. All statistical analyses were conducted with two tails. The results are presented as mean ± SEM. * represents $p$ value < 0.05 and ** represents $p$ value < 0.01.

## Reporting summary

Further information on research design is available in the Nature Portfolio Reporting Summary linked to this article.

## Data availability

All raw and processed ChIP-seq and 4C data generated in this study have been deposited in NCBI's Gene Expression Omnibus (GEO) database under GEO series accession code GSE283196. Additional data supporting the findings of this study are available within the paper and its Supplementary Information files. Source data are provided with this paper.

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

## Acknowledgements

This research was supported by the Intramural Research Program of the NIH, NCI.

## Author contributions

F.L. and C.Y. designed the study. F.L., K.K., H.F., and C.Y. conducted the experiment and collected the data. F.L., K.K., H.F., J.Z., D.L.R., M.A., and C.Y. performed the statistical analysis. F.L. and C.Y. prepared the manuscript. F.L., K.K., J.Z., D.L.R., M.A., and C.Y. revised the manuscript. All authors read and approved the final manuscript.

## Funding

## Competing interests

The authors declare no competing interests.
