## [Transparent Peer Review file · Nature Communications]

D-2-hydroxyglutarate impairs DNA repair through epigenetic reprogramming

Corresponding Author: Dr Chunzhang Yang

Version 0:

Reviewer comments:

Reviewer #1

(Remarks to the Author)

In this manuscript, Lang et al seek to determine the role of D-2-HG in chromatin remodeling via CTCF and its inability to recruit BRCA2 and RAD51. In this way they propose that the TET1/2 deficient phenotype leading to hypermethylation in IDH mutant glioma assists in revoking CTCF from binding to the DNA – preventing the final complex formation. Ultimately, this group proposes that the oncometabolite D-2-HG induces an unstable genome with different TADs and chromatin structure via CTCF/BRCA2/RAD51 – in the context of genotoxic stress. While the idea is overall interesting, the models used are insufficient to know how generalizable this finding is. This study is also limited to a few DNA repair proteins, where it was unclear how these proteins were chosen over the many other players in DNA damage repair. There is also a lack of time course data in different models, which would further strengthen the findings of this paper. Overall, this paper seems to portray a logical next step rather than a large jump in the DNA repair field, in the context of IDH mutation and genotoxic stress.

Major Comments:

1. The goal of this study seems to be to link together known observations from previous groups to show that D-2-HG is an important metabolite that regulates TET1/2 function, CTCF affects genome architecture, and that D-2-HG affects the kinetics of CTCF/BRCA2/RAD51 complex formation.

2. While this work does show some new findings in the context of IDH mutant glioma, there is only one glioma model used – U251 – which is not a common or representative model of glioma, much less IDH mutant glioma. The other cell line used is U2OS, which while commonly used in DNA repair, is not a glioma cell line. Endogenous IDH mutations are known to function differently from overexpression models – especially in a model where long-term chromatin changes are what are being investigated. Therefore, this model must be shown in an endogenous IDH mutation cell line model for the results to be convincing.

3. Additional experiments, evidence, and models are needed. One of the main points of this manuscript is to connect IDH mutant glioma with chromatin remodeling via D-2-HG. However, throughout the manuscript an endogenous IDH mutant model is not used. As the theory behind D-2-HG is the ability to inhibit enzymes, this will create lasting changes vs an overexpression model in U251 cells, and an osteosarcoma line U2OS. At a minimum, one endogenous IDH mutant model needs to be used to show that this is a tractable finding. Also, how would the authors propose to test this in a more clinically relevant model, or patient samples?

4. Data analysis seems accurate, but quantification of multiple cells per model is needed. For all the DD-Scel-GR reporter cells and DivA assays, are these quantifications from one cell each? If so, more than one cell per condition needs to be quantified. A time course should also be completed with many of these assays as the methods state the cells are allowed to recover for 10 minutes – there may be a recovery of RAD51/BRCA2 recruitment if given more time. This is true for the ChIP PCR as well – these cells may just need longer to recruit BRCA2/Rad51 instead of stating these proteins are not recruited.

5. While common DNA repair assays are used, there is no connection to IDH mutation tumorigenicity. As this paper seeks to

better understand therapeutic vulnerabilities in IDH mutant GBM, some translational aspect should be tested to show that this pathway is important for tumor cell growth. Throughout the paper we only are shown DNA repair assays, but no cell growth, viability, or cell death assays. –

6. Overall, the methods are common and standard methods in the field with adequate details provided. The methods section has adequate details, but more information should be added to the figures to assist the reader in their ability to understand the paper quickly.

6. The decision to investigate BRCA2 and not BRCA1 was not clearly explained. As BRCA1 is the dominant protein in this pathway, why was it not also investigated?

7. Figure 3G is one of the most interesting of the paper. As the authors show that γ H2AX can be recruited, it seems that it is then not signaled to leave the chromatin. As many of the factors are known that regulate γ H2AX movement off/on the chromatin, are any of these proteins inhibited by D-2-HG? As this would be one explanation for the data represented. Furthermore, total γ H2AX should be included in the WBs as it would be good to know if total H2AX is being affected, or more H2AX is being phosphorylated and deposited onto the chromatin.

Minor comments:

1. D-2-HG is used throughout the study as a treatment. The baseline levels of D-2-HG should be measured in the IDHmut models – as well as showing the decrease with the inhibitor in this model.

2. The cell line model should be included in the figures panels. This would make the paper easier for the reader to understand as the main comparison is between IDH mut and IDH WT, clearly marking these would be helpful.

Reviewer #2

(Remarks to the Author)

In general, I found this manuscript to be intriguing, with a novel and interesting mechanism to account for the changes in DSB repair efficiency linked to IDH mutations. The evidence that blocking 2-HG production reverses the phenotype is persuasive, and the work is consistent with previous literature showing HR defects in cells expressing IDH mutants. The major weakness of the paper is in defining the defect in the DSB repair pathway. There are conflicting results on PARylation independent recruitment of CTCF to DNA damage sites. (Khalid Hilmi et al. *Sci. Adv.* 3, e1601898 (2017)) found that prolonged pre-treatment with PARP inhibitors did not prevent CTCF but did prevent BRCA2 and RAD51 from being efficiently recruited to a restriction enzyme-induced DSB. The 24 hour pre-treatment of the cells with PARP inhibitor complicates the interpretation of these results. However, the zinc fingers have been directly shown to bind PAR and are required to recruit CTCF to laser microirradiation sites (Han, D., Chen, Q., Shi, J. et al. CTCF participates in DNA damage response via poly(ADP-ribosyl)ation. *Sci Rep* 7, 43530 (2017)). Is the recruitment of BRCA2 and RAD51 still sensitive to PARP inhibition while CTCF remains recruited in this cell system?

Another study used IP-mass spectrometry to detect DNA repair proteins involved in HR that interacted with CTCF. They found MRE11 and CtIP, which are important for DNA end resection that commits cells to HR-mediated repair (*Nucleic Acids Research*, Volume 47, Issue 17, 26 September 2019, Pages 9160–9179). They also showed that PARylation of CTCF was needed for BRCA2 and RAD51 recruitment (Khalid Hilmi et al. *Sci. Adv.* 3, e1601898 (2017)). Additionally, MDC1 and Rad51 were reported to bind to CTCF (*PNAS* 114:10912-7). This creates a lot of uncertainty about where the problem in DSB repair happens. Regarding the defect in homologous recombination repair, the impact on DNA end resection (re: MRE11/CtIP interaction) should be examined. A problem in end resection would prevent all later steps and explain the loss of BRCA2 and RAD51 from DSB sites in the IDH mutant cells.

How should we understand the impact of DNA methylation status on CTCF binding? Does this mean that CTCF only binds to DNA sequences that are specific to CTCF? Regarding the *Isce1* cut site, can the variation in binding be attributed to methylation of certain DNA sequences? Are there any CTCF consensus sites close to the cut site that could account for the CTCF binding? The correlation is evident, but the exact mechanism is not clear. Similarly, is it only the native CTCF binding sites that are lost around the *ASIS1* cut sites or is there a more non-specific DNA binding that is sensitive to methylation status?

The data in Figure 5A raise a similar question about the origin and mechanism of the repair defect. The paper does not mention 53BP1, which is shown in the image series and seems to have a loading deficiency as well. This could indicate a problem upstream of the interactions that this study examines and should be further investigated. If both 53BP1 and BRCA1 recruitment are impaired, this would suggest that the main defect is upstream in the signaling pathway (MRE11, CtIP, MDC1 are all possible upstream interactions where CTCF could be involved based on its known direct binding to these upstream proteins) rather than at the level of BRCA2 and RAD51 loading. The limitation is that only single cells are shown and individual cells within the same population can vary a lot depending on cell cycle position. This uncertainty about 53BP1 recruitment also emphasizes the fact that only an HR reporter system was used. The authors should also evaluate the efficiency of NHEJ in IDH mutants.

What is the interpretation of a greater number of H2AX foci in the IDH mutants? How is more damage created with the same amount of DNA damage? Does this reflect differences in the kinetics of the repair or differences in the kinetics of assembly of H2AX foci? A kinetic analysis of H2AX foci appearance and disappearance would address this uncertainty.

Cell cycle analysis should be presented to clarify how the mutants and rescued cells compare in cell cycle distribution. Presumably, activation of checkpoints in the IDH mutant cells will alter cell cycle distribution to some extent. Cell cycle directly influences DNA repair pathway utilization and so this needs to be excluded as the basis for any observed

differences in repair properties.

The elevated steady-state DNA damage/H2AX foci in the absence of damage is important and should be investigated further. There could be a number of explanations. If the cells are undergoing greater levels of replication stress, as they report, they may both be enriched in S-phase cells, which also normally have the highest incidence of DSB foci and cells with multiple DSB foci, and have higher numbers of foci specifically in S-phase cells. Post-mitotic DNA damage results in the formation of unusually large DSB foci that are typically only one or a few per nucleus and would reflect differences in DSB repair or checkpoint function at different stages in the cell cycle to explain the observed differences.

The line scans in Figure 2e are unnecessary. The more important information present in the figure would be the percent of the TMZ-treated H2AX foci that are positive for RAD51. This should be quantified and added if the manuscript is resubmitted.

While the PLA assay supports the claim that there are direct interactions between CTCF and RAD51/BRCA2, the evidence in the literature for a direct interaction is limited. CTCF interacts with BRCA2 in a PARYlation-dependent and Zinc finger-dependent manner. As zinc fingers are common PAR binding motifs and this interaction requires both Parylation and CTCF zinc fingers, this could reflect the association of both BRCA2 and CTCF with PAR polymers released as large complexes upon cell lysis. The PLA assay does not distinguish between these possibilities because DNA damage sites are regions that are concentrated in all of these factors. All that it convincingly shows is that all three proteins are recruited to DNA damage sites. If a control showing that gammaH2AX or 53BP1 does not show PLA foci at double-strand breaks, it would be compelling that the method can discriminate between direct interactions and locally high concentrations. Optimally, with around 10-20 nm distances between antibodies, it's unclear that high local concentrations won't generate the same robust signal.

An additional validation step necessary to confirm that CTCF interactions are shared with those reported in the literature is to test the sensitivity of BRCA2 and RAD51 association with CTCF in the PLA assay. They are reported to be sensitive to PARP inhibition and mediated by PARYlation. The authors should consider

Figures 2 E,F; 3A-H; Because there is variation across the cell cycle with respect to DNA repair pathway utilization, it is not appropriate to show single cells to illustrate differences. Rather, an image showing cells representing different stages of the cell cycle is required to appropriately compare to samples.

Reviewer #3

(Remarks to the Author)

Reviewer #4

(Remarks to the Author)

please see uploaded file

Reviewer #5

(Remarks to the Author)

Version 1:

Reviewer comments:

Reviewer #1

(Remarks to the Author)

In this revised manuscript, the authors made commendable efforts to address all my comments with necessary new data as well text revisions. This is an improved study that further strengthened its overall significance. This manuscript is now sufficient for its further consideration at Nature Communications.

Reviewer #2

(Remarks to the Author)

I commend the authors for taking all of my criticisms seriously and improving the manuscript with the resultant data and controls. I have no outstanding issues with the revised document.

Reviewer #3

(Remarks to the Author)

Reviewer #4

(Remarks to the Author)

Lang et al., Nature Communications manuscript revised NCOMMS-24-13290-T, -2-Hydroxyglutarate establishes BRCAness through epigenetic reprogramming

The authors have undertaken a significant amount of additional work to experimentally address our comments. However, there still remain some concerns in response to the rebuttal which have been highlighted below. To not delay the publishing of the important new findings presented in this manuscript, it is recommended that the manuscript be accepted given that the following points are adequately addressed within the text and figures of the manuscript.

In response to our previous comment number 3 noting concerns over the lack of detail included with respect to the experimental design, statistics and materials and methods, the authors have updated some aspects of the paper to include more detail. However, there is still information missing and some of the information provided has raised additional concerns.

- 1) For the majority of western blots presented within the paper there is no indication of how many times these experiments were repeated or whether this was only completed one time.
- 2) The majority of figures list a student's t test as the primary statistical analysis used to determine significance between treatment groups. This analysis is not appropriate for experiments containing more than two groups. Alternatively, an ANOVA should be used to determine statistical significance and then the correct post-hoc analysis, which corrects for multiple comparisons, should be employed to determine which treatment groups are responsible for the detected differences. Within the main text this applies to figures 1a, 1f, 1g, 1k, 1m, 1n, 2c, 2d, 3b, 3e, 3f, 4e, 4f, 4g, 4h. Within the supplement this applies to 1d, 1h, 3a, 3b, 3e, 4c, 4h, 5b, 5d, 6b, 6c, 7c, 7d, 7e, 7f, 7h, 7q, 7r, 8d.
- 3) For the quantification of microscopy images, it is unclear what the quantification within figures represent. For example, figure 2c states forty cells among three biological replicates were obtained, but the figure looks to have ~ 6 data points. Why does the figure have 6 data points? How does that relate to the number of cells and independent experiments evaluated? There is similar confusion for figures 2d, 3e, 3f. This should be clarified within the paper to make it clear to the reader how the data was prepared and analyzed.

In response to our previous comment number 2 noting concerns over the clarity and interpretation of figure five, the authors have added some clarifying information. However, more information is need for the readers to be able to interpret the results.

- 1) For figure 5b, the scale for the genomic site (units included) and exact genomic coordinates of the genome position should be provided for these types of analyses. This also applies to supplemental figures 8e, f, and h. Additionally, the CTCF enrichment tracks show a CpG site but that would include one "CG" in the genome and does not seem accurate given that figure 5C suggests there are 4 CpGs for DSB1 and 12 for DSB2. It would be preferable to show the exact CpGs within the region or to rename this to CpG island, although it is unclear based on the information provided if these genomic regions meet the criteria to be defined as CpG islands.

Reviewer #5

(Remarks to the Author)

Revision NCOMMS-24-13290-T

Reviewer 1	
In this manuscript, Lang et al seek to determine the role of D-2-HG in chromatin remodeling via CTCF and its inability to recruit BRCA2 and RAD51. In this way they propose that the TET1/2 deficient phenotype leading to hypermethylation in IDH mutant glioma assists in revoking CTCF from binding to the DNA – preventing the final complex formation. Ultimately, this group proposes that the oncometabolite D-2-HG induces an unstable genome with different TADs and chromatin structure via CTCF/BRCA2/RAD51 – in the context of genotoxic stress. While the idea is overall interesting, the models used are insufficient to know how generalizable this finding is. This study is also limited to a few DNA repair proteins, where it was unclear how these proteins were chosen over the many other players in DNA damage repair. There is also a lack of time course data in different models, which would further strengthen the findings of this paper. Overall, this paper seems to portray a logical next step rather than a large jump in the DNA repair field, in the context of IDH mutation and genotoxic stress.	We appreciate the author providing the summarization of our work. We are happy to follow the comments from the reviewer and improve our study, with an emphasis on improving the cell models and time course study.
Major Comments: 1. The goal of this study seems to be to link together known observations from previous groups to show that D-2-HG is an important metabolite that regulates TET1/2 function, CTCF affects genome architecture, and that D-2-HG affects the kinetics of CTCF/BRCA2/RAD51 complex formation.	We appreciate the insight from the reviewer. Our study is enlightened by the seminal studies that describe the inhibitory role of D-2-HG on TET1/2 function¹. The reports by Flavahan et al², and also later by Rahme et al³, from the same group discussed the loss of CTCF binding in cell lines with D-2-HG and TET malfunction. We believe besides the well-characterized CTCF/insulator malfunction, D-2-HG is relevant to other cancer biology changes in IDH-mutated cancers, such as DNA repair pathways.
2. While this work does show some new findings in the context of IDH mutant glioma, there is only one glioma model used – U251 – which is not a common or representative model of glioma, much less IDH mutant glioma. The other cell line used is U2OS, which while commonly used in DNA repair, is not a glioma cell line. Endogenous IDH mutations are known to function differently from overexpression models – especially in a model where long-term chromatin changes are what are being investigated. Therefore, this model must be	We appreciate the comments from the reviewer. We used several well-accepted cell line tools, such as U251 IDH1^{R132H}, U2OS Sce-I/DRGFP U2OS, and Asisl-ER (DivA) U2OS IDH1^{R132H} cells to explore the changes in DNA repair pathway in the context of IDH mutation/D-2-HG. We agree that validating these findings in disease-relevant glioma cell line models with intrinsic IDH mutation will be important for the results to be convincing.

shown in an endogenous IDH mutation cell line model for the results to be convincing.

We selected patient-derived glioma stem cells (GSCs) BT054 and BT142 that carry intrinsic IDH1 mutation and validated our findings. We showed through a dose-response analysis that the TMZ resistance in IDH-mutated GSCs could be enhanced with α KG or TET1/2 overexpression, suggesting the cytosine methylation is suppressing the DNA repair pathway. The new results have been included in Extended Data Fig. 7. The changes have been made in the manuscript on page 17, lines 446 – 449.

Besides, we investigated the recruitment of RAD51 in these GSCs. Our findings indicate that RAD51 recruitment to TMZ-induced γ H2A.X puncta is enhanced with the presence of α KG or TET1/2 overexpression. We believe this result indicates that reversing the cytosine methylation leads to improved homologous recombination DNA repair, which aligns with our finding in the U2OS and U251 cells. The new results are included in Extended Data Fig. 7a to 7f. The changes have been made to the manuscript on page 16, lines 439 – 441.

Puncta analysis confirmed more colocalization of γ H2A.X with RAD51 in puncta with the presence of α KG or TET1/2 overexpression.

In addition, α KG or TET1/2 overexpression reduced an overall number of γ H2A.X puncta, suggesting a reduced DNA damage.

Moreover, we conducted Western blotting experiments to confirm that the TMZ-induced DNA damage in either BT054 or BT142 cells could be relieved with the presence of α KG or TET1/2 overexpression. The new data is included as Extended Data Fig. 7g and 7h. The changes have been made to the manuscript on page 16, line 439 – 441.

Finally, we performed laser micro irradiation experiments in BT054 cells. We confirmed that α KG or TET1/2 overexpression would enhance the recruitment of RAD51 to the DSB loci, evidenced with colocalization of γ H2A.X and RAD51 puncta. The new result has been included as Extended Data Fig. 7g and 7h. The changes have been made to the manuscript on page 16, lines 434 - 436.

	3. Additional experiments, evidence, and models are needed. One of the main points of this manuscript is to connect IDH mutant glioma with chromatin remodeling via D-2-HG. However, throughout the manuscript an endogenous IDH mutant model is not used. As The theory behind D-2-HG is the ability to inhibit enzymes, this will create lasting changes vs an overexpression model in U251 cells, and an osteosarcoma line U2OS. At a minimum, one endogenous IDH mutant model needs to be used to show that this is a tractable finding. Also, how would the authors propose to test this in a more clinically relevant model, or patient samples?	We appreciate the comments from the reviewer. As included in the responses to point 2, we will use disease-relevant GSC cells to validate our findings. Glioma stem cells are recently developed in vitro models that recapitulate the cancer biology and therapy resistance from their tumor origin, and hence could serve as a more clinically relevant model than the U251 and U2OS cell lines. We also included AG-120, a pharmacological IDH1 mutant inhibitor for the experiments using these cells, to evaluate the changes in DNA repair efficiency when neomorphic activity is suppressed. The example could be found in Extended Data Fig. 2a and 2b, lines 295 - 302.
4. Data analysis seems accurate, but quantification of multiple cells per model is needed. For all the DD-Scel-GR reporter cells and DivA assays, are these quantifications from one cell each? If so, more than one cell per condition needs to be quantified. A time course should also be completed with many of these assays as the methods state the cells are allowed to recover for 10 minutes – there may be a recovery of RAD51/BRCA2 recruitment if given more time. This is true for the ChIP PCR as well – these cells may just need longer to recruit BRCA2/Rad51 instead of stating these proteins are not recruited.	We appreciate the comments from the reviewer. For morphological data, we quantify at least 40 cells among several biological replicates to obtain statistically meaningful data. The reviewer raised a brilliant point that a time course experiment will help understand the dynamic changes during DNA repair. Following this guidance, we tracked the recruitment of RAD51 through time-lapse microscopy. DNA damage was established through laser micro-irradiation on RAD51-RFP-transfected U251 cells. Within one hour of observation, we found that with the presence of D-2-HG, the RAD51 recruitment to the DNA damage sites are significantly delayed.

Similarly, we performed a time course imaging on BT054 cells carrying intrinsic IDH1 mutation. We discovered that the RAD51 recruitment could be enhanced with the presence of α KG. These findings suggest that the recruitment of DNA repair proteins such as RAD51 is significantly reduced with the presence of IDH mutation or D-2-HG.

These new results are included as Extended Data Fig. 7j. to 7m. The modifications have been made to the manuscript on pages 16-17, lines 442 – 445.

5. While common DNA repair assays are used, there is no connection to IDH mutation tumorigenicity. As this paper seeks to better understand therapeutic vulnerabilities in IDH mutant GBM, some translational aspect should be tested to show that this pathway is important for tumor cell growth. Throughout the paper we only are shown DNA repair assays, but no cell growth, viability, or cell death assays. –

We appreciate the comments from the reviewer. As the reviewer pointed out correctly, our present study seeks to understand the therapy vulnerabilities in IDH-mutated cancers, with a focus on DNA damage and repair pathways. We agree that the measurements on phenotypic outcomes, such as cell proliferation, viability, or cell death assays, would provide important information to highlight the importance of the molecular mechanism discussed in this study.

We have previously measured the impact of IDH mutation on cell growth and sensitivity to temozolomide-induced DNA damage. IDH mutations lead to a slight decrease in cell proliferation and enhanced sensitivity to DNA damage. The data is replotted from Lu et al., 2017⁴.

We further conducted apoptosis analysis and colony formation in the patient-derived glioma stem cells with endogenous IDH mutation.

Further, we conducted colony formation assays using BT054 and BT142 cells. We discovered that TMZ-induced growth suppression could be partially reversed with α KG or TET1/2 overexpression. The new results have been included as Extended Data Fig. 7s, 7t, and 7u. The changes have been made to the manuscript on page 17, lines 446 – 449.

Finally, we evaluated the cellular apoptosis in BT054 and BT142 cells. We found that TMZ-induced apoptosis in these cells could be reversed with α KG or TET1/2 overexpression, which aligns with our finding in Extended Data Fig. 7n and 7o. The new findings have been included in Extended Data Fig. 7p to 7r. The changes have been made to the manuscript on page 17, lines 447 – 449.

6. Overall, the methods are common and standard methods in the field with adequate details provided. The methods section has adequate details, but more information should be added to the figures to assist the reader in their ability to understand the paper quickly.	We appreciate this comment. We have included more details in the methods to ensure clarity and reproducibility. The changes are updated in the method section from page 5 to 11.
6. The decision to investigate BRCA2 and not BRCA1 was not clearly explained. As BRCA1 is the dominant protein in this pathway, why was it not also investigated?	We appreciate the comment from the reviewer. While BRCA1 and BRCA2 are both involved in DNA repair pathways, BRCA1 is a pleiotropic DNA repair protein that is involved in both checkpoint activation, replication fork stabilization, and DNA repair, whereas BRCA2 is a dedicated mediator of the core mechanism of homologous recombination DNA repair. Several previous studies indicate the physical interaction among CTCF, BRCA2, and RAD51^{5,6}, which we found more relevant in our present study with a focus on the step of homologous recombination.
7. Figure 3G is one of the most interesting of the paper. As the authors show that γH2AX can be recruited, it seems that it is then not signaled to leave the chromatin. As many of the factors are known that regulate γH2AX movement off/on the chromatin, are any of these proteins inhibited by D-2-HG? As this would be one explanation for the data represented. Furthermore, total γH2AX should be included in the WBs as it would be good to know if total H2AX is being affected, or more H2AX is being phosphorylated and deposited onto the chromatin.	We appreciate the comment from the reviewer. The reviewer raised an interesting point regarding the other factors that regulate γH2A.X chromatin integration. γH2A.X, the phosphorylated form of H2A.X at Serine 139, is managed by DNA repair-related kinases such as ATM, ATR, and DNA-PK, as well as protein phosphatases. To better understand the dynamics between H2A.X phosphorylation, we evaluated the phosphorylation of H2A.X kinases ATM, ATR, and DNA-PK and discovered that D-2-HG resulted in elevated phosphorylation in these kinases, as well as elevated γH2A.X signal. The addition of kinase inhibitors reverses the phosphorylation and γH2A.X signal. We believe this result indicated that D-2-HG results in obstacles in the DNA repair pathway, which leads to the prolonged activation of DNA damage related kinases, and γH2A.X signal.

We also tested the phosphatase activity through *in vitro* experiments in the presence of D-2-HG. Our finding shows that D-2-HG does not obviously change phosphatase activity.

Overall, these evidence indicate that D-2-HG does not exhibit an obvious inhibitory effect on the phosphatases that modify H2A.X, whereas the D-2-HG is more likely to act through DNA repair machineries on the chromatin.

Minor comments:

1. D-2-HG is used throughout the study as a treatment. The baseline levels of D-2-HG should be measured in the IDHmut models – as well as showing the decrease with the inhibitor in this model.

We appreciate the comment. We have measured the levels of D-2-HG in our DlvA cell line models. We confirmed that the D-2-HG level is substantially elevated in our cell line models. The levels of U251 IDH1^{R132H} have been reported in our previous work⁴.

2. The cell line model should be included in the figures panels. This would make the paper easier for the reader to understand as the main comparison is between IDH mut and IDH WT, clearly marking these would be helpful.

We appreciate the comment from the reviewer. We have included the cell line model information in the figure panels.

Reviewer 2

In general, I found this manuscript to be intriguing, with a novel and interesting mechanism to account for the changes in DSB repair efficiency linked to IDH mutations. The evidence that blocking 2-HG production reverses the phenotype is persuasive, and the work is consistent with previous literature showing HR defects in cells expressing IDH mutants.

The major weakness of the paper is in defining the defect in the DSB repair pathway. There are conflicting results on PARylation independent recruitment of CTCF to DNA damage sites. (Khalid Hilmi et al. *Sci. Adv.*3,e1601898(2017)) found that prolonged pre-treatment with PARP inhibitors did not prevent CTCF but did prevent BRCA2 and RAD52 from being efficiently recruited to a restriction enzyme-induced DSB. The 24 hour pre-treatment of the cells with PARP inhibitor complicates the interpretation of these results. However, the zinc fingers have been directly shown to bind PAR and are required to recruit CTCF to laser microirradiation sites (Han, D., Chen, Q., Shi, J. et al. CTCF participates in DNA damage response via poly(ADP-ribose)ylation. *Sci Rep* 7, 43530 (2017).). Is the recruitment of BRCA2 and RAD51 still sensitive to PARP inhibition while CTCF remains recruited in this cell system?

We appreciate the summary and the comment from the reviewer.

We appreciate the comment. We postulate that, in general, appropriate epigenetic status at the DSB site is necessary for efficient DNA repair. CTCF recruitment to the DNA damage site may be facilitated by local chromatin signaling such as PARylation.

To better understand the role of PARylation in our model system, we pre-incubated the cells with PARP inhibitor Olaparib (Ola) and discovered that the recruitment of BRCA2 and RAD51, but not CTCF, was compromised upon DNA damage induction.

Besides, we also conducted PLA assay to evaluate the physical interaction among CTCF, BRCA2, and RAD51. We do notice that PARP inhibition leads to suppressed CTCF/BRCA2 and CTCF/RAD51 interaction, suggesting that the CTCF-guided recruitment of BRCA2/RAD51 require PARylation.

Based on these findings, it seems in our model that CTCF recruitment to the DSB site is less affected by PARylation, whereas the subsequent recruitment of BRCA2/RAD51 is affected by PARylation level.

Another study used IP-mass spectrometry to detect DNA repair proteins involved in HR that interacted with CTCF. They found MRE11 and CtIP, which are important for DNA end resection that commits cells to HR-mediated repair (Nucleic Acids Research, Volume 47, Issue 17, 26 September 2019, Pages 9160–9179). They also showed that PARylation of CTCF was needed for BRCA2 and RAD51 recruitment (Khalid Hilmi et al. Sci. Adv. 3, e1601898 (2017)). Additionally, MDC1 and Rad51 were reported to bind to CTCF (PNAS 114:10912-7). This creates a lot of uncertainty about where the problem in DSB repair happens. Regarding the defect in homologous recombination repair, the impact on DNA end resection (re: MRE11/CtIP interaction) should be examined. A problem in end resection would prevent all later steps and explain the loss of BRCA2 and RAD51 from DSB sites in the IDH mutant cells.

We appreciate the comment from the reviewer. We agree that examining the efficacy of DNA end resection by Mre11/CtIP could be important to understand the impact of D-2-HG on the DNA damage and repair process. First, we conducted co-immunoprecipitation experiments to understand the interaction between Mre11 and CtIP in the context of IDH mutation. Our finding suggests that TMZ exposure resulted in enhanced physical interaction between Mre11 and CtIP, whereas this interaction could be compromised with the presence of D-2-HG. Based on this data, we postulated that D-2-HG impacts the DNA repair pathway through several different levels, ranging from the assembly of end resection complex, DNA topology changes, and HR repair.

CTCF has been previously shown to play a critical role in DNA end resection (Nucleic Acids Research, Hwang et al., 2019). Specifically, CTCF facilitates the recruitment of CtIP to double-strand break

(DSB) sites and interacts with Mre11 through its N-terminal domain. CTCF retention at DSB sites relies on the zinc finger domain, which binds directly to DNA. Therefore, DNA hypermethylation that disrupts CTCF binding can impair the end resection process, potentially serving as an upstream event that compromises homologous recombination repair.

The new data has been included as Extended Data Fig. 6f to 6g. The changes have been made to the manuscript on page 16, lines 430 – 433.

Further, we measured the end resection efficiency in D1vA cells through a previously reported method⁷. We observed compromised end resection efficiency in both DSB sites investigated, which aligns with the finding above.

The new data has been included as Extended Data Fig. 6d to 6e. The changes have been made to the manuscript on page 16, lines 428 – 429.

How should we understand the impact of DNA methylation status on CTCF binding? Does this mean that CTCF only binds to DNA sequences that are specific to CTCF? Regarding the *Isce1* cut site, can the variation in binding be attributed to methylation of certain DNA sequences? Are there any CTCF consensus sites close to the cut site that could account for the CTCF binding? The correlation is evident, but the exact mechanism is not clear. Similarly, is it only the native CTCF binding sites that are lost around the *ASIS1* cut sites or is there a more non-specific DNA binding that is sensitive to methylation status?

We appreciate the comment. Several pioneering studies indicate that CTCF binding is affected by the methylation status in the genome, which is largely through the CTCF-binding site². CpG islands in or next to the CTCF binding site could affect CTCF affinity based on their methylation status. In the present study, we believe that D-2-HG may impede the binding of CTCF by affecting the CpG island methylation status. We confirmed the substantially elevated CpG island methylation status next to the CTCF peaks at the *Asis 1* DSB sites. The result can be found in Fig. 5C and Extended Data Fig. 8i.

Further, we performed a Motif analysis for the CTCF peaks obtained from DivA cells that are next to the *Asis 1* DSB sites, to explore the consensus sequence of the binding sites. It seems most of the CTCF binding aligns with the CTCF-binding motif.

Homer de novo Motif Results (Output_Lostpeaks_treated//)

Known Motif Enrichment Results
Gene Ontology Enrichment Results

If Homer is having trouble matching a motif to a known motif, try copy/pasting the matrix file into STAMP
 More information on motif finding results: HOMER | Description of Results | Tips
 Total target sequences = 4219
 Total background sequences = 4558
 * = possible false positive

Rank	Motif	P-value	log P-value	% of Targets	% of Background	STD(B) STD(D)	Best Match/Details	Motif File
1	CCACTAGGGGGC	1e-779	-1.794e+03	34.56%	5.00%	38.2bp (57.6bp)	J0RUS(Z)K562-CTCF-ChIP-Seq(GSE32465)/Homer(0.927) More Information Similar Motifs Found	motif file (matrix)
2	CAAGCCAC	1e-116	-2.692e+02	19.58%	8.27%	47.9bp (58.6bp)	CTCF/MA1102.2/Jaspar(0.747) More Information Similar Motifs Found	motif file (matrix)
3	AAACCAGCCACC	1e-27	-6.369e+01	0.36%	0.00%	42.5bp (0.0bp)	G12(Z)GM2-G12-ChIP-Seq(GSE112702)/Homer(0.625) More Information Similar Motifs Found	motif file (matrix)
4	ATGGGCCAGAA	1e-23	-5.358e+01	0.31%	0.00%	45.8bp (0.0bp)	Pfaff1/MA1615.1/Jaspar(0.658) More Information Similar Motifs Found	motif file (matrix)
5	CTACCCATT	1e-21	-4.863e+01	0.28%	0.00%	53.0bp (0.0bp)	PP0128.1_Gcm1.2/Jaspar(0.679) More Information Similar Motifs Found	motif file (matrix)

The data in Figure 5A raise a similar question about the origin and mechanism of the repair defect. The paper does not mention 53BP1, which is shown in the image series and seems to have a loading deficiency as well. This could indicate a problem upstream of the interactions that this study examines and should be further investigated. If both 53BP1 and BRCA1 recruitment are impaired,

We appreciate the comment. We monitored 53BP1 levels to monitor the DNA damage response. Our preliminary finding indicates that its concomitant NHEJ pathway is unchanged upon the exposure of D-2-HG. The NHEJ efficiency has been measured in the context of IDH mutants or D-2-HG previously⁸. Sulkowski et al. discovered that the NHEJ efficacy is minimally changed in two cell lines with IDH

this would suggest that the main defect is upstream in the signaling pathway (MRE11, CtIP, MDC1 are all possible upstream interactions where CTCF could be involved based on its known direct binding to these upstream proteins) rather than at the level of BRCA2 and RAD51 loading. The limitation is that only single cells are shown and individual cells within the same population can vary a lot depending on cell cycle position. This uncertainty about 53BP1 recruitment also emphasizes the fact that only an HR reporter system was used. The authors should also evaluate the efficiency of NHEJ in IDH mutants.

mutation, whereas a significant loss of HR activity was seen in the same models.

As the reviewer instructed, we measured the NHEJ efficiency through EJ5 and EJ7 reporter systems in our laboratory. Both of the reporter systems indicate that NHEJ is minimally affected by D-2-HG.

The new results are included as Extended Data Fig. 1c and 1d. The changes have been made to the manuscript on page 12, lines 292 – 295.

What is the interpretation of a greater number of H2AX foci in the IDH mutants? How is more damage created with the same amount of DNA damage? Does this reflect differences in the kinetics of the repair or differences in the kinetics of assembly of H2AX foci? A kinetic analysis of H2AX foci appearance and disappearance would address this uncertainty.

We appreciate the comment. We performed a time course IF staining to understand the kinetics of γ H2A.X foci in DivA cells upon 4-OHT-induced DNA damage. We discovered that in both IDH1^{WT} and IDH1^{R132H} cells, γ H2A.X foci appear in 2 - 4 hr after 4-OHT treatment. The γ H2A.X foci were reduced in wild-type cells 12 - 24 hr after induction, whereas the foci kept accumulating in mutant cells and the diminish of γ H2A.X foci appeared to be very slow. We believe this difference in kinetics is primarily due to inefficient DNA repair pathways in IDH mutant cells.

The new result is included as Extended Data Fig. 1a and 1b. The modifications have been made to the manuscript on page 12, lines 291 – 292.

Cell cycle analysis should be presented to clarify how the mutants and rescued cells compare in cell cycle distribution. Presumably, activation of checkpoints in the IDH mutant cells will alter cell cycle distribution to some extent. Cell cycle directly influences DNA repair pathway utilization and so this needs to be excluded as the basis for any observed differences in repair properties.

We appreciate the comment from the reviewer. In our previous finding, IDH mutant enzyme does not induce substantial changes in cell cycle distribution in untreated cells, whereas a significantly elevated G2/M population could be observed under TMZ exposure⁹.

We performed a more dedicated EdU/PI flow cytometry analysis in the cell line models used in the present study. We confirmed similar cell cycle distribution between IDH1^{WT} and IDH1^{R132H} cells, with slightly decreased EdU incorporation in mutant cells.

	The elevated steady-state DNA damage/H2AX foci in the absence of damage is important and should be investigated further. There could be a number of explanations. If the cells are undergoing greater levels of replication stress, as they report, they may both be enriched in S-phase cells, which also normally have the highest incidence of DSB foci and cells with multiple DSB foci, and have higher numbers of foci specifically in S-phase cells. Post-mitotic DNA damage results in the formation of unusually large DSB foci that are typically only one or a few per nucleus and would reflect differences in DSB repair or checkpoint function at different stages in the cell cycle to explain the observed differences.	The reviewer raised a brilliant point about the replication stress. Schwartzman et al., reported recently that IDH-mutated cells exhibit heterochromatin-derived replication stress¹⁰. The replication stress is associated with elevated γH2A.X signal in mutant cells. The decreased EdU incorporation shown in the cell cycle flow cytometry above also indicates the presence of replication stress in IDH1^{R132H} cells. We are actively pursuing the investigation of the replication stress at this point and will present the findings through another dedicated manuscript.
The line scans are unnecessary in Figure 2e are unnecessary. The more important information present in the figure would be the percent of the TMZ-treated H2AX foci that are positive for RAD51. This should be quantified and added if the manuscript is resubmitted.	We appreciate the comment. We hope to use the line profile to highlight the spatial colocalization between molecules. As the reviewer suggested, we performed image analysis to highlight the colocalization of γH2A.X foci and RAD51. The data is in Extended Data Fig. 3a and 3b. The changes have been made to the manuscript on page 13, lines 329 – 332.
While the PLA assay supports the claim that there are direct interactions between CTCF and RAD51/BRCA2, the evidence in the literature for a direct interaction is limited. CTCF interacts with BRCA2 in a PARylation-dependent and Zinc finger-dependent manner. As zinc fingers are common PAR binding motifs and this interaction requires both PARylation and CTCF zinc fingers, this could reflect the association of both BRCA2 and CTCF with PAR polymers released as large complexes upon cell lysis. The PLA assay does not distinguish between these possibilities because DNA	We appreciate the comment from the reviewer. Regarding the interaction among CTCF, RAD51, and BRCA2, besides the PLA assay, our co-immunoprecipitation findings demonstrate the physical interactions among these DNA repair factors. We agree with the reviewer that using a control condition could be helpful to rule out the possibility of local high concentration.

damage sites are regions that are concentrated in all of these factors. All that it convincingly shows is that all three proteins are recruited to DNA damage sites. If a control showing that gammaH2AX or 53BP1 does not show PLA foci at double-strand breaks, it would be compelling that the method can discriminate between direct interactions and locally high concentrations. Optimally, with around 10-20 nm distances between antibodies, it's unclear that high local concentrations won't generate the same robust signal.

53BP1 is known having minimal interaction with CTCF or BRCA2, so we could perform PLA with antibodies against CTCF/BRCA2 as the control to solidify the conclusion from PLA results.

We discovered that the PLA signal is very low when using antibodies against RAD51 and 53BP1, suggesting that DNA repair proteins with different pathways should not give PLA signals, and thus distinguish direct interactions and locally high concentration. In contrast, the BRCA2/CTCF and RAD51/CTCF antibody combination gives potent PLA signals at the same assay, as shown below and in Figure 4. The inclusion of PARP inhibitor Olaparib (Ola) greatly reduces the BRCA2/CTCF and RAD51/CTCF foci. It seems in our model system, the PARylation is necessary for the assembly of BRCA2/RAD51/CTCF complex.

An additional validation step necessary to confirm that CTCF interactions are shared with those reported in the literature is to test the sensitivity of BRCA2 and RAD51 association with CTCF in the PLA assay. They are reported to be sensitive to PARP inhibition and mediated by PARylation. The authors should consider Figures 2 E, F; 3A-H; Because there is variation across the cell cycle with respect to DNA repair pathway utilization, it is not appropriate to show single cells to illustrate differences. Rather, an image showing cells representing different stages of the cell cycle is required to appropriately compare to samples.

We appreciate the comment from the reviewer. For the morphology analysis, we took images from multiple cells (> 60 per condition) through several biological replicates to rule out the influence of the cell cycle. We presented a single cell to provide clear visualization of the foci and protein colocalizations in Figures 2 E, F. We have updated Figure 4A-H by replacing the images for the PLA foci number assays with ones that display multiple cells.

Reviewer 3

We appreciate the input from the reviewer.

Reviewer 4

The manuscript presented by Lang et al entitled “*D-2-Hydroxyglutarate establishes BRCAness through epigenetic reprogramming*” explores the molecular mechanism responsible for the diminished DNA repair ability of IDH mutant malignancies, and, therefore, sensitivity to therapeutic targeting by DNA repair inhibitors. The authors reproduce these preliminary findings in their model system and demonstrate that the diminished DNA repair observed in IDH mutant cells is dependent on the mutant IDH neomorphic activity that results in production of D-2-hydroxyglutarate (D-2-HG), a known potent inhibitor of α -ketoglutarate dependent demethylating enzymes (including those acting on DNA, histone, and RNA). As exogenous addition of α -ketoglutarate can rescue these defects in DNA damage repair of IDH mutant cells, the authors go on to specifically evaluate a role for the DNA demethylating enzymes, TETs. They demonstrate overexpression of TET1 or TET2 can partially rescue the defect in DNA damage repair caused by D-2-HG production, suggesting a role for TETs in DNA damage repair. This role is additionally supported by the observations that TET1 or TET2 deficient cells also exhibit defects in DNA damage repair, and that sites of DNA damage accumulate the TET enzymatic product, 5hmC. Moreover, they show that local DNA damage-induced recruitment of known proteins associated with DNA repair by homologous recombination (including CTCF, BRCA2, and RAD51) is TET-dependent. Lastly, the authors explore whether hypermethylation-dependent loss of CTCF alters high-order chromatin structure changes in IDH mutant cells that are associated with double stranded breaks. Together these findings demonstrate a clear role for TET enzymes in the repair of DNA damage by homologous recombination. This has been previously suggested based on the reported co-localization of 5hmC with α H2AX at sites of DNA damage (Kafer, G.R., et al. Cell Rep 2016), and the TET catalytic-dependent regulation of DNA damage repair in response to phosphorylation by the DNA damage sensors ATR/ATM (Jiang, D.,

We appreciate the comments from the reviewer. The valuable insights greatly improve our work from its current form. We are happy to address the concerns from the reviewer and perform additional experiment for more evidence.

et al. Brain, 2015 & Jiang, D., et al. EMBO Reports, 2017). The authors are ultimately able to corroborate and validate these previously reported findings using multiple models of DNA damage (Genotoxic stress, Aphidicolin, DRGFP reporter system, and DlvA system). In addition, the authors are able to extend this observation to a new biological setting by demonstrating the loss of TET-dependent regulation of DNA damage in IDH mutant cells contributes to the mechanistic basis of their defects in DNA damage repair. Moreover, the authors are able to begin to unravel mechanistically how TET proteins contribute to DNA damage repair by aiding in the recruitment and assembly of DNA damage repair machinery (CTCF, BRCA2, and Rad51), which has not previously been reported to our knowledge. In total, this work advances our understanding of the contribution of TETs to DNA damage repair and establishes a novel molecular mechanism that underlies IDH mutant malignancies.

1. Is TET-dependent regulation of DNA damage repair the primary mechanism lost in IDH mutant cells?

The foundational premise of the manuscript is based on the observation that the defect in DNA damage repair in IDH mutant cells is due to D-2-HG production, which inhibits α -ketoglutarate demethylating enzymes, and that this defect can be rescued by exogenous addition of α -ketoglutarate. The authors explore the contribution of the TET DNA demethylating proteins to this defect by individual reconstitution of TET1 or TET2 by overexpression. While this provides some rescue over treatment with D-2-HG alone, the extent of rescue is variable between experiments (1f, 1j, 2c, 2d, 3b, 3e, 3f, 3g) and based on the figures does not always reach the level of rescue achieved with exogenous addition of α -ketoglutarate. As the text implies that D-2-HG-dependent defects in DNA damage repair are TET-dependent, it would be important to address this discrepancy in the ability of either TET1 or TET2 to rescue this defect compared to α -ketoglutarate. Is this discrepancy due to inefficient expression of TET proteins, the need to have both TET1 and TET2 to achieve full rescue or the involvement of other pathways in D-2-HG-dependent inhibition of DNA damage?

As previous mechanisms have been reported to contribute to the D-2-HG-dependent inhibition of DNA damage repair (Sulkowski, P.L.,

We appreciate the insight from the reviewer. We agree that a more in-depth investigation on the dependency on TET1 and TET2 could be more convincing. As the reviewer correctly point out, the D-2-HG dependent DNA repair deficiency may be derived from multiple paralleled mechanisms. Sulkowski et al. discussed the involvement of H3K9me3 and DNA repair deficiency, which could be a parallel mechanism with our proposed theory.

Through the revision, we detected the rescue effects of α KG on DDR in the absence of TET1 and TET2 in U251 cells. We discovered that α KG resulted in less DDR with no TET1 or TET2 in the cell, although to a less extent in TET1/2 proficient cells. These results suggest that several mechanisms may be involved in D-2-HG-associated DNA damage sensitivity, including TET1/2-dependent and TET1/2-independent pathways.

et al. Nature 2020, Wang, P., et al. Cell Reports, 2015), it would be important to address, experimentally, whether rescue relies on restoration of both TETs or a contribution of these other pathways. This is especially important as in the discussion the authors describe the mechanism identified in this manuscript to be a “parallel” mechanism to that described within Sulkowski, P.L., et al (Line 429).a. While simultaneous reconstitution of both TET1 and TET2 would be functionally difficult to achieve to assess rescue, the authors should do the dual TET1 and TET2 knockout within their experimental model system of TMZ treatment with D-2-HG and assess whether this double knockout completely abrogates the ability of α -ketoglutarate to rescue the defect in DNA damage repair. Additionally, they could assess aspects of the histone demethylase mechanism described by Sulkowski et al and determine if this mechanism is active in their model system. This could be achieved by evaluating H3K9me3 deposition around DNA breaks in experiments similar to those in Figure 2h and 3g.

Further, we validated the involvement of H3K9me3 methylation in Sce-I DRGFP U2OS cells. We recorded a rapid transient increase in H3K9me3 modification upon DSB induction, whereas the H3K9me3 level was maintained at a high level in cells exposed to D-2-HG. This aligns with the findings shown in the previous study⁸. Importantly, α KG, but not TET1 or TET2 overexpression resulted in the rescue of the H3K9 coverage, suggesting the involvement of H3K9me3 guided DNA repair mechanism.

The new result has been included as Extended Data Fig. 8a and 8b. The changes have been made to the manuscript on page 17, lines 452 – 456.

	Overall, our findings suggest the presence of paralleled DNA repair mechanisms that are affected by D-2-HG and epigenetic reprogramming.
b. These experiments would provide clarity to whether the TET-dependent mechanism is the sole mechanism active within this model of IDH mutant malignancy as is currently implied. Should inhibition of TET function be the sole mechanism responsible for the observed defects in DNA damage repair in this system in response to D-2-HG, the authors should discuss 1) the relative differences between their model of IDH mutant malignancy and the other previously described models, 2) why these different IDH mutant models rely on distinct molecular mechanisms for D-2-HG disruption of DNA damage repair, and 3) what these differences mean with respect to understanding IDH mutant defects in DNA repair. If, however, both mechanisms are active in this model system, then the presence of this other contributing mechanism should be more clearly indicated in the results.	We appreciate the comment from the reviewer. With the new findings above, we incline to believe there are multiple paralleled mechanisms to establish DNA repair deficiency in IDH-mutated cells. With more data available throughout the revision, we could be more confident to address this claim.
2. Clarity and interpretation of figure 5 and extended data figure 5. As they are currently presented, these figures have limited details present in the legend/figures that make technical and biological interpretation difficult and raise additional questions that require addressing/clarification.	We apologize the limited details presented in the legend and figures. We have included more information as indicated by the reviewer. The new content could be found in the method section on page 5 – 11, as well as the updated figure legends.
a. Extended data figure 5 shows the 4C signal and CTCF ChIP for two selected double stranded breaks. This figure could benefit from the addition of a distance scale to give the reader some indication of the physical spread of contacts and CTCF binding sites in the WT and IDH mutant cells at baseline and in response to DNA damage.	We appreciate the comment from the reviewer. We have included a better ruler with annotations on Asis I sites and viewpoints used for 4C-seq analysis. The new panel is included as Extended Data Fig. 8h. The modifications have been made to the manuscript on page 18, lines 479 – 480.

i. As it is presented, the CTCF ChIP peaks appear to be from untreated WT and IDH mutant cells. Is this correct? Has the CTCF peak status been assessed in the 4OHT treated cells to specifically evaluate changes in CTCF localization around the potential sites of breaks in response to damage? While the authors show the 4-OHT treatment and CTCF peaks in Figure 5d, this is for bulk localization and does not show individual CTCF peaks around the potential and actual break sites.	We appreciate the comment from the reviewer. The CTCF peaks in the previous version are from non-treated cells. The line profiles and heatmaps show the CTCF peaks around 80 of Asis I sites. As the reviewer recommended, we have updated the CTCF peaks before and after 4-OHT treatment in the revised Figure 5a and 5b for better clarity. In addition, we have included ChIP-seq tracks to better illustrate CTCF peaks next to the DSB sites. The data is also aligned with the 4C-seq signals. Our finding highlights the decreased CTCF coverage in IDH1^{R132H} cells, regardless of the presence of DSB. The decrease of CTCF coverage aligns with the lack of 4C signal changes, suggesting the compromised conformation adjustment in the DSB sites.

The new results have been included as Figure 5a and 5b. The changes have been made to the manuscript on page 17, lines 465 – 466.

b. Figure 5e is bisulfite sequencing for IDH WT and mutant cells. Additional information in the legend would aid readers unfamiliar with this technique in interpreting this analysis. What do dark and light circles indicate? What are the column and row numbers indicating? Additionally, are the presented results for the cells with or without 4-OHT?

We appreciate the comment. Each column indicates one CpG island that is close to a CTCF peak next to the DNA damage sites. As the bisulfite reaction leads to mixed genotypes in the CpG islands, we performed Sanger sequencings on multiple clones (row) from each site and calculated the incidence of methylated CpG islands.

We have included figure legends in the data legend for better clarity. We have also included both untreated and treated cells in the updated Figure 5c.

i. The authors should include the sequence of the selected CTCF peaks for this methylation analysis. In this sequence they should indicate the potential sites of methylation and the predicted/actual CTCF binding site to aid the readers in determining whether there is specifically disruption of CTCF binding based on increased methylation within the CTCF consensus sequence as the authors hypothesized.

We appreciate the comment. We have included the genomic regions selected for methylation analysis in supplementary data.

ii. This figure aims to support the connection between methylation status and CTCF binding across the genome impacting high-order chromatin structure. However, the presented data is evaluating CpG methylation status for one author-selected CTCF peak (indicated in extended data figure 5) flanking a double stranded break for two double stranded breaks in a model system that has ~100 breaks. Based on this limited analysis, it is difficult to ascertain whether these results are largely representative of the relationship between DNA hypermethylation and CTCF binding around potential sites of double stranded breaks. Additionally, as it is presented, it is unclear as to whether the observed hypermethylation is at baseline or in response to a double stranded break in the WT and IDH mutant cells making it difficult to interpret the biological meaning of these results. The authors should expand their analysis to further substantiate the claims that DNA hypermethylation is preventing CTCF binding and therefore higher-order chromatin structures.

We appreciate the comment from the reviewer. The DNA hypermethylation preventing CTCF binding has been previously demonstrated¹¹. To enhance the present study, we have selected 4 additional CTCF/CpG sites to confirm the changes in DNA methylation on CTCF binding sites. We have also included both untreated and treated cells for the methylation analysis.

The new results are included as Extended Data Fig. 8i. The changes have been made to the manuscript on page 18, lines 480 – 482.

Further, we evaluated the association of CTCF to the chromatin upon DSB establishment and TET1 activity. We discovered that CTCF chromatin association is decreased in IDH1^{R132H} cells, whereas ectopic expression of the TET1 catalytic domain (CD) rescues the CTCF chromatin association. We believe this finding suggests that IDH1 mutation and D-2-HG compromise CTCF

chromatin binding through a TET-dependent manner, presumably by affecting DNA methylation status throughout the genome.

The new findings have been included in Extended Data Fig. 8g. The changes have been made to the manuscript on page 18, lines 475 – 476.

iii. To that end, does the methylation status in this experiment change with or without 4OHT treatment?

As discussed above, the DNA methylation analysis on six different genomic loci indicates that methylation status is determined by the presence of IDH1 mutant, but less on 4-OHT treatment.

c. Figure 5f/g. There is no scale on the y-axis to discern the relative peak size across treatment groups. What do the asterisks and arrow in the figure indicate? This could also benefit from a genomic distance scale. Is f the data for the break one and g for the break two from figure 5e?d. Figure 5h/i. What is the difference between the left and right plots for figure h/i and what is the relationship of these plots to f and g? How was the 4C signal quantitated? Is this taking all signal from f/g or simply the yellow highlighted sections?

We appreciate the comment from the reviewer. We have included more details for Figures 5f and 5g, which are currently assigned to Figure 5b. The 4C signal is quantified based on the normalized read depth from the ectopic peaks. The changes in 4C signal are calculated with $\text{Log}_2[(+DSB/-DSB)]$ ratio. To better illustrate the changes of 4C signal throughout the DSB loci, we plotted the 4C ratio data track using a previous method¹². Our finding reveals elevated 4C signal at the DSB sites within a spanning of several million base pairs, which is likely within DSB-associated TAD. The 4C signal change is less observed in IDH1^{R132H} cells.

	B Chr1 (DSB1) Chr20 (DSB2) 80 AsisI_90 100 25 30 AsisI 35 40 4C-seq(+DSB/-DSB) Viewpoint IDH1^{WT} IDH1^{R132H} 10 -DSB +DSB CpG CTCF enrichment IDH1^{WT} IDH1^{R132H} 10 -DSB +DSB
3. Experimental design, statistics, materials and methods. a. As the paper is currently written there is no information regarding the number of independent experiments that have been completed for any of the experiments/figures. While the methods denote that the presented data are the mean +/- SEM there is no indication for individual figures as to the number of independent experiments that have been performed and whether the presented data are representative of independent experiments or a compiled analysis of all independent experiments.	We have included information regarding the number of biological replications (independent experiments) that have been performed. The information is included in the method section and updated figure legends.
b. The statistical section of the methods simply says t-tests and one-way ANOVA were used. Interpretation of the analysis would be improved by providing the individual test used for each analysis within the figure legends. Additionally, the presentation of statistical measurements across figures is variable and for some figures it is difficult to ascertain what the statistical comparison being presented	We appreciate the comment from the reviewer. We have run through the manuscript to ensure consistency in statistics and the presentation.

is. For example: 1f uses lines between compared groups, 1k uses brackets, 3b uses asterisks without a line such that the comparison is unclear. A unified presentation of the statistical analysis would make it easier for the reader to interpret.																			
i. The methods state statistical significance is set as less than 0.05. However, figures use both * and ** to indicate significance. What is the difference between these significance indications?	We have included explanations of * and ** in the method section and figure legends.																		
ii. For figure 1K is the right bracket indicating that all of the 5 right groups are statistically significantly increased compared to the third group? The text seems to imply this is what is meant.	We appreciate the comment. We have updated the format in Figure 1K with better clarity.   <caption>Data for Figure 1K</caption>   Condition GFP⁺ (%)     Scp-1 ~1   Scp-1 + ~9   Scp-1 +, D-2-HG + ~3   Scp-1 +, D-2-HG +, α-KG + ~8   Scp-1 +, D-2-HG +, α-KG +, TET1 + ~5.5   Scp-1 +, D-2-HG +, α-KG +, TET2 + ~7   Scp-1 +, D-2-HG +, TET1 +, TET2 + ~5.5   Scp-1 +, D-2-HG +, TET2 + ~7   	Condition	GFP ⁺ (%)	Scp-1	~1	Scp-1 +	~9	Scp-1 +, D-2-HG +	~3	Scp-1 +, D-2-HG +, α-KG +	~8	Scp-1 +, D-2-HG +, α-KG +, TET1 +	~5.5	Scp-1 +, D-2-HG +, α-KG +, TET2 +	~7	Scp-1 +, D-2-HG +, TET1 +, TET2 +	~5.5	Scp-1 +, D-2-HG +, TET2 +	~7
Condition	GFP ⁺ (%)																		
Scp-1	~1																		
Scp-1 +	~9																		
Scp-1 +, D-2-HG +	~3																		
Scp-1 +, D-2-HG +, α-KG +	~8																		
Scp-1 +, D-2-HG +, α-KG +, TET1 +	~5.5																		
Scp-1 +, D-2-HG +, α-KG +, TET2 +	~7																		
Scp-1 +, D-2-HG +, TET1 +, TET2 +	~5.5																		
Scp-1 +, D-2-HG +, TET2 +	~7																		
iii. For figure 2C/D and 3E/F is the right line indicating the fifth group is statistically different from the second or is it that groups 3, 4, and 5 are statistically significantly increased compared to the second group? The text implies all three are able to rescue but the way that it is denoted by line would traditionally only indicate the two connected groups by the line are significantly different.	We have reformatted Figures 2C, 2D, 3E, and 3F for better clarity.																		
iv. For figure 3b what is being compared for the presented statistical analysis?	We have reformatted Figure 3B for better clarity.																		
v. Clarification of these statistical analyses will be important for ensuring proper analysis and biological interpretation of the experiments.	We appreciate this comment and have revised the manuscript for the details of statistical analyses.																		
c. The methods, as they are currently written, are not sufficiently detailed to allow for reproduction of all experimental procedures and some methods need to be added including: 5hmc immunofluorescence, siRNA experiments for knockdown, how the	We have included details in the experiment procedures including 5hmc immunofluorescence, RNA interference for knockdown, CRISPR-mediated TET1/2 KO, the 4C-seq data analysis, and primers used for ChIP with the DR-GFP reporter system. The																		

knockouts were generated for TET1 and TET2, the 4C data analysis and quantification, and primers used for ChIP with the DR-GFP reporter system.	updated details can be found in the method section on page 5 – 11, and in supplementary files.
i. Additionally, it would dramatically aid readers to have a more detailed explanation of the individual components of the DR-GFP reporter and DiVA systems to improve their understanding of these model systems. To that end, figures diagramming the general basis of these model systems would dramatically aid the reader in easily understanding their utility.	We have included details in the experiment procedures and composed a schematic illustration of the DRGFP system. Please find the information in the method section and at the bottom of Figure 2h. More details in the DD-Sce-I-GR DRGFP U2Os system could be found in the method section on page 5, lines 99 – 112.
4. Conceptual contribution of Figure 5, and inconsistent/overstatement of scientific claims. a. In its current form figures 1-4 build to a model of TET-dependent regulation of DNA damage repair at the site of DNA damage that cohesively fit together. Figure 5, however, begins to explore the contribution of higher-order chromatin structures/TADs to the regulation of DNA damage and whether these are correlated with changes in methylation status/CTCF binding. While tangentially related, figure five feels like the first figure of a new paper focused on exploring the contribution of these structures to DNA damage repair, not the conclusion of the first four figures, especially considering major comment number 2. The impact of the presented work would benefit from further substantiating the claims in figures 1-4 and saving figure 5 for a future work where the contribution of these higher-order structures could be further evaluated and substantiated.	We appreciate the comment from the reviewer. Indeed, Figure 5 describes our discoveries that are a step further from the mechanistic study based on TET/CTCF. We feel strongly that the TAD and DNA topology provide an important explanation of how CTCF guides the DNA repair pathway, which is a key step in D-2-HG-associated BRCAness. We have performed additional experiments to strengthen the link among TET, CTCF, and chromatin conformation in the revised manuscript.
i. Along these lines there seems to be competing presentation of two mechanistic bases for the TET-dependent regulation of DNA damage repair within the text: 1) methylation dependent changes in chromatin organization/structure that impairs DNA damage repair by changing overall CTCF localization, 2) methylation-dependent prevention of CTCF binding based on DNA sequence methylation and recruitment to local sites of damage. It is possible that both mechanisms contribute to DNA damage repair; however, the respective contributions are not immediately clear based upon the currently presented data. b. To this end, as it is currently written, the abstract is not representative of the findings and to some extent overstates the presented work and its conclusions. The synopsis of the work provided at the end of the introduction appears to be a more	We appreciate the comment from the reviewer. The reviewer raised an important point regarding how CTCF is involved in DNA repair. We believe the involvement of CTCF is associated with DSB repair through multiple dimensions, ranging from defining the DSB-associated TAD boundary to facilitating the recruitment of DDR factors. Many of these processes may be sensitive to local chromatin methylation status. The CTCF-guided DDR factor recruitment has been reported previously^{5,6}. Our study extended these seminal studies by linking CTCF with chromatin topological adjustment, a phenomenon also guided by CTCF. In Figure 3g and Extended Data Fig. 8g, we showed that ectopic expression of TET1/TET2 rescued CTCF recruitment, which

accurate reflection of the presented work. This should be addressed by adapting the abstract to be more reflective of the included findings or performing additional experiments to support these statements. As it is currently written, the abstract reads as though the CTCF high-order chromatin structure is directed by TETs and the loss of these structures forms the basis for failed DNA repair in IDH mutant cells. However, this manuscript, in its current form, does not establish these connections, but instead reveals a local TET-dependent recruitment of DNA repair proteins at sites of DNA damage (based on the local 5hmC deposition by IF and MeDIP, within a range of ~2000 bp around the cut site). Additionally, the work presents evidence to suggest IDH mutant cells also exhibit diminished remodeling of high-order chromatin structures (50kb-1Mb around cut site) in response to breaks (however with some caveat based on major comment 2), but the relationship of these structures to TET function and the contribution of these structures to DNA damage repair remain to be fully elucidated. If the authors wish to include additional experiments to support these statements, they would need to demonstrate the reliance of these damage induced contacts on TETs and CTCF potentially using the rescue by TETs and CTCF knockdowns used to establish the local relationship at sites of damage. Moreover, the contribution of these more distant contacts to the repair process would need to be assessed.

supports the claim that DNA methylation status would affect CTCF chromatin association.

Further, we conducted an additional 4C-seq experiment to understand the connection among CTCF, TET catalytic function, and high-order chromatin structure.

We found that CTCF depletion through RNA interference resulted in substantial changes in the 4C signal that is associated with DSB induction. The changes of the 4C signal were transformed into $\log_2(\text{siRNA}/\text{Control RNA})$. Loss of chromatin contact will be shown as negative values in the plot. This finding suggests that CTCF is necessary to modulate the TADs that are associated with DDR.

Further, we evaluated the connection between TET catalytic activity and DSB-associated DNA topological adjustment. To this end, we reached out to Dr. Anjana Rao's group and obtained the TET1 catalytic domain (CD) constructs. The 4C-seq analysis on these cell lines revealed that TET1 catalytic activity substantially improved the chromatin interaction upon DSB induction compared to the mutant TET1 dCD, suggesting the cytosine methylation status is closely connected to the organization of DSB-associated TAD. The 4C-seq data is transformed as $\log_2(\text{TET CD} / \text{TET dCD})$. Increased chromatin contact is shown in positive value in the graph.

These new findings are included in Extended Data Fig. 8e and f. The changes have been made to the manuscript on page 17, lines 467 – 470.

Overall, our additional findings strongly indicate the cytosine methylation status at the DSB sites affects the organization of DNA repair-associated TAD, probably in a CTCF-dependent manner.

c. Line 432. *“Overall, our findings reveal the molecular mechanisms of metabolic and epigenetic predisposition on “BRCAness” in malignancies.”* This is a strong statement that overstates the applicability of the work in its current form. While this work does reveal a novel contribution of TET- dependent regulation of the DNA damage response to IDH mutant malignancy, the presented work is largely confined to one mutant condition in one cell type and therefore statements of this broad utility to malignancy in total seem inaccurate. Additionally, considering other mechanisms have been proposed (see major comment 1) which are not assessed in this work, this mechanism, based on the data available at present, seems unlikely to be the sole basis of “BRCAness in malignancy”

We appreciate the comment from the reviewer. We have adjusted the wording for better accuracy. The changes have been made to the manuscript on page 20, lines 537 – 538.

5. Additional experimentation/work to bolster findings.

a. Given the 5hmC is co-localizing to these sites of DNA damage, it appears that catalytic activity of TETs is required for this process, which is further supported by the rescue with TET1 or TET2. These findings would be significantly bolstered by performing these rescue experiments with a catalytically dead (HxD) mutant to further confirm that local TET activity at the time of damage is required for this response.

b. The time-dependent establishment of γ H2AX foci and the subsequent recruitment of CTCF, BRCA2, and RAD51 to sites of DNA damage in figure 3g very nicely contribute to the mechanistic basis of TET-dependent regulation of this response. Figure 3b connects these findings with the local deposition of 5hmC (and 5mC in Extended Figure 3c) using the same model system, although it is unclear at what time point in the response this information is collected? Additionally, is this performed only using GFP+ cells in this reporter system? The manuscript would benefit by performing a similar time course as experiment 3g for the 5hmC and 5mC to show the local TET-dependent regulation of DNA methylation in response to breaks. This would also reveal if TETs are acting simply to deplete existing 5mC at these sites.

c. Addition of a model to collectively describe the included findings and their contribution

6. Experimental concerns.

a. Figure 1H: In the TET1 western blot image, why are lanes 2 and 3 so white? Even the background greying shows loss in this area.

We appreciate the comment from the reviewer. As shown above in point 4i, we have included TET1 CD and TET1 dCD construct to confirm the role of TET activity in DNA repair and chromatin topological adjustment.

We appreciate the comment from the reviewer. In Figure 3b, the DSB is induced by 0.5 μ M Shield1 and 0.2 mM TA for 2 hr in the DRGFP U2OS cells. We have updated the experimental details in the revised manuscript on page 30, lines 724 – 727.

As the reviewer recommended, we performed a time course measurement through MeDIP and hMeDIP assay to show local TET-dependent methylation status by measuring 5mC and 5hmC levels. We discovered that while 5mC levels are generally stable among treatments, DSB results in a transient elevation in the 5hmC level about 2 hr after DSB induction. The 5mC level is elevated with the presence of D-2-HG and could be reversed with TET1 or TET2 overexpression.

The new findings are included as Figure 5e. The changes are made to the manuscript on page 18, lines 478 – 480.

We have included a model illustration for the findings as Extended Data Fig. 9.

The low signals in these areas might be caused by the short exposure time or lack of recognized antigens by antibodies. We have provided the original blotting image for Figure 1h.

	b. Figure 1L: For the DDK blot, why are lanes 1 and 4 so white? Again, the background greying shows loss in these regions.	We have provided the original blotting image for Figure 1L. c. Figure 4E: Part of the y-axis title seems to be missing based on f-h.	We have included the y-axis information with the resubmitted files.
Minor comments: 1. Language use and writing. a. At times the authors refer to gH2AX and pATM using the word expression. However, expression is generally reserved for processes associated with transcription and translation. Given gH2AX and pATM are phosphorylation based post-translational events it would improve clarity for the reader by selecting another word in place of expression, such as modification.	We appreciate the comment from the reviewer. We have revised the wording accordingly.
b. The title for figure three refers to “cytidine hypermethylation,” however, everywhere else in the manuscript it is cytosine, which is the standard for the field in referencing this process.	We have revised the wording to ensure consistency. The change has been made to the manuscript on page 29, line 718.
c. Line 416. “D-2-HG blockade on CTCF binding and concomitant DNA repair complex establishment (Fig. 4).” It is unclear what this sentence is intended to convey. As it is currently written, it seems incomplete.	We have revised the wording accordingly. The change has been made on page 19, lines 520 – 521.
The presented manuscript by Lang et al provides an important confirmation of the previously suggested role for TETs in DNA damage repair, which has been met with some skepticism based on the limited number of observations presented to date. In addition, the authors begin to elucidate some key aspects of the mechanistic basis by which TETs aid in DNA damage repair and demonstrate how this is relevant to IDH mutant malignancies.	We appreciate the comment from the reviewer. We have carefully gone through all the comments and will revise our work accordingly.

However, in its current form, the presentation and interpretation of the completed work requires some revision to ensure the claims are supported by the work and accurately reflect the included content (see comments above).	
References written in bold in this response are not currently included in the manuscript but should be cited	We have cited the reference as indicated.
Synopsis of comments The manuscript by Lang et al entitled "D-2-hydroxyglutarate establishes "BRCAness" through epigenetic reprogramming" identifies and begins to mechanistically explore a role for TET proteins in the regulation of DNA damage repair. Moreover, this work assesses the contribution of this pathway to the failed DNA damage repair evident in IDH mutant malignancy. These findings present important new findings to the field. However, there are some aspects of the work that require revision/clarification prior to publication. These are indicated in a brief synopsis of the major points below:	We appreciate the comment from the reviewer.
1. The authors need to validate whether the TET-dependent regulation of DNA damage repair is the sole mechanism lost in IDH mutant cells as they suggest, by demonstrating that double deletion of TET1 and TET2 prevents the ability of α-ketoglutarate to rescue the DNA damage repair defect in IDH mutant cells. They also need to acknowledge the other proposed mechanism (H3K9 methylation) in the Results section instead of only in the Discussion.	We have made the revisions as indicated above.
2. Data figure 5 and extended data figure 5 require revision to improve clarity as to the experiments performed to be able to interpret their contribution to the work.	We have made the revisions as indicated above.
3. The materials and methods, statistical analysis and experimental details need to be revised such that there is sufficient information for reproduction of the included work and interpretation of the results.	We have made the revisions as indicated above.
4. The manuscript requires editing such that the written text accurately reflects the experimental figures to ensure that the scientific claims are sufficiently substantiated.	We have made the revisions as indicated above.
5. Additional experiments are needed to demonstrate 1) the contribution of TET catalytic activity to the local response to double stranded break repair, 2) the time dependent establishment of 5hmC at sites of DNA damage.	We have made the revisions as indicated above.

Reviewer 5

References:

- 1 Xu, W. *et al.* Oncometabolite 2-hydroxyglutarate is a competitive inhibitor of alpha-ketoglutarate-dependent dioxygenases. *Cancer Cell* **19**, 17-30 (2011). <https://doi.org:10.1016/j.ccr.2010.12.014>
- 2 Flavahan, W. A. *et al.* Insulator dysfunction and oncogene activation in IDH mutant gliomas. *Nature* **529**, 110-114 (2016). <https://doi.org:10.1038/nature16490>
- 3 Rahme, G. J. *et al.* Modeling epigenetic lesions that cause gliomas. *Cell* **186**, 3674-3685 e3614 (2023). <https://doi.org:10.1016/j.cell.2023.06.022>
- 4 Lu, J. *et al.* CNTF receptor subunit alpha as a marker for glioma tumor-initiating cells and tumor grade: laboratory investigation. *J Neurosurg* **117**, 1022-1031 (2012). <https://doi.org:10.3171/2012.9.JNS1212>
- 5 Hilmi, K. *et al.* CTCF facilitates DNA double-strand break repair by enhancing homologous recombination repair. *Sci Adv* **3**, e1601898 (2017). <https://doi.org:10.1126/sciadv.1601898>
- 6 Lang, F. *et al.* CTCF prevents genomic instability by promoting homologous recombination-directed DNA double-strand break repair. *Proc Natl Acad Sci U S A* **114**, 10912-10917 (2017). <https://doi.org:10.1073/pnas.1704076114>
- 7 Fitieh, A., Locke, A. J., Mashayekhi, F., Khaliqdina, F., Sharma, A. K. & Ismail, I. H. BMI-1 regulates DNA end resection and homologous recombination repair. *Cell Rep* **38**, 110536 (2022). <https://doi.org:10.1016/j.celrep.2022.110536>
- 8 Sulkowski, P. L. *et al.* 2-Hydroxyglutarate produced by neomorphic IDH mutations suppresses homologous recombination and induces PARP inhibitor sensitivity. *Sci Transl Med* **9** (2017). <https://doi.org:10.1126/scitranslmed.aal2463>
- 9 Lu, Y. *et al.* Chemosensitivity of IDH1-Mutated Gliomas Due to an Impairment in PARP1-Mediated DNA Repair. *Cancer Res* **77**, 1709-1718 (2017). <https://doi.org:10.1158/0008-5472.CAN-16-2773>
- 10 Schvartzman, J. M. *et al.* Oncogenic IDH mutations increase heterochromatin-related replication stress without impacting homologous recombination. *Mol Cell* **83**, 2347-2356 e2348 (2023). <https://doi.org:10.1016/j.molcel.2023.05.026>
- 11 Flavahan, W. A. *et al.* Insulator dysfunction and oncogene activation in IDH mutant gliomas. *Nature* **529**, 110-+ (2016). <https://doi.org:10.1038/nature16490>
- 12 Arnould, C. *et al.* Loop extrusion as a mechanism for formation of DNA damage repair foci. *Nature* **590**, 660-665 (2021). <https://doi.org:10.1038/s41586-021-03193-z>

Revision NCOMMS-24-13290-T

Reviewer 1	
In this manuscript, Lang et al seek to determine the role of D-2-HG in chromatin remodeling via CTCF and its inability to recruit BRCA2 and RAD51. In this way they propose that the TET1/2 deficient phenotype leading to hypermethylation in IDH mutant glioma assists in revoking CTCF from binding to the DNA – preventing the final complex formation. Ultimately, this group proposes that the oncometabolite D-2-HG induces an unstable genome with different TADs and chromatin structure via CTCF/BRCA2/RAD51 – in the context of genotoxic stress. While the idea is overall interesting, the models used are insufficient to know how generalizable this finding is. This study is also limited to a few DNA repair proteins, where it was unclear how these proteins were chosen over the many other players in DNA damage repair. There is also a lack of time course data in different models, which would further strengthen the findings of this paper. Overall, this paper seems to portray a logical next step rather than a large jump in the DNA repair field, in the context of IDH mutation and genotoxic stress.	We appreciate the author providing the summarization of our work. We are happy to follow the comments from the reviewer and improve our study, with an emphasis on improving the cell models and time course study.
Major Comments: 1. The goal of this study seems to be to link together known observations from previous groups to show that D-2-HG is an important metabolite that regulates TET1/2 function, CTCF affects genome architecture, and that D-2-HG affects the kinetics of CTCF/BRCA2/RAD51 complex formation.	We appreciate the insight from the reviewer. Our study is enlightened by the seminal studies that describe the inhibitory role of D-2-HG on TET1/2 function¹. The reports by Flavahan et al²., and also later by Rahme et al³., from the same group discussed the loss of CTCF binding in cell lines with D-2-HG and TET malfunction. We believe besides the well-characterized CTCF/insulator malfunction, D-2-HG is relevant to other cancer biology changes in IDH-mutated cancers, such as DNA repair pathways.
2. While this work does show some new findings in the context of IDH mutant glioma, there is only one glioma model used – U251 – which is not a common or representative model of glioma, much less IDH mutant glioma. The other cell line used is U2OS, which while commonly used in DNA repair, is not a glioma cell line. Endogenous IDH mutations are known to function differently from overexpression models – especially in a model where long-term chromatin changes are what are being investigated. Therefore, this model must be	We appreciate the comments from the reviewer. We used several well-accepted cell line tools, such as U251 IDH1^{R132H}, U2OS Sce-I/DRGFP U2OS, and Asisl-ER (DivA) U2OS IDH1^{R132H} cells to explore the changes in DNA repair pathway in the context of IDH mutation/D-2-HG. We agree that validating these findings in disease-relevant glioma cell line models with intrinsic IDH mutation will be important for the results to be convincing.

shown in an endogenous IDH mutation cell line model for the results to be convincing.

We selected patient-derived glioma stem cells (GSCs) BT054 and BT142 that carry intrinsic IDH1 mutation and validated our findings. We showed through a dose-response analysis that the TMZ resistance in IDH-mutated GSCs could be enhanced with α KG or TET1/2 overexpression, suggesting the cytosine methylation is suppressing the DNA repair pathway. The new results have been included in Extended Data Fig. 7. The changes have been made in the manuscript on page 17, lines 446 – 449.

Besides, we investigated the recruitment of RAD51 in these GSCs. Our findings indicate that RAD51 recruitment to TMZ-induced γ H2A.X puncta is enhanced with the presence of α KG or TET1/2 overexpression. We believe this result indicates that reversing the cytosine methylation leads to improved homologous recombination DNA repair, which aligns with our finding in the U2OS and U251 cells. The new results are included in Extended Data Fig. 7a to 7f. The changes have been made to the manuscript on page 16, lines 439 – 441.

Puncta analysis confirmed more colocalization of γ H2A.X with RAD51 in puncta with the presence of α KG or TET1/2 overexpression.

In addition, α KG or TET1/2 overexpression reduced an overall number of γ H2A.X puncta, suggesting a reduced DNA damage.

Moreover, we conducted Western blotting experiments to confirm that the TMZ-induced DNA damage in either BT054 or BT142 cells could be relieved with the presence of α KG or TET1/2 overexpression. The new data is included as Extended Data Fig. 7g and 7h. The changes have been made to the manuscript on page 16, line 439 – 441.

Finally, we performed laser micro irradiation experiments in BT054 cells. We confirmed that α KG or TET1/2 overexpression would enhance the recruitment of RAD51 to the DSB loci, evidenced with colocalization of γ H2A.X and RAD51 puncta. The new result has been included as Extended Data Fig. 7g and 7h. The changes have been made to the manuscript on page 16, lines 434 - 436.

	 Percentage of RAD51 overlapping with γH2A.X    Condition Percentage of RAD51 overlapping with γH2A.X (approx. median)     DMSO 0.1   αKG 0.8   TET1 OE 0.5   TET2 OE 0.5   	Condition	Percentage of RAD51 overlapping with γ H2A.X (approx. median)	DMSO	0.1	α KG	0.8	TET1 OE	0.5	TET2 OE	0.5
Condition	Percentage of RAD51 overlapping with γ H2A.X (approx. median)										
DMSO	0.1										
α KG	0.8										
TET1 OE	0.5										
TET2 OE	0.5										
3. Additional experiments, evidence, and models are needed. One of the main points of this manuscript is to connect IDH mutant glioma with chromatin remodeling via D-2-HG. However, throughout the manuscript an endogenous IDH mutant model is not used. As The theory behind D-2-HG is the ability to inhibit enzymes, this will create lasting changes vs an overexpression model in U251 cells, and an osteosarcoma line U2OS. At a minimum, one endogenous IDH mutant model needs to be used to show that this is a tractable finding. Also, how would the authors propose to test this in a more clinically relevant model, or patient samples?	We appreciate the comments from the reviewer. As included in the responses to point 2, we will use disease-relevant GSC cells to validate our findings. Glioma stem cells are recently developed in vitro models that recapitulate the cancer biology and therapy resistance from their tumor origin, and hence could serve as a more clinically relevant model than the U251 and U2OS cell lines. We also included AG-120, a pharmacological IDH1 mutant inhibitor for the experiments using these cells, to evaluate the changes in DNA repair efficiency when neomorphic activity is suppressed. The example could be found in Extended Data Fig. 2a and 2b, lines 295 - 302.										
4. Data analysis seems accurate, but quantification of multiple cells per model is needed. For all the DD-Scel-GR reporter cells and DivA assays, are these quantifications from one cell each? If so, more than one cell per condition needs to be quantified. A time course should also be completed with many of these assays as the methods state the cells are allowed to recover for 10 minutes – there may be a recovery of RAD51/BRCA2 recruitment if given more time. This is true for the ChIP PCR as well – these cells may just need longer to recruit BRCA2/Rad51 instead of stating these proteins are not recruited.	We appreciate the comments from the reviewer. For morphological data, we quantify at least 40 cells among several biological replicates to obtain statistically meaningful data. The reviewer raised a brilliant point that a time course experiment will help understand the dynamic changes during DNA repair. Following this guidance, we tracked the recruitment of RAD51 through time-lapse microscopy. DNA damage was established through laser micro-irradiation on RAD51-RFP-transfected U251 cells. Within one hour of observation, we found that with the presence of D-2-HG, the RAD51 recruitment to the DNA damage sites are significantly delayed.										

Similarly, we performed a time course imaging on BT054 cells carrying intrinsic IDH1 mutation. We discovered that the RAD51 recruitment could be enhanced with the presence of α KG. These findings suggest that the recruitment of DNA repair proteins such as RAD51 is significantly reduced with the presence of IDH mutation or D-2-HG.

These new results are included as Extended Data Fig. 7j. to 7m. The modifications have been made to the manuscript on pages 16-17, lines 442 – 445.

5. While common DNA repair assays are used, there is no connection to IDH mutation tumorigenicity. As this paper seeks to better understand therapeutic vulnerabilities in IDH mutant GBM, some translational aspect should be tested to show that this pathway is important for tumor cell growth. Throughout the paper we only are shown DNA repair assays, but no cell growth, viability, or cell death assays. –

We appreciate the comments from the reviewer. As the reviewer pointed out correctly, our present study seeks to understand the therapy vulnerabilities in IDH-mutated cancers, with a focus on DNA damage and repair pathways. We agree that the measurements on phenotypic outcomes, such as cell proliferation, viability, or cell death assays, would provide important information to highlight the importance of the molecular mechanism discussed in this study.

We have previously measured the impact of IDH mutation on cell growth and sensitivity to temozolomide-induced DNA damage. IDH mutations lead to a slight decrease in cell proliferation and enhanced sensitivity to DNA damage. The data is replotted from Lu et al., 2017⁴.

We further conducted apoptosis analysis and colony formation in the patient-derived glioma stem cells with endogenous IDH mutation.

Further, we conducted colony formation assays using BT054 and BT142 cells. We discovered that TMZ-induced growth suppression could be partially reversed with α KG or TET1/2 overexpression. The new results have been included as Extended Data Fig. 7s, 7t, and 7u. The changes have been made to the manuscript on page 17, lines 446 – 449.

Finally, we evaluated the cellular apoptosis in BT054 and BT142 cells. We found that TMZ-induced apoptosis in these cells could be reversed with α KG or TET1/2 overexpression, which aligns with our finding in Extended Data Fig. 7n and 7o. The new findings have been included in Extended Data Fig. 7p to 7r. The changes have been made to the manuscript on page 17, lines 447 – 449.

6. Overall, the methods are common and standard methods in the field with adequate details provided. The methods section has adequate details, but more information should be added to the figures to assist the reader in their ability to understand the paper quickly.	We appreciate this comment. We have included more details in the methods to ensure clarity and reproducibility. The changes are updated in the method section from page 5 to 11.
6. The decision to investigate BRCA2 and not BRCA1 was not clearly explained. As BRCA1 is the dominant protein in this pathway, why was it not also investigated?	We appreciate the comment from the reviewer. While BRCA1 and BRCA2 are both involved in DNA repair pathways, BRCA1 is a pleiotropic DNA repair protein that is involved in both checkpoint activation, replication fork stabilization, and DNA repair, whereas BRCA2 is a dedicated mediator of the core mechanism of homologous recombination DNA repair. Several previous studies indicate the physical interaction among CTCF, BRCA2, and RAD51^{5,6}, which we found more relevant in our present study with a focus on the step of homologous recombination.
7. Figure 3G is one of the most interesting of the paper. As the authors show that γH2AX can be recruited, it seems that it is then not signaled to leave the chromatin. As many of the factors are known that regulate γH2AX movement off/on the chromatin, are any of these proteins inhibited by D-2-HG? As this would be one explanation for the data represented. Furthermore, total γH2AX should be included in the WBs as it would be good to know if total H2AX is being affected, or more H2AX is being phosphorylated and deposited onto the chromatin.	We appreciate the comment from the reviewer. The reviewer raised an interesting point regarding the other factors that regulate γH2A.X chromatin integration. γH2A.X, the phosphorylated form of H2A.X at Serine 139, is managed by DNA repair-related kinases such as ATM, ATR, and DNA-PK, as well as protein phosphatases. To better understand the dynamics between H2A.X phosphorylation, we evaluated the phosphorylation of H2A.X kinases ATM, ATR, and DNA-PK and discovered that D-2-HG resulted in elevated phosphorylation in these kinases, as well as elevated γH2A.X signal. The addition of kinase inhibitors reverses the phosphorylation and γH2A.X signal. We believe this result indicated that D-2-HG results in obstacles in the DNA repair pathway, which leads to the prolonged activation of DNA damage related kinases, and γH2A.X signal.

We also tested the phosphatase activity through *in vitro* experiments in the presence of D-2-HG. Our finding shows that D-2-HG does not obviously change phosphatase activity.

Overall, these evidence indicate that D-2-HG does not exhibit an obvious inhibitory effect on the phosphatases that modify H2A.X, whereas the D-2-HG is more likely to act through DNA repair machineries on the chromatin.

Minor comments:
 1. D-2-HG is used throughout the study as a treatment. The baseline levels of D-2-HG should be measured in the IDHmut models – as well as showing the decrease with the inhibitor in this model.

We appreciate the comment. We have measured the levels of D-2-HG in our DlvA cell line models. We confirmed that the D-2-HG level is substantially elevated in our cell line models. The levels of U251 IDH1^{R132H} have been reported in our previous work⁴.

2. The cell line model should be included in the figures panels. This would make the paper easier for the reader to understand as the main comparison is between IDH mut and IDH WT, clearly marking these would be helpful.

We appreciate the comment from the reviewer. We have included the cell line model information in the figure panels.

Reviewer 2

In general, I found this manuscript to be intriguing, with a novel and interesting mechanism to account for the changes in DSB repair efficiency linked to IDH mutations. The evidence that blocking 2-HG production reverses the phenotype is persuasive, and the work is consistent with previous literature showing HR defects in cells expressing IDH mutants.

The major weakness of the paper is in defining the defect in the DSB repair pathway. There are conflicting results on PARylation independent recruitment of CTCF to DNA damage sites. (Khalid Hilmi et al. *Sci. Adv.*3,e1601898(2017)) found that prolonged pre-treatment with PARP inhibitors did not prevent CTCF but did prevent BRCA2 and RAD52 from being efficiently recruited to a restriction enzyme-induced DSB. The 24 hour pre-treatment of the cells with PARP inhibitor complicates the interpretation of these results. However, the zinc fingers have been directly shown to bind PAR and are required to recruit CTCF to laser microirradiation sites (Han, D., Chen, Q., Shi, J. et al. CTCF participates in DNA damage response via poly(ADP-ribosyl)ation. *Sci Rep* 7, 43530 (2017).). Is the recruitment of BRCA2 and RAD51 still sensitive to PARP inhibition while CTCF remains recruited in this cell system?

We appreciate the summary and the comment from the reviewer.

We appreciate the comment. We postulate that, in general, appropriate epigenetic status at the DSB site is necessary for efficient DNA repair. CTCF recruitment to the DNA damage site may be facilitated by local chromatin signaling such as PARylation.

To better understand the role of PARylation in our model system, we pre-incubated the cells with PARP inhibitor Olaparib (Ola) and discovered that the recruitment of BRCA2 and RAD51, but not CTCF, was compromised upon DNA damage induction.

Besides, we also conducted PLA assay to evaluate the physical interaction among CTCF, BRCA2, and RAD51. We do notice that PARP inhibition leads to suppressed CTCF/BRCA2 and CTCF/RAD51 interaction, suggesting that the CTCF-guided recruitment of BRCA2/RAD51 require PARylation.

Based on these findings, it seems in our model that CTCF recruitment to the DSB site is less affected by PARylation, whereas the subsequent recruitment of BRCA2/RAD51 is affected by PARylation level.

Another study used IP-mass spectrometry to detect DNA repair proteins involved in HR that interacted with CTCF. They found MRE11 and CtIP, which are important for DNA end resection that commits cells to HR-mediated repair (Nucleic Acids Research, Volume 47, Issue 17, 26 September 2019, Pages 9160–9179). They also showed that PARylation of CTCF was needed for BRCA2 and RAD51 recruitment (Khalid Hilmi et al. Sci. Adv. 3, e1601898 (2017)). Additionally, MDC1 and Rad51 were reported to bind to CTCF (PNAS 114:10912-7). This creates a lot of uncertainty about where the problem in DSB repair happens. Regarding the defect in homologous recombination repair, the impact on DNA end resection (re: MRE11/CtIP interaction) should be examined. A problem in end resection would prevent all later steps and explain the loss of BRCA2 and RAD51 from DSB sites in the IDH mutant cells.

We appreciate the comment from the reviewer. We agree that examining the efficacy of DNA end resection by Mre11/CtIP could be important to understand the impact of D-2-HG on the DNA damage and repair process. First, we conducted co-immunoprecipitation experiments to understand the interaction between Mre11 and CtIP in the context of IDH mutation. Our finding suggests that TMZ exposure resulted in enhanced physical interaction between Mre11 and CtIP, whereas this interaction could be compromised with the presence of D-2-HG. Based on this data, we postulated that D-2-HG impacts the DNA repair pathway through several different levels, ranging from the assembly of end resection complex, DNA topology changes, and HR repair.

CTCF has been previously shown to play a critical role in DNA end resection (Nucleic Acids Research, Hwang et al., 2019). Specifically, CTCF facilitates the recruitment of CtIP to double-strand break

(DSB) sites and interacts with Mre11 through its N-terminal domain. CTCF retention at DSB sites relies on the zinc finger domain, which binds directly to DNA. Therefore, DNA hypermethylation that disrupts CTCF binding can impair the end resection process, potentially serving as an upstream event that compromises homologous recombination repair.

The new data has been included as Extended Data Fig. 6f to 6g. The changes have been made to the manuscript on page 16, lines 430 – 433.

Further, we measured the end resection efficiency in D1vA cells through a previously reported method⁷. We observed compromised end resection efficiency in both DSB sites investigated, which aligns with the finding above.

The new data has been included as Extended Data Fig. 6d to 6e. The changes have been made to the manuscript on page 16, lines 428 – 429.

How should we understand the impact of DNA methylation status on CTCF binding? Does this mean that CTCF only binds to DNA sequences that are specific to CTCF? Regarding the *Isce1* cut site, can the variation in binding be attributed to methylation of certain DNA sequences? Are there any CTCF consensus sites close to the cut site that could account for the CTCF binding? The correlation is evident, but the exact mechanism is not clear. Similarly, is it only the native CTCF binding sites that are lost around the *ASIS1* cut sites or is there a more non-specific DNA binding that is sensitive to methylation status?

We appreciate the comment. Several pioneering studies indicate that CTCF binding is affected by the methylation status in the genome, which is largely through the CTCF-binding site². CpG islands in or next to the CTCF binding site could affect CTCF affinity based on their methylation status. In the present study, we believe that D-2-HG may impede the binding of CTCF by affecting the CpG island methylation status. We confirmed the substantially elevated CpG island methylation status next to the CTCF peaks at the *Asis 1* DSB sites. The result can be found in Fig. 5C and Extended Data Fig. 8i.

Further, we performed a Motif analysis for the CTCF peaks obtained from DivA cells that are next to the *Asis 1* DSB sites, to explore the consensus sequence of the binding sites. It seems most of the CTCF binding aligns with the CTCF-binding motif.

Homer de novo Motif Results (Output_Lostpeaks_treated//)

Known Motif Enrichment Results
Gene Ontology Enrichment Results

If Homer is having trouble matching a motif to a known motif, try copy/pasting the matrix file into STAMP
 More information on motif finding results: HOMER | Description of Results | Tips
 Total target sequences = 4219
 Total background sequences = 4558
 * = possible false positive

Rank	Motif	P-value	log P-value	% of Targets	% of Background	STD(B) STD(D)	Best Match/Details	Motif File
1	CCACTAGGGGGC	1e-779	-1.794e+03	34.56%	5.00%	38.2bp (57.6bp)	J0RUS(Z)K562-CTCF-ChIP-Seq(GSE32465)/Homer(0.927) More Information Similar Motifs Found	motif file (matrix)
2	CAAGCCAC	1e-116	-2.692e+02	19.58%	8.27%	47.9bp (58.6bp)	CTCF/MA1102.2/Jaspar(0.747) More Information Similar Motifs Found	motif file (matrix)
3	AAACCAGCCACC	1e-27	-6.369e+01	0.36%	0.00%	42.5bp (0.0bp)	G12(Z)GM2-G12-ChIP-Seq(GSE112702)/Homer(0.625) More Information Similar Motifs Found	motif file (matrix)
4	ATGGGCCAGAA	1e-23	-5.358e+01	0.31%	0.00%	45.8bp (0.0bp)	Pfaff1/MA1615.1/Jaspar(0.658) More Information Similar Motifs Found	motif file (matrix)
5	CTACCCATT	1e-21	-4.863e+01	0.28%	0.00%	53.0bp (0.0bp)	PR0128.1_Gcm1.2/Jaspar(0.679) More Information Similar Motifs Found	motif file (matrix)

The data in Figure 5A raise a similar question about the origin and mechanism of the repair defect. The paper does not mention 53BP1, which is shown in the image series and seems to have a loading deficiency as well. This could indicate a problem upstream of the interactions that this study examines and should be further investigated. If both 53BP1 and BRCA1 recruitment are impaired,

We appreciate the comment. We monitored 53BP1 levels to monitor the DNA damage response. Our preliminary finding indicates that its concomitant NHEJ pathway is unchanged upon the exposure of D-2-HG. The NHEJ efficiency has been measured in the context of IDH mutants or D-2-HG previously⁸. Sulkowski et al. discovered that the NHEJ efficacy is minimally changed in two cell lines with IDH

this would suggest that the main defect is upstream in the signaling pathway (MRE11, CtIP, MDC1 are all possible upstream interactions where CTCF could be involved based on its known direct binding to these upstream proteins) rather than at the level of BRCA2 and RAD51 loading. The limitation is that only single cells are shown and individual cells within the same population can vary a lot depending on cell cycle position. This uncertainty about 53BP1 recruitment also emphasizes the fact that only an HR reporter system was used. The authors should also evaluate the efficiency of NHEJ in IDH mutants.

mutation, whereas a significant loss of HR activity was seen in the same models.

As the reviewer instructed, we measured the NHEJ efficiency through EJ5 and EJ7 reporter systems in our laboratory. Both of the reporter systems indicate that NHEJ is minimally affected by D-2-HG.

The new results are included as Extended Data Fig. 1c and 1d. The changes have been made to the manuscript on page 12, lines 292 – 295.

What is the interpretation of a greater number of H2AX foci in the IDH mutants? How is more damage created with the same amount of DNA damage? Does this reflect differences in the kinetics of the repair or differences in the kinetics of assembly of H2AX foci? A kinetic analysis of H2AX foci appearance and disappearance would address this uncertainty.

We appreciate the comment. We performed a time course IF staining to understand the kinetics of γ H2A.X foci in DivA cells upon 4-OHT-induced DNA damage. We discovered that in both IDH1^{WT} and IDH1^{R132H} cells, γ H2A.X foci appear in 2 - 4 hr after 4-OHT treatment. The γ H2A.X foci were reduced in wild-type cells 12 - 24 hr after induction, whereas the foci kept accumulating in mutant cells and the diminish of γ H2A.X foci appeared to be very slow. We believe this difference in kinetics is primarily due to inefficient DNA repair pathways in IDH mutant cells.

The new result is included as Extended Data Fig. 1a and 1b. The modifications have been made to the manuscript on page 12, lines 291 – 292.

Cell cycle analysis should be presented to clarify how the mutants and rescued cells compare in cell cycle distribution. Presumably, activation of checkpoints in the IDH mutant cells will alter cell cycle distribution to some extent. Cell cycle directly influences DNA repair pathway utilization and so this needs to be excluded as the basis for any observed differences in repair properties.

We appreciate the comment from the reviewer. In our previous finding, IDH mutant enzyme does not induce substantial changes in cell cycle distribution in untreated cells, whereas a significantly elevated G2/M population could be observed under TMZ exposure⁹.

We performed a more dedicated EdU/PI flow cytometry analysis in the cell line models used in the present study. We confirmed similar cell cycle distribution between IDH1^{WT} and IDH1^{R132H} cells, with slightly decreased EdU incorporation in mutant cells.

	The elevated steady-state DNA damage/H2AX foci in the absence of damage is important and should be investigated further. There could be a number of explanations. If the cells are undergoing greater levels of replication stress, as they report, they may both be enriched in S-phase cells, which also normally have the highest incidence of DSB foci and cells with multiple DSB foci, and have higher numbers of foci specifically in S-phase cells. Post-mitotic DNA damage results in the formation of unusually large DSB foci that are typically only one or a few per nucleus and would reflect differences in DSB repair or checkpoint function at different stages in the cell cycle to explain the observed differences.	The reviewer raised a brilliant point about the replication stress. Schwartzman et al., reported recently that IDH-mutated cells exhibit heterochromatin-derived replication stress¹⁰. The replication stress is associated with elevated γH2A.X signal in mutant cells. The decreased EdU incorporation shown in the cell cycle flow cytometry above also indicates the presence of replication stress in IDH1^{R132H} cells. We are actively pursuing the investigation of the replication stress at this point and will present the findings through another dedicated manuscript.
The line scans are unnecessary in Figure 2e are unnecessary. The more important information present in the figure would be the percent of the TMZ-treated H2AX foci that are positive for RAD51. This should be quantified and added if the manuscript is resubmitted.	We appreciate the comment. We hope to use the line profile to highlight the spatial colocalization between molecules. As the reviewer suggested, we performed image analysis to highlight the colocalization of γH2A.X foci and RAD51. The data is in Extended Data Fig. 3a and 3b. The changes have been made to the manuscript on page 13, lines 329 – 332.
While the PLA assay supports the claim that there are direct interactions between CTCF and RAD51/BRCA2, the evidence in the literature for a direct interaction is limited. CTCF interacts with BRCA2 in a PARylation-dependent and Zinc finger-dependent manner. As zinc fingers are common PAR binding motifs and this interaction requires both PARylation and CTCF zinc fingers, this could reflect the association of both BRCA2 and CTCF with PAR polymers released as large complexes upon cell lysis. The PLA assay does not distinguish between these possibilities because DNA	We appreciate the comment from the reviewer. Regarding the interaction among CTCF, RAD51, and BRCA2, besides the PLA assay, our co-immunoprecipitation findings demonstrate the physical interactions among these DNA repair factors. We agree with the reviewer that using a control condition could be helpful to rule out the possibility of local high concentration.

damage sites are regions that are concentrated in all of these factors. All that it convincingly shows is that all three proteins are recruited to DNA damage sites. If a control showing that gammaH2AX or 53BP1 does not show PLA foci at double-strand breaks, it would be compelling that the method can discriminate between direct interactions and locally high concentrations. Optimally, with around 10-20 nm distances between antibodies, it's unclear that high local concentrations won't generate the same robust signal.

53BP1 is known having minimal interaction with CTCF or BRCA2, so we could perform PLA with antibodies against CTCF/BRCA2 as the control to solidify the conclusion from PLA results.

We discovered that the PLA signal is very low when using antibodies against RAD51 and 53BP1, suggesting that DNA repair proteins with different pathways should not give PLA signals, and thus distinguish direct interactions and locally high concentration. In contrast, the BRCA2/CTCF and RAD51/CTCF antibody combination gives potent PLA signals at the same assay, as shown below and in Figure 4. The inclusion of PARP inhibitor Olaparib (Ola) greatly reduces the BRCA2/CTCF and RAD51/CTCF foci. It seems in our model system, the PARylation is necessary for the assembly of BRCA2/RAD51/CTCF complex.

An additional validation step necessary to confirm that CTCF interactions are shared with those reported in the literature is to test the sensitivity of BRCA2 and RAD51 association with CTCF in the PLA assay. They are reported to be sensitive to PARP inhibition and mediated by PARylation. The authors should consider Figures 2 E, F; 3A-H; Because there is variation across the cell cycle with respect to DNA repair pathway utilization, it is not appropriate to show single cells to illustrate differences. Rather, an image showing cells representing different stages of the cell cycle is required to appropriately compare to samples.

We appreciate the comment from the reviewer. For the morphology analysis, we took images from multiple cells (> 60 per condition) through several biological replicates to rule out the influence of the cell cycle. We presented a single cell to provide clear visualization of the foci and protein colocalizations in Figures 2 E, F. We have updated Figure 4A-H by replacing the images for the PLA foci number assays with ones that display multiple cells.

Reviewer 3

We appreciate the input from the reviewer.

Reviewer 4

The manuscript presented by Lang et al entitled “*D-2-Hydroxyglutarate establishes BRCAness through epigenetic reprogramming*” explores the molecular mechanism responsible for the diminished DNA repair ability of IDH mutant malignancies, and, therefore, sensitivity to therapeutic targeting by DNA repair inhibitors. The authors reproduce these preliminary findings in their model system and demonstrate that the diminished DNA repair observed in IDH mutant cells is dependent on the mutant IDH neomorphic activity that results in production of D-2-hydroxyglutarate (D-2-HG), a known potent inhibitor of α -ketoglutarate dependent demethylating enzymes (including those acting on DNA, histone, and RNA). As exogenous addition of α -ketoglutarate can rescue these defects in DNA damage repair of IDH mutant cells, the authors go on to specifically evaluate a role for the DNA demethylating enzymes, TETs. They demonstrate overexpression of TET1 or TET2 can partially rescue the defect in DNA damage repair caused by D-2-HG production, suggesting a role for TETs in DNA damage repair. This role is additionally supported by the observations that TET1 or TET2 deficient cells also exhibit defects in DNA damage repair, and that sites of DNA damage accumulate the TET enzymatic product, 5hmC. Moreover, they show that local DNA damage-induced recruitment of known proteins associated with DNA repair by homologous recombination (including CTCF, BRCA2, and RAD51) is TET-dependent. Lastly, the authors explore whether hypermethylation-dependent loss of CTCF alters high-order chromatin structure changes in IDH mutant cells that are associated with double stranded breaks. Together these findings demonstrate a clear role for TET enzymes in the repair of DNA damage by homologous recombination. This has been previously suggested based on the reported co-localization of 5hmC with α H2AX at sites of DNA damage (Kafer, G.R., et al. Cell Rep 2016), and the TET catalytic-dependent regulation of DNA damage repair in response to phosphorylation by the DNA damage sensors ATR/ATM (Jiang, D.,

We appreciate the comments from the reviewer. The valuable insights greatly improve our work from its current form. We are happy to address the concerns from the reviewer and perform additional experiment for more evidence.

et al. Brain, 2015 & Jiang, D., et al. EMBO Reports, 2017). The authors are ultimately able to corroborate and validate these previously reported findings using multiple models of DNA damage (Genotoxic stress, Aphidicolin, DRGFP reporter system, and DlvA system). In addition, the authors are able to extend this observation to a new biological setting by demonstrating the loss of TET-dependent regulation of DNA damage in IDH mutant cells contributes to the mechanistic basis of their defects in DNA damage repair. Moreover, the authors are able to begin to unravel mechanistically how TET proteins contribute to DNA damage repair by aiding in the recruitment and assembly of DNA damage repair machinery (CTCF, BRCA2, and Rad51), which has not previously been reported to our knowledge. In total, this work advances our understanding of the contribution of TETs to DNA damage repair and establishes a novel molecular mechanism that underlies IDH mutant malignancies.

1. Is TET-dependent regulation of DNA damage repair the primary mechanism lost in IDH mutant cells?

The foundational premise of the manuscript is based on the observation that the defect in DNA damage repair in IDH mutant cells is due to D-2-HG production, which inhibits α -ketoglutarate demethylating enzymes, and that this defect can be rescued by exogenous addition of α -ketoglutarate. The authors explore the contribution of the TET DNA demethylating proteins to this defect by individual reconstitution of TET1 or TET2 by overexpression. While this provides some rescue over treatment with D-2-HG alone, the extent of rescue is variable between experiments (1f, 1j, 2c, 2d, 3b, 3e, 3f, 3g) and based on the figures does not always reach the level of rescue achieved with exogenous addition of α -ketoglutarate. As the text implies that D-2-HG-dependent defects in DNA damage repair are TET-dependent, it would be important to address this discrepancy in the ability of either TET1 or TET2 to rescue this defect compared to α -ketoglutarate. Is this discrepancy due to inefficient expression of TET proteins, the need to have both TET1 and TET2 to achieve full rescue or the involvement of other pathways in D-2-HG-dependent inhibition of DNA damage?

As previous mechanisms have been reported to contribute to the D-2-HG-dependent inhibition of DNA damage repair (Sulkowski, P.L.,

We appreciate the insight from the reviewer. We agree that a more in-depth investigation on the dependency on TET1 and TET2 could be more convincing. As the reviewer correctly point out, the D-2-HG dependent DNA repair deficiency may be derived from multiple paralleled mechanisms. Sulkowski et al. discussed the involvement of H3K9me3 and DNA repair deficiency, which could be a parallel mechanism with our proposed theory.

Through the revision, we detected the rescue effects of α KG on DDR in the absence of TET1 and TET2 in U251 cells. We discovered that α KG resulted in less DDR with no TET1 or TET2 in the cell, although to a less extent in TET1/2 proficient cells. These results suggest that several mechanisms may be involved in D-2-HG-associated DNA damage sensitivity, including TET1/2-dependent and TET1/2-independent pathways.

et al. Nature 2020, Wang, P., et al. Cell Reports, 2015), it would be important to address, experimentally, whether rescue relies on restoration of both TETs or a contribution of these other pathways. This is especially important as in the discussion the authors describe the mechanism identified in this manuscript to be a “parallel” mechanism to that described within Sulkowski, P.L., et al (Line 429).a. While simultaneous reconstitution of both TET1 and TET2 would be functionally difficult to achieve to assess rescue, the authors should do the dual TET1 and TET2 knockout within their experimental model system of TMZ treatment with D-2-HG and assess whether this double knockout completely abrogates the ability of α -ketoglutarate to rescue the defect in DNA damage repair. Additionally, they could assess aspects of the histone demethylase mechanism described by Sulkowski et al and determine if this mechanism is active in their model system. This could be achieved by evaluating H3K9me3 deposition around DNA breaks in experiments similar to those in Figure 2h and 3g.

Further, we validated the involvement of H3K9me3 methylation in Sce-I DRGFP U2OS cells. We recorded a rapid transient increase in H3K9me3 modification upon DSB induction, whereas the H3K9me3 level was maintained at a high level in cells exposed to D-2-HG. This aligns with the findings shown in the previous study⁸. Importantly, α KG, but not TET1 or TET2 overexpression resulted in the rescue of the H3K9 coverage, suggesting the involvement of H3K9me3 guided DNA repair mechanism.

The new result has been included as Extended Data Fig. 8a and 8b. The changes have been made to the manuscript on page 17, lines 452 – 456.

	Overall, our findings suggest the presence of paralleled DNA repair mechanisms that are affected by D-2-HG and epigenetic reprogramming.
b. These experiments would provide clarity to whether the TET-dependent mechanism is the sole mechanism active within this model of IDH mutant malignancy as is currently implied. Should inhibition of TET function be the sole mechanism responsible for the observed defects in DNA damage repair in this system in response to D-2-HG, the authors should discuss 1) the relative differences between their model of IDH mutant malignancy and the other previously described models, 2) why these different IDH mutant models rely on distinct molecular mechanisms for D-2-HG disruption of DNA damage repair, and 3) what these differences mean with respect to understanding IDH mutant defects in DNA repair. If, however, both mechanisms are active in this model system, then the presence of this other contributing mechanism should be more clearly indicated in the results.	We appreciate the comment from the reviewer. With the new findings above, we incline to believe there are multiple paralleled mechanisms to establish DNA repair deficiency in IDH-mutated cells. With more data available throughout the revision, we could be more confident to address this claim.
2. Clarity and interpretation of figure 5 and extended data figure 5. As they are currently presented, these figures have limited details present in the legend/figures that make technical and biological interpretation difficult and raise additional questions that require addressing/clarification.	We apologize the limited details presented in the legend and figures. We have included more information as indicated by the reviewer. The new content could be found in the method section on page 5 – 11, as well as the updated figure legends.
a. Extended data figure 5 shows the 4C signal and CTCF ChIP for two selected double stranded breaks. This figure could benefit from the addition of a distance scale to give the reader some indication of the physical spread of contacts and CTCF binding sites in the WT and IDH mutant cells at baseline and in response to DNA damage.	We appreciate the comment from the reviewer. We have included a better ruler with annotations on Asis I sites and viewpoints used for 4C-seq analysis. The new panel is included as Extended Data Fig. 8h. The modifications have been made to the manuscript on page 18, lines 479 – 480.

i. As it is presented, the CTCF ChIP peaks appear to be from untreated WT and IDH mutant cells. Is this correct? Has the CTCF peak status been assessed in the 4OHT treated cells to specifically evaluate changes in CTCF localization around the potential sites of breaks in response to damage? While the authors show the 4-OHT treatment and CTCF peaks in Figure 5d, this is for bulk localization and does not show individual CTCF peaks around the potential and actual break sites.	We appreciate the comment from the reviewer. The CTCF peaks in the previous version are from non-treated cells. The line profiles and heatmaps show the CTCF peaks around 80 of Asis I sites. As the reviewer recommended, we have updated the CTCF peaks before and after 4-OHT treatment in the revised Figure 5a and 5b for better clarity. In addition, we have included ChIP-seq tracks to better illustrate CTCF peaks next to the DSB sites. The data is also aligned with the 4C-seq signals. Our finding highlights the decreased CTCF coverage in IDH1^{R132H} cells, regardless of the presence of DSB. The decrease of CTCF coverage aligns with the lack of 4C signal changes, suggesting the compromised conformation adjustment in the DSB sites.

The new results have been included as Figure 5a and 5b. The changes have been made to the manuscript on page 17, lines 465 – 466.

b. Figure 5e is bisulfite sequencing for IDH WT and mutant cells. Additional information in the legend would aid readers unfamiliar with this technique in interpreting this analysis. What do dark and light circles indicate? What are the column and row numbers indicating? Additionally, are the presented results for the cells with or without 4-OHT?

We appreciate the comment. Each column indicates one CpG island that is close to a CTCF peak next to the DNA damage sites. As the bisulfite reaction leads to mixed genotypes in the CpG islands, we performed Sanger sequencings on multiple clones (row) from each site and calculated the incidence of methylated CpG islands.

We have included figure legends in the data legend for better clarity. We have also included both untreated and treated cells in the updated Figure 5c.

i. The authors should include the sequence of the selected CTCF peaks for this methylation analysis. In this sequence they should indicate the potential sites of methylation and the predicted/actual CTCF binding site to aid the readers in determining whether there is specifically disruption of CTCF binding based on increased methylation within the CTCF consensus sequence as the authors hypothesized.

We appreciate the comment. We have included the genomic regions selected for methylation analysis in supplementary data.

ii. This figure aims to support the connection between methylation status and CTCF binding across the genome impacting high-order chromatin structure. However, the presented data is evaluating CpG methylation status for one author-selected CTCF peak (indicated in extended data figure 5) flanking a double stranded break for two double stranded breaks in a model system that has ~100 breaks. Based on this limited analysis, it is difficult to ascertain whether these results are largely representative of the relationship between DNA hypermethylation and CTCF binding around potential sites of double stranded breaks. Additionally, as it is presented, it is unclear as to whether the observed hypermethylation is at baseline or in response to a double stranded break in the WT and IDH mutant cells making it difficult to interpret the biological meaning of these results. The authors should expand their analysis to further substantiate the claims that DNA hypermethylation is preventing CTCF binding and therefore higher-order chromatin structures.

We appreciate the comment from the reviewer. The DNA hypermethylation preventing CTCF binding has been previously demonstrated¹¹. To enhance the present study, we have selected 4 additional CTCF/CpG sites to confirm the changes in DNA methylation on CTCF binding sites. We have also included both untreated and treated cells for the methylation analysis.

The new results are included as Extended Data Fig. 8i. The changes have been made to the manuscript on page 18, lines 480 – 482.

Further, we evaluated the association of CTCF to the chromatin upon DSB establishment and TET1 activity. We discovered that CTCF chromatin association is decreased in IDH1^{R132H} cells, whereas ectopic expression of the TET1 catalytic domain (CD) rescues the CTCF chromatin association. We believe this finding suggests that IDH1 mutation and D-2-HG compromise CTCF

chromatin binding through a TET-dependent manner, presumably by affecting DNA methylation status throughout the genome.

The new findings have been included in Extended Data Fig. 8g. The changes have been made to the manuscript on page 18, lines 475 – 476.

iii. To that end, does the methylation status in this experiment change with or without 4OHT treatment?

As discussed above, the DNA methylation analysis on six different genomic loci indicates that methylation status is determined by the presence of IDH1 mutant, but less on 4-OHT treatment.

c. Figure 5f/g. There is no scale on the y-axis to discern the relative peak size across treatment groups. What do the asterisks and arrow in the figure indicate? This could also benefit from a genomic distance scale. Is f the data for the break one and g for the break two from figure 5e? d. Figure 5h/i. What is the difference between the left and right plots for figure h/i and what is the relationship of these plots to f and g? How was the 4C signal quantitated? Is this taking all signal from f/g or simply the yellow highlighted sections?

We appreciate the comment from the reviewer. We have included more details for Figures 5f and 5g, which are currently assigned to Figure 5b. The 4C signal is quantified based on the normalized read depth from the ectopic peaks. The changes in 4C signal are calculated with $\text{Log}_2[(+DSB/-DSB)]$ ratio. To better illustrate the changes of 4C signal throughout the DSB loci, we plotted the 4C ratio data track using a previous method¹². Our finding reveals elevated 4C signal at the DSB sites within a spanning of several million base pairs, which is likely within DSB-associated TAD. The 4C signal change is less observed in IDH1^{R132H} cells.

	B Chr1 (DSB1) Chr20 (DSB2) 80 AsisI_90 100 25 30 AsisI 35 40 4C-seq(+DSB/-DSB) Viewpoint IDH1^{WT} IDH1^{R132H} 10 -DSB +DSB CpG CTCF enrichment CpG 10 0 -DSB +DSB IDH1^{WT} IDH1^{R132H}
3. Experimental design, statistics, materials and methods. a. As the paper is currently written there is no information regarding the number of independent experiments that have been completed for any of the experiments/figures. While the methods denote that the presented data are the mean +/- SEM there is no indication for individual figures as to the number of independent experiments that have been performed and whether the presented data are representative of independent experiments or a compiled analysis of all independent experiments.	We have included information regarding the number of biological replications (independent experiments) that have been performed. The information is included in the method section and updated figure legends.
b. The statistical section of the methods simply says t-tests and one-way ANOVA were used. Interpretation of the analysis would be improved by providing the individual test used for each analysis within the figure legends. Additionally, the presentation of statistical measurements across figures is variable and for some figures it is difficult to ascertain what the statistical comparison being presented	We appreciate the comment from the reviewer. We have run through the manuscript to ensure consistency in statistics and the presentation.

is. For example: 1f uses lines between compared groups, 1k uses brackets, 3b uses asterisks without a line such that the comparison is unclear. A unified presentation of the statistical analysis would make it easier for the reader to interpret.																			
i. The methods state statistical significance is set as less than 0.05. However, figures use both * and ** to indicate significance. What is the difference between these significance indications?	We have included explanations of * and ** in the method section and figure legends.																		
ii. For figure 1K is the right bracket indicating that all of the 5 right groups are statistically significantly increased compared to the third group? The text seems to imply this is what is meant.	We appreciate the comment. We have updated the format in Figure 1K with better clarity.   <caption>Data for Figure 1K (GFP+ %)</caption>   Condition GFP+ (%)     Scp-1 ~1   Scp-1 + ~9.5   Scp-1 +, D-2-HG + ~3   Scp-1 +, D-2-HG +, alpha-KG + ~8   Scp-1 +, D-2-HG +, alpha-KG +, TET1 + ~5.5   Scp-1 +, D-2-HG +, alpha-KG +, TET2 + ~7   Scp-1 +, D-2-HG +, TET1 +, TET2 + ~5.5   Scp-1 +, D-2-HG +, TET2 + ~7   	Condition	GFP+ (%)	Scp-1	~1	Scp-1 +	~9.5	Scp-1 +, D-2-HG +	~3	Scp-1 +, D-2-HG +, alpha-KG +	~8	Scp-1 +, D-2-HG +, alpha-KG +, TET1 +	~5.5	Scp-1 +, D-2-HG +, alpha-KG +, TET2 +	~7	Scp-1 +, D-2-HG +, TET1 +, TET2 +	~5.5	Scp-1 +, D-2-HG +, TET2 +	~7
Condition	GFP+ (%)																		
Scp-1	~1																		
Scp-1 +	~9.5																		
Scp-1 +, D-2-HG +	~3																		
Scp-1 +, D-2-HG +, alpha-KG +	~8																		
Scp-1 +, D-2-HG +, alpha-KG +, TET1 +	~5.5																		
Scp-1 +, D-2-HG +, alpha-KG +, TET2 +	~7																		
Scp-1 +, D-2-HG +, TET1 +, TET2 +	~5.5																		
Scp-1 +, D-2-HG +, TET2 +	~7																		
iii. For figure 2C/D and 3E/F is the right line indicating the fifth group is statistically different from the second or is it that groups 3, 4, and 5 are statistically significantly increased compared to the second group? The text implies all three are able to rescue but the way that it is denoted by line would traditionally only indicate the two connected groups by the line are significantly different.	We have reformatted Figures 2C, 2D, 3E, and 3F for better clarity.																		
iv. For figure 3b what is being compared for the presented statistical analysis?	We have reformatted Figure 3B for better clarity.																		
v. Clarification of these statistical analyses will be important for ensuring proper analysis and biological interpretation of the experiments.	We appreciate this comment and have revised the manuscript for the details of statistical analyses.																		
c. The methods, as they are currently written, are not sufficiently detailed to allow for reproduction of all experimental procedures and some methods need to be added including: 5hmc immunofluorescence, siRNA experiments for knockdown, how the	We have included details in the experiment procedures including 5hmc immunofluorescence, RNA interference for knockdown, CRISPR-mediated TET1/2 KO, the 4C-seq data analysis, and primers used for ChIP with the DR-GFP reporter system. The																		

knockouts were generated for TET1 and TET2, the 4C data analysis and quantification, and primers used for ChIP with the DR-GFP reporter system.	updated details can be found in the method section on page 5 – 11, and in supplementary files.
i. Additionally, it would dramatically aid readers to have a more detailed explanation of the individual components of the DR-GFP reporter and DiVA systems to improve their understanding of these model systems. To that end, figures diagramming the general basis of these model systems would dramatically aid the reader in easily understanding their utility.	We have included details in the experiment procedures and composed a schematic illustration of the DRGFP system. Please find the information in the method section and at the bottom of Figure 2h. More details in the DD-Sce-I-GR DRGFP U2Os system could be found in the method section on page 5, lines 99 – 112.
4. Conceptual contribution of Figure 5, and inconsistent/overstatement of scientific claims. a. In its current form figures 1-4 build to a model of TET-dependent regulation of DNA damage repair at the site of DNA damage that cohesively fit together. Figure 5, however, begins to explore the contribution of higher-order chromatin structures/TADs to the regulation of DNA damage and whether these are correlated with changes in methylation status/CTCF binding. While tangentially related, figure five feels like the first figure of a new paper focused on exploring the contribution of these structures to DNA damage repair, not the conclusion of the first four figures, especially considering major comment number 2. The impact of the presented work would benefit from further substantiating the claims in figures 1-4 and saving figure 5 for a future work where the contribution of these higher-order structures could be further evaluated and substantiated.	We appreciate the comment from the reviewer. Indeed, Figure 5 describes our discoveries that are a step further from the mechanistic study based on TET/CTCF. We feel strongly that the TAD and DNA topology provide an important explanation of how CTCF guides the DNA repair pathway, which is a key step in D-2-HG-associated BRCAness. We have performed additional experiments to strengthen the link among TET, CTCF, and chromatin conformation in the revised manuscript.
i. Along these lines there seems to be competing presentation of two mechanistic bases for the TET-dependent regulation of DNA damage repair within the text: 1) methylation dependent changes in chromatin organization/structure that impairs DNA damage repair by changing overall CTCF localization, 2) methylation-dependent prevention of CTCF binding based on DNA sequence methylation and recruitment to local sites of damage. It is possible that both mechanisms contribute to DNA damage repair; however, the respective contributions are not immediately clear based upon the currently presented data. b. To this end, as it is currently written, the abstract is not representative of the findings and to some extent overstates the presented work and its conclusions. The synopsis of the work provided at the end of the introduction appears to be a more	We appreciate the comment from the reviewer. The reviewer raised an important point regarding how CTCF is involved in DNA repair. We believe the involvement of CTCF is associated with DSB repair through multiple dimensions, ranging from defining the DSB-associated TAD boundary to facilitating the recruitment of DDR factors. Many of these processes may be sensitive to local chromatin methylation status. The CTCF-guided DDR factor recruitment has been reported previously^{5,6}. Our study extended these seminal studies by linking CTCF with chromatin topological adjustment, a phenomenon also guided by CTCF. In Figure 3g and Extended Data Fig. 8g, we showed that ectopic expression of TET1/TET2 rescued CTCF recruitment, which

accurate reflection of the presented work. This should be addressed by adapting the abstract to be more reflective of the included findings or performing additional experiments to support these statements. As it is currently written, the abstract reads as though the CTCF high-order chromatin structure is directed by TETs and the loss of these structures forms the basis for failed DNA repair in IDH mutant cells. However, this manuscript, in its current form, does not establish these connections, but instead reveals a local TET-dependent recruitment of DNA repair proteins at sites of DNA damage (based on the local 5hmC deposition by IF and MeDIP, within a range of ~2000 bp around the cut site). Additionally, the work presents evidence to suggest IDH mutant cells also exhibit diminished remodeling of high-order chromatin structures (50kb-1Mb around cut site) in response to breaks (however with some caveat based on major comment 2), but the relationship of these structures to TET function and the contribution of these structures to DNA damage repair remain to be fully elucidated. If the authors wish to include additional experiments to support these statements, they would need to demonstrate the reliance of these damage induced contacts on TETs and CTCF potentially using the rescue by TETs and CTCF knockdowns used to establish the local relationship at sites of damage. Moreover, the contribution of these more distant contacts to the repair process would need to be assessed.

supports the claim that DNA methylation status would affect CTCF chromatin association.

Further, we conducted an additional 4C-seq experiment to understand the connection among CTCF, TET catalytic function, and high-order chromatin structure.

We found that CTCF depletion through RNA interference resulted in substantial changes in the 4C signal that is associated with DSB induction. The changes of the 4C signal were transformed into $\log_2(\text{siRNA}/\text{Control RNA})$. Loss of chromatin contact will be shown as negative values in the plot. This finding suggests that CTCF is necessary to modulate the TADs that are associated with DDR.

Further, we evaluated the connection between TET catalytic activity and DSB-associated DNA topological adjustment. To this end, we reached out to Dr. Anjana Rao's group and obtained the TET1 catalytic domain (CD) constructs. The 4C-seq analysis on these cell lines revealed that TET1 catalytic activity substantially improved the chromatin interaction upon DSB induction compared to the mutant TET1 dCD, suggesting the cytosine methylation status is closely connected to the organization of DSB-associated TAD. The 4C-seq data is transformed as $\log_2(\text{TET CD} / \text{TET dCD})$. Increased chromatin contact is shown in positive value in the graph.

These new findings are included in Extended Data Fig. 8e and f. The changes have been made to the manuscript on page 17, lines 467 – 470.

Overall, our additional findings strongly indicate the cytosine methylation status at the DSB sites affects the organization of DNA repair-associated TAD, probably in a CTCF-dependent manner.

c. Line 432. *“Overall, our findings reveal the molecular mechanisms of metabolic and epigenetic predisposition on “BRCAness” in malignancies.”* This is a strong statement that overstates the applicability of the work in its current form. While this work does reveal a novel contribution of TET-dependent regulation of the DNA damage response to IDH mutant malignancy, the presented work is largely confined to one mutant condition in one cell type and therefore statements of this broad utility to malignancy in total seem inaccurate. Additionally, considering other mechanisms have been proposed (see major comment 1) which are not assessed in this work, this mechanism, based on the data available at present, seems unlikely to be the sole basis of “BRCAness in malignancy”

We appreciate the comment from the reviewer. We have adjusted the wording for better accuracy. The changes have been made to the manuscript on page 20, lines 537 – 538.

5. Additional experimentation/work to bolster findings.

a. Given the 5hmC is co-localizing to these sites of DNA damage, it appears that catalytic activity of TETs is required for this process, which is further supported by the rescue with TET1 or TET2. These findings would be significantly bolstered by performing these rescue experiments with a catalytically dead (HxD) mutant to further confirm that local TET activity at the time of damage is required for this response.

b. The time-dependent establishment of γ H2AX foci and the subsequent recruitment of CTCF, BRCA2, and RAD51 to sites of DNA damage in figure 3g very nicely contribute to the mechanistic basis of TET-dependent regulation of this response. Figure 3b connects these findings with the local deposition of 5hmC (and 5mC in Extended Figure 3c) using the same model system, although it is unclear at what time point in the response this information is collected? Additionally, is this performed only using GFP+ cells in this reporter system? The manuscript would benefit by performing a similar time course as experiment 3g for the 5hmC and 5mC to show the local TET-dependent regulation of DNA methylation in response to breaks. This would also reveal if TETs are acting simply to deplete existing 5mC at these sites.

c. Addition of a model to collectively describe the included findings and their contribution

6. Experimental concerns.

a. Figure 1H: In the TET1 western blot image, why are lanes 2 and 3 so white? Even the background greying shows loss in this area.

We appreciate the comment from the reviewer. As shown above in point 4i, we have included TET1 CD and TET1 dCD construct to confirm the role of TET activity in DNA repair and chromatin topological adjustment.

We appreciate the comment from the reviewer. In Figure 3b, the DSB is induced by 0.5 μ M Shield1 and 0.2 mM TA for 2 hr in the DRGFP U2OS cells. We have updated the experimental details in the revised manuscript on page 30, lines 724 – 727.

As the reviewer recommended, we performed a time course measurement through MeDIP and hMeDIP assay to show local TET-dependent methylation status by measuring 5mC and 5hmC levels. We discovered that while 5mC levels are generally stable among treatments, DSB results in a transient elevation in the 5hmC level about 2 hr after DSB induction. The 5mC level is elevated with the presence of D-2-HG and could be reversed with TET1 or TET2 overexpression.

The new findings are included as Figure 5e. The changes are made to the manuscript on page 18, lines 478 – 480.

We have included a model illustration for the findings as Extended Data Fig. 9.

The low signals in these areas might be caused by the short exposure time or lack of recognized antigens by antibodies. We have provided the original blotting image for Figure 1h.

	b. Figure 1L: For the DDK blot, why are lanes 1 and 4 so white? Again, the background greying shows loss in these regions.	We have provided the original blotting image for Figure 1L. c. Figure 4E: Part of the y-axis title seems to be missing based on f-h.	We have included the y-axis information with the resubmitted files.
Minor comments: 1. Language use and writing. a. At times the authors refer to gH2AX and pATM using the word expression. However, expression is generally reserved for processes associated with transcription and translation. Given gH2AX and pATM are phosphorylation based post-translational events it would improve clarity for the reader by selecting another word in place of expression, such as modification.	We appreciate the comment from the reviewer. We have revised the wording accordingly.
b. The title for figure three refers to “cytidine hypermethylation,” however, everywhere else in the manuscript it is cytosine, which is the standard for the field in referencing this process.	We have revised the wording to ensure consistency. The change has been made to the manuscript on page 29, line 718.
c. Line 416. “D-2-HG blockade on CTCF binding and concomitant DNA repair complex establishment (Fig. 4).” It is unclear what this sentence is intended to convey. As it is currently written, it seems incomplete.	We have revised the wording accordingly. The change has been made on page 19, lines 520 – 521.
The presented manuscript by Lang et al provides an important confirmation of the previously suggested role for TETs in DNA damage repair, which has been met with some skepticism based on the limited number of observations presented to date. In addition, the authors begin to elucidate some key aspects of the mechanistic basis by which TETs aid in DNA damage repair and demonstrate how this is relevant to IDH mutant malignancies.	We appreciate the comment from the reviewer. We have carefully gone through all the comments and will revise our work accordingly.

However, in its current form, the presentation and interpretation of the completed work requires some revision to ensure the claims are supported by the work and accurately reflect the included content (see comments above).	
References written in bold in this response are not currently included in the manuscript but should be cited	We have cited the reference as indicated.
Synopsis of comments The manuscript by Lang et al entitled "D-2-hydroxyglutarate establishes "BRCAness" through epigenetic reprogramming" identifies and begins to mechanistically explore a role for TET proteins in the regulation of DNA damage repair. Moreover, this work assesses the contribution of this pathway to the failed DNA damage repair evident in IDH mutant malignancy. These findings present important new findings to the field. However, there are some aspects of the work that require revision/clarification prior to publication. These are indicated in a brief synopsis of the major points below:	We appreciate the comment from the reviewer.
1. The authors need to validate whether the TET-dependent regulation of DNA damage repair is the sole mechanism lost in IDH mutant cells as they suggest, by demonstrating that double deletion of TET1 and TET2 prevents the ability of α-ketoglutarate to rescue the DNA damage repair defect in IDH mutant cells. They also need to acknowledge the other proposed mechanism (H3K9 methylation) in the Results section instead of only in the Discussion.	We have made the revisions as indicated above.
2. Data figure 5 and extended data figure 5 require revision to improve clarity as to the experiments performed to be able to interpret their contribution to the work.	We have made the revisions as indicated above.
3. The materials and methods, statistical analysis and experimental details need to be revised such that there is sufficient information for reproduction of the included work and interpretation of the results.	We have made the revisions as indicated above.
4. The manuscript requires editing such that the written text accurately reflects the experimental figures to ensure that the scientific claims are sufficiently substantiated.	We have made the revisions as indicated above.
5. Additional experiments are needed to demonstrate 1) the contribution of TET catalytic activity to the local response to double stranded break repair, 2) the time dependent establishment of 5hmC at sites of DNA damage.	We have made the revisions as indicated above.

Reviewer 5

References:

- 1 Xu, W. *et al.* Oncometabolite 2-hydroxyglutarate is a competitive inhibitor of alpha-ketoglutarate-dependent dioxygenases. *Cancer Cell* **19**, 17-30 (2011). <https://doi.org:10.1016/j.ccr.2010.12.014>
- 2 Flavahan, W. A. *et al.* Insulator dysfunction and oncogene activation in IDH mutant gliomas. *Nature* **529**, 110-114 (2016). <https://doi.org:10.1038/nature16490>
- 3 Rahme, G. J. *et al.* Modeling epigenetic lesions that cause gliomas. *Cell* **186**, 3674-3685 e3614 (2023). <https://doi.org:10.1016/j.cell.2023.06.022>
- 4 Lu, J. *et al.* CNTF receptor subunit alpha as a marker for glioma tumor-initiating cells and tumor grade: laboratory investigation. *J Neurosurg* **117**, 1022-1031 (2012). <https://doi.org:10.3171/2012.9.JNS1212>
- 5 Hilmi, K. *et al.* CTCF facilitates DNA double-strand break repair by enhancing homologous recombination repair. *Sci Adv* **3**, e1601898 (2017). <https://doi.org:10.1126/sciadv.1601898>
- 6 Lang, F. *et al.* CTCF prevents genomic instability by promoting homologous recombination-directed DNA double-strand break repair. *Proc Natl Acad Sci U S A* **114**, 10912-10917 (2017). <https://doi.org:10.1073/pnas.1704076114>
- 7 Fitieh, A., Locke, A. J., Mashayekhi, F., Khaliqdina, F., Sharma, A. K. & Ismail, I. H. BMI-1 regulates DNA end resection and homologous recombination repair. *Cell Rep* **38**, 110536 (2022). <https://doi.org:10.1016/j.celrep.2022.110536>
- 8 Sulkowski, P. L. *et al.* 2-Hydroxyglutarate produced by neomorphic IDH mutations suppresses homologous recombination and induces PARP inhibitor sensitivity. *Sci Transl Med* **9** (2017). <https://doi.org:10.1126/scitranslmed.aal2463>
- 9 Lu, Y. *et al.* Chemosensitivity of IDH1-Mutated Gliomas Due to an Impairment in PARP1-Mediated DNA Repair. *Cancer Res* **77**, 1709-1718 (2017). <https://doi.org:10.1158/0008-5472.CAN-16-2773>
- 10 Schvartzman, J. M. *et al.* Oncogenic IDH mutations increase heterochromatin-related replication stress without impacting homologous recombination. *Mol Cell* **83**, 2347-2356 e2348 (2023). <https://doi.org:10.1016/j.molcel.2023.05.026>
- 11 Flavahan, W. A. *et al.* Insulator dysfunction and oncogene activation in IDH mutant gliomas. *Nature* **529**, 110-+ (2016). <https://doi.org:10.1038/nature16490>
- 12 Arnould, C. *et al.* Loop extrusion as a mechanism for formation of DNA damage repair foci. *Nature* **590**, 660-665 (2021). <https://doi.org:10.1038/s41586-021-03193-z>

Comments:	Responses:
Reviewer 3	
In this revised manuscript by Lang et al some questions have been answered, but some have yet to be addressed. Major comment: Throughout the manuscript the authors use TMZ treatment at 24 hours (now added in some figure legends and methods). However, the mechanism for TMZ is to first add an O6-methyl adduct that will become a DSB after replication. Using 24-hour treatment is unlikely to result in a DSB as is evident with Figure 1a. While the IDH mutation does cause a robust increase in γH2A.X foci it's evident for those working with IDH mutant cells that they grow slower than IDH WT cells. The authors should at least discuss this in the discussion as more than likely this is some sort of non-canonical signaling – or show that a cell cycle has occurred with the IDH mutation.	We appreciate the reviewer's insightful comments regarding the timing and mechanisms of double-strand break (DSB) formation in response to TMZ, particularly in the context of IDH mutation. The dynamic γH2A.X formation is complicated and could vary among models. The elevation of γH2A.X signal within 24 hr has been reported in several independent studies [1-5]. Notably in the present study, the high dose of TMZ may result in non-canonical signaling pathways, such as replication stress-associated DSB, which contributes to DNA damage beyond the classical mechanisms. We have included the information in discussion, page 19, lines 536 – 537; page 20, lines 538 – 540.
While the theory, I'm gathering from the manuscript, is that D-2HG will block TET function (known) which will increase cytosine methylation (known) and impair CTCF binding (known) that ultimately affects BRCA2/Rad51 accumulation – how can the authors explain the rapid formation of DSBs post TMZ treatment in IDH mutant cells when a cell cycle is necessary for the DSBs to form? This mechanism makes sense for the micro irradiation assays where DSBs are formed rapidly, but becomes confusing for the TMZ treatment assays.	We appreciate the reviewer's comments. Many lines of evidence suggest that IDH-mutant cells are generally deficient in maintaining nucleotide pools and DNA repair activity [6-8], making them more susceptible to DNA replication challenges and DNA damage. The rapid accumulation of DSBs may result from a combination of replication stress, deficiencies in DNA repair pathways, and impaired detoxification of DNA-damaging agents.
Addition of the IDH mutant GSCs is a welcome and necessary addition to the manuscript. As in Figure 1 where the IDH inhibitor is used, this should also be included in the IDH mutant GSCs as the role of IDH itself isn't being tested, but rather very high concentrations of aKG and TET function. Which while TET function is related to IDH mutation, this is not the only pathway affected by IDH mutation.	Thank you for the reviewer's comments on the GSC experiment section. We have included the data in the new Extended Figure 8. We discovered that IDH mutant inhibitor AG-120 alleviated TMZ-induced DNA damage, which highlights the connection between IDH mutation and sensitivity to alkylating agent. The changes have been made to the manuscript on page 17, line 451-454.

The authors also need to include a graphical abstract of the how they believe this pathway is occurring. Extended figure 9 is not sufficient to map out how the authors believe this is occurring.	Thank you for the reviewer's comments. We have included a new illustration describing our findings with more detailed information in the Extended Figure 10.
Minor comments: Cell lines are still not listed throughout the manuscript in each panel. This is necessary. Eg. Figure 1 f, g, k; 3g, Figure 5.	We appreciate the comment from the reviewer. We have included the cell line model information in the figure panels.
TMZ treatment timing also needs to be added to the figures/legends. Eg. Figure 3a does not say how long TMZ treatment was used. Figure 4a-d.	We appreciate the comment from the reviewer. We have included the treatment information in the figure legends.
Extended figure 7k/m TB054 should be BT054.	We have revised the wording to ensure accuracy.
The authors should also include a limitations of the study section as the text suggests that this will impact IDH mutant gliomas, but there are no patient samples or mouse models to show that this same pathway will take place in a more realistic patient setting.	Thank you for the reviewer's comments on the limitation section. We agree that incorporating patient-derived samples or in vivo evidence would significantly strengthen our conclusions and provide more translational relevance. We acknowledge this limitation and will aim to address it in future studies by incorporating patient-derived glioma samples or preclinical models to validate our findings in a more physiologically relevant context. We have included this in the manuscript on page 20, lines 554 – 556.

Reviewer 4	
Lang et al., Nature Communications manuscript revised NCOMMS-24-13290-T, -2-Hydroxyglutarate establishes BRCAness through epigenetic reprogramming The authors have undertaken a significant amount of additional work to experimentally address our comments. However, there still remain some concerns in response to the rebuttal which have been highlighted below. To not delay the publishing of the important new findings presented in this manuscript, it is recommended that the manuscript be accepted given that the following points are adequately addressed within the text and figures of the manuscript.	We appreciate the comment from the reviewer.
In response to our previous comment number 3 noting concerns over the lack of detail included with respect to the experimental design, statistics and materials and methods, the authors have updated some aspects of the paper to include more detail. However, there is still information missing and some of the information provided has raised additional concerns.	We appreciate this valuable comment. We would be happy to provide additional details for clarity.
1) For the majority of western blots presented within the paper there is no indication of how many times these experiments were repeated or whether this was only completed one time.	Thanks for the reviewer for the comments. We performed at least two times of the western blotting experiments and selected representative images for the presentation.
2) The majority of figures list a student's t test as the primary statistical analysis used to determine significance between treatment groups. This analysis is not appropriate for experiments containing more than two groups. Alternatively, an ANOVA should be used to determine statistical significance and then the correct post-hoc analysis, which corrects for multiple comparisons, should be employed to determine which treatment groups are responsible for the detected differences. Within the main text this applies to figures 1a, 1f, 1g, 1k, 1m, 1n, 2c, 2d, 3b, 3e, 3f, 4e, 4f, 4g, 4h. Within the supplement this applies to 1d, 1h, 3a, 3b, 3e, 4c, 4h, 5b, 5d, 6b, 6c, 7c, 7d, 7e, 7f, 7h, 7q, 7r, 8d.	We appreciate the reviewer's advice for the statistical analysis. We have re-performed the data analysis with ANOVA and post-hoc analysis in the indicated figures. All the significance among the comparisons is consistent. There is one minor change in Figure 1m as follows. The p value between TET1 KO and TET1 KO+ TET1 OE in Fig. 1m is changed from ** to *. The change has been included in Fig. 1m.

3) For the quantification of microscopy images, it is unclear what the quantification within figures represent. For example, figure 2c states forty cells among three biological replicates were obtained, but the figure looks to have ~ 6 data points. Why does the figure have 6 data points? How does that relate to the number of cells and independent experiments evaluated? There is similar confusion for figures 2d, 3e, 3f. This should be clarified within the paper to make it clear to the reader how the data was prepared and analyzed.	Thanks for the reviewer's comments. We collected 5 – 7 images from independent experiments, containing a total of at least 40 laser-irradiated cells for quantification. For each image, we calculated the proportion of cells showing colocalization of the targeted proteins with the γH2A.X line signals. The data from all images were then combined to finalize the quantification.
In response to our previous comment number 2 noting concerns over the clarity and interpretation of figure five, the authors have added some clarifying information. However, more information is need for the readers to be able to interpret the results. 1) For figure 5b, the scale for the genomic site (units included) and exact genomic coordinates of the genome position should be provided for these types of analyses. This also applies to supplemental figures 8e, f, and h. Additionally, the CTCF enrichment tracks show a CpG site but that would include one "CG" in the genome and does not seem accurate given that figure 5C suggests there are 4 CpGs for DSB1 and 12 for DSB2. It would be preferable to show the exact CpGs within the region or to rename this to CpG island, although it is unclear based on the information provided if these genomic regions meet the criteria to be defined as CpG islands.	We appreciate the reviewer's suggestion for a better presentation of the figures. We have included the genomic site (units included) and exact genomic coordinates of the genome position in figure 5b, supplemental figures 9e, 9f, and 9h for better understanding. We have included the sequence information in the supplementary files and highlighted the CpGs within the regions. The precise coordinates of DSB1 and DSB2 are included in the manuscript on page 17, lines 471 – 472.

References

1. Ganesa, S., A. Sule, R.K. Sundaram, and R.S. Bindra, *Mismatch repair proteins play a role in ATR activation upon temozolomide treatment in MGMT-methylated glioblastoma*. Sci Rep, 2022. **12**(1): p. 5827.
2. Tomicic, M.T., et al., *Apoptosis induced by temozolomide and nimustine in glioblastoma cells is supported by JNK/c-Jun-mediated induction of the BH3-only protein BIM*. Oncotarget, 2015. **6**(32): p. 33755-68.
3. Gupta, S.K., et al., *Discordant in vitro and in vivo chemopotentiating effects of the PARP inhibitor veliparib in temozolomide-sensitive versus -resistant glioblastoma multiforme xenografts*. Clin Cancer Res, 2014. **20**(14): p. 3730-41.
4. Navarro-Carrasco, E. and P.A. Lazo, *VRK1 Depletion Facilitates the Synthetic Lethality of Temozolomide and Olaparib in Glioblastoma Cells*. Front Cell Dev Biol, 2021. **9**: p. 683038.
5. Yi, G.Z., et al., *Acquired temozolomide resistance in MGMT-deficient glioblastoma cells is associated with regulation of DNA repair by DHC2*. Brain, 2019. **142**: p. 2352-2366.
6. Dang, L., S. Jin, and S.M. Su, *IDH mutations in glioma and acute myeloid leukemia*. Trends Mol Med, 2010. **16**(9): p. 387-97.
7. Shi, D.D., et al., *De novo pyrimidine synthesis is a targetable vulnerability in IDH mutant glioma*. Cancer Cell, 2022. **40**(9): p. 939-956 e16.
8. Han, S., et al., *IDH mutation in glioma: molecular mechanisms and potential therapeutic targets*. Br J Cancer, 2020. **122**(11): p. 1580-1589.

Comments to the authors

The manuscript presented by Lang et al entitled “*D-2-Hydroxyglutarate establishes BRCAness through epigenetic reprogramming*” explores the molecular mechanism responsible for the diminished DNA repair ability of IDH mutant malignancies, and, therefore, sensitivity to therapeutic targeting by DNA repair inhibitors. The authors reproduce these preliminary findings in their model system and demonstrate that the diminished DNA repair observed in IDH mutant cells is dependent on the mutant IDH neomorphic activity that results in production of D-2-hydroxyglutarate (D-2-HG), a known potent inhibitor of α -ketoglutarate dependent demethylating enzymes (including those acting on DNA, histone, and RNA). As exogenous addition of α -ketoglutarate can rescue these defects in DNA damage repair of IDH mutant cells, the authors go on to specifically evaluate a role for the DNA demethylating enzymes, TETs. They demonstrate overexpression of TET1 or TET2 can partially rescue the defect in DNA damage repair caused by D-2-HG production, suggesting a role for TETs in DNA damage repair. This role is additionally supported by the observations that TET1 or TET2 deficient cells also exhibit defects in DNA damage repair, and that sites of DNA damage accumulate the TET enzymatic product, 5hmC. Moreover, they show that local DNA damage-induced recruitment of known proteins associated with DNA repair by homologous recombination (including CTCF, BRCA2, and RAD51) is TET-dependent. Lastly, the authors explore whether hypermethylation-dependent loss of CTCF alters high-order chromatin structure changes in IDH mutant cells that are associated with double stranded breaks.

Together these findings demonstrate a clear role for TET enzymes in the repair of DNA damage by homologous recombination. This has been previously suggested based on the reported co-localization of 5hmC with γ H2AX at sites of DNA damage (Kafer, G.R., et al. Cell Rep 2016), and the TET catalytic-dependent regulation of DNA damage repair in response to phosphorylation by the DNA damage sensors ATR/ATM (Jiang, D., et al. Brain, 2015 & Jiang, D., et al. EMBO Reports, 2017). The authors are ultimately able to corroborate and validate these previously reported findings using multiple models of DNA damage (Genotoxic stress, Aphidicolin, DRGFP reporter system, and DivA system). In addition, the authors are able to extend this observation to a new biological setting by demonstrating the loss of TET-dependent regulation of DNA damage in IDH mutant cells contributes to the mechanistic basis of their defects in DNA damage repair. Moreover, the authors are able to begin to unravel mechanistically how TET proteins contribute to DNA damage repair by aiding in the recruitment and assembly of DNA damage repair machinery (CTCF, BRCA2, and Rad51), which has not previously been reported to our knowledge. In total, this work advances our understanding of the contribution of TETs to DNA damage repair and establishes a novel molecular mechanism that underlies IDH mutant malignancies.

Major Comments:

1. Is TET-dependent regulation of DNA damage repair the primary mechanism lost in IDH mutant cells?

The foundational premise of the manuscript is based on the observation that the defect in DNA damage repair in IDH mutant cells is due to D-2-HG production, which inhibits α -ketoglutarate demethylating enzymes, and that this defect can be rescued by exogenous addition of α -ketoglutarate. The authors explore the contribution of the TET DNA demethylating proteins to this defect by individual reconstitution of TET1 or TET2 by overexpression. While this provides some rescue over treatment with D-2-HG alone, the extent of rescue is variable between experiments (1f, 1j, 2c, 2d, 3b, 3e, 3f, 3g) and based on the figures does not always reach the level of rescue achieved with exogenous addition of α -ketoglutarate. As the text implies that D-2-HG-dependent defects in DNA damage repair are TET-dependent, it would be important to address this discrepancy in the ability of either TET1 or TET2 to rescue this defect compared to α -ketoglutarate. Is this discrepancy due to inefficient expression of TET proteins, the need to have both TET1 and TET2 to achieve full rescue or the involvement of other pathways in D-2-HG-dependent inhibition of DNA damage? As previous mechanisms have been reported to contribute to the D-2-HG-dependent inhibition of DNA damage repair (Sulkowski, P.L., et al. Nature 2020, Wang, P., et al. Cell Reports, 2015), it would be important to address, experimentally, whether rescue relies on restoration of both TETs or a contribution of these other pathways. This is especially important as in the discussion the authors describe the mechanism identified in this manuscript to be a “parallel” mechanism to that described within Sulkowski, P.L., et al (Line 429).

- a. While simultaneous reconstitution of both TET1 and TET2 would be functionally difficult to achieve to assess rescue, the authors should do the dual TET1 and TET2 knockout within their experimental model system of TMZ treatment with D-2-HG and assess whether this double knockout completely abrogates the ability of α -ketoglutarate to rescue the defect in DNA damage repair. Additionally, they could assess aspects of the histone demethylase mechanism described by Sulkowski et al and determine if this mechanism is active in their model system. This could be achieved by evaluating H3K9me3 deposition around DNA breaks in experiments similar to those in Figure 2h and 3g.
- b. These experiments would provide clarity to whether the TET-dependent mechanism is the sole mechanism active within this model of IDH mutant malignancy as is currently implied. Should inhibition of TET function be the sole mechanism responsible for the observed defects in DNA damage repair in this system in response to D-2-HG, the authors should discuss 1) the relative differences between their model of IDH mutant malignancy and the other previously described models, 2) why these different IDH mutant models rely on distinct molecular mechanisms for D-2-HG disruption of DNA damage repair, and 3) what these differences mean with respect to understanding IDH mutant defects in DNA repair. If, however, both mechanisms are active in this model system, then the presence of this other contributing mechanism should be more clearly indicated in the results.

2. Clarity and interpretation of figure 5 and extended data figure 5. As they are currently presented, these figures have limited details present in the legend/figures that make technical and biological interpretation difficult and raise additional questions that require addressing/clarification.

- a. Extended data figure 5 shows the 4C signal and CTCF ChIP for two selected double stranded breaks. This figure could benefit from the addition of a distance scale to give the reader some indication of the physical spread of contacts and CTCF binding sites in the WT and IDH mutant cells at baseline and in response to DNA damage.
 - i. As it is presented, the CTCF ChIP peaks appear to be from untreated WT and IDH mutant cells. Is this correct? Has the CTCF peak status been assessed in the 4OHT treated cells to specifically evaluate changes in CTCF localization around the potential sites of breaks in response to damage? While the authors show the 4-OHT treatment and CTCF peaks in Figure 5d, this is for bulk localization and does not show individual CTCF peaks around the potential and actual break sites.
- b. Figure 5e is bisulfite sequencing for IDH WT and mutant cells. Additional information in the legend would aid readers unfamiliar with this technique in interpreting this analysis. What do dark and light circles indicate? What are the column and row numbers indicating? Additionally, are the presented results for the cells with or without 4-OHT?
 - i. The authors should include the sequence of the selected CTCF peaks for this methylation analysis. In this sequence they should indicate the potential sites of methylation and the predicted/actual CTCF binding site to aid the readers in determining whether there is specifically disruption of CTCF binding based on increased methylation within the CTCF consensus sequence as the authors hypothesized.
 - ii. This figure aims to support the connection between methylation status and CTCF binding across the genome impacting high-order chromatin structure. However, the presented data is evaluating CpG methylation status for one author-selected CTCF peak (indicated in extended data figure 5) flanking a double stranded break for two double stranded breaks in a model system that has ~100 breaks. Based on this limited analysis, it is difficult to ascertain whether these results are largely representative of the relationship between DNA hypermethylation and CTCF binding around potential sites of double stranded breaks. Additionally, as it is presented, it is unclear as to whether the observed hypermethylation is at baseline or in response to a double stranded break in the WT and IDH mutant cells making it difficult to interpret the biological meaning of these results. The authors should expand their analysis to further substantiate the claims that DNA hypermethylation is preventing CTCF binding and therefore higher-order chromatin structures.
 - iii. To that end, does the methylation status in this experiment change with or without 4OHT treatment?
- c. Figure 5f/g. There is no scale on the y-axis to discern the relative peak size across treatment groups. What do the asterisks and arrow in the figure indicate? This could also benefit from a genomic distance scale. Is f the data for the break one and g for the break two from figure 5e?

- d. Figure 5h/i. What is the difference between the left and right plots for figure h/i and what is the relationship of these plots to f and g? How was the 4C signal quantitated? Is this taking all signal from f/g or simply the yellow highlighted sections?

3. Experimental design, statistics, materials and methods.

- a. As the paper is currently written there is no information regarding the number of independent experiments that have been completed for any of the experiments/figures. While the methods denote that the presented data are the mean \pm SEM there is no indication for individual figures as to the number of independent experiments that have been performed and whether the presented data are representative of independent experiments or a compiled analysis of all independent experiments.
- b. The statistical section of the methods simply says t-tests and one-way ANOVA were used. Interpretation of the analysis would be improved by providing the individual test used for each analysis within the figure legends. Additionally, the presentation of statistical measurements across figures is variable and for some figures it is difficult to ascertain what the statistical comparison being presented is. For example: 1f uses lines between compared groups, 1k uses brackets, 3b uses asterisks without a line such that the comparison is unclear. A unified presentation of the statistical analysis would make it easier for the reader to interpret.
- The methods state statistical significance is set as less than 0.05. However, figures use both * and ** to indicate significance. What is the difference between these significance indications?
 - For figure 1K is the right bracket indicating that all of the 5 right groups are statistically significantly increased compared to the third group? The text seems to imply this is what is meant.
 - For figure 2C/D and 3E/F is the right line indicating the fifth group is statistically different from the second or is it that groups 3, 4, and 5 are statistically significantly increased compared to the second group? The text implies all three are able to rescue but the way that it is denoted by line would traditionally only indicate the two connected groups by the line are significantly different.
 - For figure 3b what is being compared for the presented statistical analysis?
 - Clarification of these statistical analyses will be important for ensuring proper analysis and biological interpretation of the experiments.
- c. The methods, as they are currently written, are not sufficiently detailed to allow for reproduction of all experimental procedures and some methods need to be added including: 5hmc immunofluorescence, siRNA experiments for knockdown, how the knockouts were generated for TET1 and TET2, the 4C data analysis and quantification, and primers used for ChIP with the DR-GFP reporter system.
- Additionally, it would dramatically aid readers to have a more detailed explanation of the individual components of the DR-GFP reporter and DivA systems to improve their understanding of these model systems. To that end, figures diagramming the general basis of these model systems would dramatically aid the reader in easily understanding their utility.

4. Conceptual contribution of Figure 5, and inconsistent/overstatement of scientific claims.

- a. In its current form figures 1-4 build to a model of TET-dependent regulation of DNA damage repair at the site of DNA damage that cohesively fit together. Figure 5, however, begins to explore the contribution of higher-order chromatin structures/TADs to the regulation of DNA damage and whether these are correlated with changes in methylation status/CTCF binding. While tangentially related, figure five feels like the first figure of a new paper focused on exploring the contribution of these structures to DNA damage repair, not the conclusion of the first four figures, especially considering major comment number 2. The impact of the presented work would benefit from further substantiating the claims in figures 1-4 and saving figure 5 for a future work where the contribution of these higher-order structures could be further evaluated and substantiated.
- Along these lines there seems to be competing presentation of two mechanistic bases for the TET-dependent regulation of DNA damage repair within the text: 1) methylation dependent changes in chromatin organization/structure that impairs DNA damage repair by changing overall CTCF localization, 2) methylation-dependent prevention of CTCF binding based on DNA sequence methylation and recruitment to local sites of damage. It is possible that both mechanisms contribute to DNA damage repair; however, the respective contributions are not immediately clear based upon the currently presented data.

- b. To this end, as it is currently written, the abstract is not representative of the findings and to some extent overstates the presented work and its conclusions. The synopsis of the work provided at the end of the introduction appears to be a more accurate reflection of the presented work. This should be addressed by adapting the abstract to be more reflective of the included findings or performing additional experiments to support these statements. As it is currently written, the abstract reads as though the CTCF high-order chromatin structure is directed by TETs and the loss of these structures forms the basis for failed DNA repair in IDH mutant cells. However, this manuscript, in its current form, does not establish these connections, but instead reveals a local TET-dependent recruitment of DNA repair proteins at sites of DNA damage (based on the local 5hmC deposition by IF and MeDIP, within a range of ~2000 bp around the cut site). Additionally, the work presents evidence to suggest IDH mutant cells also exhibit diminished remodeling of high-order chromatin structures (50kb-1Mb around cut site) in response to breaks (however with some caveat based on major comment 2), but the relationship of these structures to TET function and the contribution of these structures to DNA damage repair remain to be fully elucidated. If the authors wish to include additional experiments to support these statements, they would need to demonstrate the reliance of these damage induced contacts on TETs and CTCF potentially using the rescue by TETs and CTCF knockdowns used to establish the local relationship at sites of damage. Moreover, the contribution of these more distant contacts to the repair process would need to be assessed.
- c. Line 432. *“Overall, our findings reveal the molecular mechanisms of metabolic and epigenetic predisposition on “BRCAness” in malignancies.”* This is a strong statement that overstates the applicability of the work in its current form. While this work does reveal a novel contribution of TET-dependent regulation of the DNA damage response to IDH mutant malignancy, the presented work is largely confined to one mutant condition in one cell type and therefore statements of this broad utility to malignancy in total seem inaccurate. Additionally, considering other mechanisms have been proposed (see major comment 1) which are not assessed in this work, this mechanism, based on the data available at present, seems unlikely to be the sole basis of “BRCAness in malignancy”.

5. Additional experimentation/work to bolster findings.

- a. Given the 5hmC is co-localizing to these sites of DNA damage, it appears that catalytic activity of TETs is required for this process, which is further supported by the rescue with TET1 or TET2. These findings would be significantly bolstered by performing these rescue experiments with a catalytically dead (HxD) mutant to further confirm that local TET activity at the time of damage is required for this response.
- b. The time-dependent establishment of γ H2AX foci and the subsequent recruitment of CTCF, BRCA2, and RAD51 to sites of DNA damage in figure 3g very nicely contribute to the mechanistic basis of TET-dependent regulation of this response. Figure 3b connects these findings with the local deposition of 5hmC (and 5mC in Extended Figure 3c) using the same model system, although it is unclear at what time point in the response this information is collected? Additionally, is this performed only using GFP+ cells in this reporter system? The manuscript would benefit by performing a similar time course as experiment 3g for the 5hmC and 5mC to show the local TET-dependent regulation of DNA methylation in response to breaks. This would also reveal if TETs are acting simply to deplete existing 5mC at these sites.
- c. Addition of a model to collectively describe the included findings and their contribution to IDH mutant malignancy in the regulation of DNA damage repair would benefit the manuscript.

6. Experimental concerns.

- a. Figure 1H: In the TET1 western blot image, why are lanes 2 and 3 so white? Even the background greying shows loss in this area.
- b. Figure 1L: For the DDK blot, why are lanes 1 and 4 so white? Again, the background greying shows loss in these regions.
- c. Figure 4E: Part of the y-axis title seems to be missing based on f-h.

Minor comments:

1. Language use and writing.

- a. At times the authors refer to gH2AX and pATM using the word expression. However, expression is generally reserved for processes associated with transcription and translation. Given gH2AX and

pATM are phosphorylation based post-translational events it would improve clarity for the reader by selecting another word in place of expression, such as modification.

- b. The title for figure three refers to “cytidine hypermethylation,” however, everywhere else in the manuscript it is cytosine, which is the standard for the field in referencing this process.
- c. Line 416. “D-2-HG blockade on CTCF binding and concomitant DNA repair complex establishment (Fig. 4).” It is unclear what this sentence is intended to convey. As it is currently written, it seems incomplete.

The presented manuscript by Lang et al provides an important confirmation of the previously suggested role for TETs in DNA damage repair, which has been met with some skepticism based on the limited number of observations presented to date. In addition, the authors begin to elucidate some key aspects of the mechanistic basis by which TETs aid in DNA damage repair and demonstrate how this is relevant to IDH mutant malignancies. However, in its current form, the presentation and interpretation of the completed work requires some revision to ensure the claims are supported by the work and accurately reflect the included content (see comments above).

References written in bold in this response are not currently included in the manuscript but should be cited

Synopsis of comments

The manuscript by Lang et al entitled “*D-2-hydroxyglutarate establishes “BRCAness” through epigenetic reprogramming*” identifies and begins to mechanistically explore a role for TET proteins in the regulation of DNA damage repair. Moreover, this work assesses the contribution of this pathway to the failed DNA damage repair evident in IDH mutant malignancy. These findings present important new findings to the field. However, there are some aspects of the work that require revision/clarification prior to publication. These are indicated in a brief synopsis of the major points below:

1. The authors need to validate whether the TET-dependent regulation of DNA damage repair is the sole mechanism lost in IDH mutant cells as they suggest, by demonstrating that double deletion of TET1 and TET2 prevents the ability of α -ketoglutarate to rescue the DNA damage repair defect in IDH mutant cells. They also need to acknowledge the other proposed mechanism (H3K9 methylation) in the Results section instead of only in the Discussion.
2. Data figure 5 and extended data figure 5 require revision to improve clarity as to the experiments performed to be able to interpret their contribution to the work.
3. The materials and methods, statistical analysis and experimental details need to be revised such that there is sufficient information for reproduction of the included work and interpretation of the results.
4. The manuscript requires editing such that the written text accurately reflects the experimental figures to ensure that the scientific claims are sufficiently substantiated.
5. Additional experiments are needed to demonstrate 1) the contribution of TET catalytic activity to the local response to double stranded break repair, 2) the time dependent establishment of 5hmC at sites of DNA damage.

Major comments still not addressed:

In this revised manuscript by Lang et al some questions have been answered, but some have yet to be addressed.

Major comment: Throughout the manuscript the authors use TMZ treatment at 24 hours (now added in some figure legends and methods). However, the mechanism for TMZ is to first add an O6-methyl adduct that will become a DSB after replication. Using 24-hour treatment is unlikely to result in a DSB as is evident with Figure 1a. While the IDH mutation does cause a robust increase in γ H2A.X foci it's evident for those working with IDH mutant cells that they grow slower than IDH WT cells. The authors should at least discuss this in the discussion as more than likely this is some sort of non-canonical signaling – or show that a cell cycle has occurred with the IDH mutation. While the theory, I'm gathering from the manuscript, is that D-2HG will block TET function (known) which will increase cytosine methylation (known) and impair CTCF binding (known) that ultimately affects BRCA2/Rad51 accumulation – how can the authors explain the rapid formation of DSBs post TMZ treatment in IDH mutant cells when a cell cycle is necessary for the DSBs to form? This mechanism makes sense for the micro irradiation assays where DSBs are formed rapidly, but becomes confusing for the TMZ treatment assays.

Addition of the IDH mutant GSCs is a welcome and necessary addition to the manuscript. As in Figure 1 where the IDH inhibitor is used, this should also be included in the IDH mutant GSCs as the role of IDH itself isn't being tested, but rather very high concentrations of aKG and TET function. Which while TET function is related to IDH mutation, this is not the only pathway affected by IDH mutation.

The authors also need to include a graphical abstract of the how they believe this pathway is occurring. Extended figure 9 is not sufficient to map out how the authors believe this is occurring.

Minor comments:

Cell lines are still not listed throughout the manuscript in each panel. This is necessary. Eg. Figure 1 f, g, k; 3g, Figure 5.

TMZ treatment timing also needs to be added to the figures/legends. Eg. Figure 3a does not say how long TMZ treatment was used. Figure 4a-d.

Extended figure 7k/m TB054 should be BT054.

The authors should also include a limitations of the study section as the text suggests that this will impact IDH mutant gliomas, but there are no patient samples or mouse models to show that this same pathway will take place in a more realistic patient setting.

Major comments still not addressed:

In this revised manuscript by Lang et al some questions have been answered, but some have yet to be addressed.

Major comment: Throughout the manuscript the authors use TMZ treatment at 24 hours (now added in some figure legends and methods). However, the mechanism for TMZ is to first add an O6-methyl adduct that will become a DSB after replication. Using 24-hour treatment is unlikely to result in a DSB as is evident with Figure 1a. While the IDH mutation does cause a robust increase in γ H2A.X foci it's evident for those working with IDH mutant cells that they grow slower than IDH WT cells. The authors should at least discuss this in the discussion as more than likely this is some sort of non-canonical signaling – or show that a cell cycle has occurred with the IDH mutation. While the theory, I'm gathering from the manuscript, is that D-2HG will block TET function (known) which will increase cytosine methylation (known) and impair CTCF binding (known) that ultimately affects BRCA2/Rad51 accumulation – how can the authors explain the rapid formation of DSBs post TMZ treatment in IDH mutant cells when a cell cycle is necessary for the DSBs to form? This mechanism makes sense for the micro irradiation assays where DSBs are formed rapidly, but becomes confusing for the TMZ treatment assays.

Addition of the IDH mutant GSCs is a welcome and necessary addition to the manuscript. As in Figure 1 where the IDH inhibitor is used, this should also be included in the IDH mutant GSCs as the role of IDH itself isn't being tested, but rather very high concentrations of aKG and TET function. Which while TET function is related to IDH mutation, this is not the only pathway affected by IDH mutation.

The authors also need to include a graphical abstract of the how they believe this pathway is occurring. Extended figure 9 is not sufficient to map out how the authors believe this is occurring.

Minor comments:

Cell lines are still not listed throughout the manuscript in each panel. This is necessary. Eg. Figure 1 f, g, k; 3g, Figure 5.

TMZ treatment timing also needs to be added to the figures/legends. Eg. Figure 3a does not say how long TMZ treatment was used. Figure 4a-d.

Extended figure 7k/m TB054 should be BT054.

The authors should also include a limitations of the study section as the text suggests that this will impact IDH mutant gliomas, but there are no patient samples or mouse models to show that this same pathway will take place in a more realistic patient setting.